# Fair Clustering via Alignment

**Kunwoong Kim** [1]   **Jihu Lee** [1]   **Sangchul Park** [2]   **Yongdai Kim** [1]

## Abstract

Algorithmic fairness in clustering aims to balance the proportions of instances assigned to each cluster with respect to a given sensitive attribute. While recently developed fair clustering algorithms optimize clustering objectives under specific fairness constraints, their inherent complexity or approximation often results in suboptimal clustering utility or numerical instability in practice. To resolve these limitations, we propose a new fair clustering algorithm based on a novel decomposition of the fair $K$-means clustering objective function. The proposed algorithm, called Fair Clustering via Alignment (FCA), operates by alternately (i) finding a joint probability distribution to align the data from different protected groups, and (ii) optimizing cluster centers in the aligned space. A key advantage of FCA is that it theoretically guarantees approximately optimal clustering utility for any given fairness level without complex constraints, thereby enabling high-utility fair clustering in practice. Experiments show that FCA outperforms existing methods by (i) attaining a superior trade-off between fairness level and clustering utility, and (ii) achieving near-perfect fairness without numerical instability.

## 1. Introduction

As artificial intelligence (AI) technology has advanced and been successfully applied to diverse domains and tasks, the requirement for AI systems to make fair decisions (i.e., algorithmic fairness) has emerged as an important societal issue. This requirement is particularly necessary when observed data possess historical biases with respect to specific sensitive attributes, leading to unfair outcomes of learned models based on such biased data (Angwin et al., 2016;

Ingold & Soper, 2016; Damodaran et al., 2018; Mehrabi et al., 2019). Moreover, non-discrimination laws are also increasingly emphasizing the importance of fair decision making based on AI systems (Hellman, 2019). Specifically, group fairness is a category within algorithmic fairness that ensures models do not discriminate against certain protected groups, which are defined by specific sensitive attributes (e.g., race). In response, a large amount of research has been conducted to develop algorithms for mitigating such biases in various supervised learning tasks such as classification (Zafar et al., 2017; Donini et al., 2018; Agarwal et al., 2018; Quadrianto et al., 2019; Jiang et al., 2020) and regression (Agarwal et al., 2019; Chzhen et al., 2020).

Along with supervised learning, algorithmic fairness for unsupervised learning tasks, such as clustering, has also gathered significant interest. Clustering algorithms have long been employed as fundamental unsupervised learning methods for machine learning, such as recommendation systems (Widiyaningtyas et al., 2021), image processing (Le, 2013; Guo et al., 2020; Mittal et al., 2022), and language modeling (Butnaru & Ionescu, 2017; Zhang et al., 2023).

**Related works for Fair Clustering (FC)**   Combining algorithmic fairness and clustering, the notion of Fair Clustering (FC) was initially introduced in Chierichetti et al. (2017). FC operates under the goal that the proportion of each protected group within each cluster should be similar to that in the population. To achieve this goal, various algorithms have been developed to minimize a given clustering objective under pre-specified fairness constraints (Bera et al., 2019; Kleindessner et al., 2019; Backurs et al., 2019; Li et al., 2020; Esmaeili et al., 2021; Ziko et al., 2021; Zeng et al., 2023), to name a few.

We can roughly categorize the existing FC algorithms into three: (i) pre-processing, (ii) in-processing, and (iii) post-processing. Pre-processing methods (Chierichetti et al., 2017; Backurs et al., 2019) involve transforming instances into a fair space based on the concept of fairlets. Fairlets are small subsets that satisfy (perfect) fairness, and thus performing standard clustering on the fairlet space yields a fair clustering. In-processing methods (Kleindessner et al., 2019; Ziko et al., 2021; Li et al., 2020; Zeng et al., 2023) aim to simultaneously find both the cluster centers and assignments of the fair clustering by solving constrained op-

[1]Department of Statistics, Seoul National University, Republic of Korea [2]School of Law, Seoul National University, Republic of Korea. Correspondence to: Yongdai Kim <ydkim0903@gmail.com>.

*Proceedings of the 42nd International Conference on Machine Learning*, Vancouver, Canada. PMLR 267, 2025. Copyright 2025 by the author(s).

timization problems. Post-processing methods (Bera et al., 2019; Harb & Lam, 2020) focus on finding fair assignments given fixed cluster centers. The fixed cluster centers are typically predetermined by a standard clustering algorithm.

**Our contributions**   In this paper, we focus on the *trade-off between fairness level and clustering utility*: our goal is to maximize clustering utility while satisfying a given fairness level. While the trade-off between the fairness and utility is inevitable (Bertsimas et al., 2011; Chhabra et al., 2021), achieving the optimal trade-off with existing FC algorithms remains challenging. For example, pre- or post-processing algorithms usually result in suboptimal clustering utility due to indirect maximization of clustering utility (e.g., Backurs et al. (2019); Esmaeili et al. (2021)). Even when designed for achieving reasonable trade-off, in-processing algorithms may have trouble due to numerical instability, particularly when a given fairness level is high (e.g., Ziko et al. (2021)).

This paper aims to address these challenges by developing a new in-processing algorithm that can practically achieve a superior trade-off between fairness level and clustering utility without numerical instability. The primary idea of our proposed algorithm is to optimally align data from different protected groups by transforming them into a common space (called the aligned space), and then applying a standard clustering algorithm in the aligned space. We prove that the optimal fair clustering, i.e., the clustering with minimal clustering cost under a given fairness constraint, is equivalent to the optimal clustering in the aligned space.

Based on the theoretical result, we devise a new FC algorithm, called **Fair Clustering via Alignment (FCA)**[1]. FCA alternately finds the aligned space and the (approximately) optimal clustering in the aligned space until convergence. To find the aligned space, we develop a modified version of an algorithm for finding the optimal transport map (Kantorovich, 2006), while a standard clustering algorithm (e.g., the $K$-means++ algorithm (Arthur & Vassilvitskii, 2007)) is applied to find the (approximately) optimal clustering in the aligned space. It is worth noting that FCA can be particularly compared to the fairlet-based methods (e.g., Backurs et al. (2019)). While existing fairlet-based methods first find fairlets and then perform clustering sequentially, FCA simultaneously builds the aligned space and performs clustering to obtain an approximately optimal fair clustering. A detailed comparison is provided in Remark 3.2.

The main contributions of this paper can be summarized as:

◇ We provide a novel decomposition of the fair clustering cost into two components: (i) the transport cost with respect to a joint distribution between two pro-

tected groups, and (ii) the clustering cost with respect to cluster centers in the aligned space.
◇ Building on this decomposition, we develop a novel FC algorithm called FCA (Fair Clustering via Alignment), which is stable and guarantees convergence.
◇ Theoretically, we prove that FCA achieves an approximately optimal trade-off between fairness level and clustering utility, for any given fairness level.
◇ Experimentally, we show that FCA (i) outperforms existing baseline FC algorithms in terms of the trade-off and numerical stability, and (ii) effectively controls the trade-off across various fairness levels.

## 2. Preliminaries

**Notations**   Let $\mathcal{D} = \{(\mathbf{x}_i, s_i)\}_{i=1}^n$ be a given dataset (i.e., a set of observed instances), where $\mathbf{x}_i \in \mathbb{R}^d$ and $s_i \in \{0, 1\}$ are $d$-dimensional data and binary variable for the sensitive attribute, respectively. We denote $(\mathbf{X}, S)$ as the random vector whose joint distribution denoted as $\mathbb{P}$ is the empirical distribution on $\mathcal{D}$. Let $\mathbb{P}_s$ represents the conditional distribution of $\mathbf{X}$ given $S = s$. In this paper, we specifically define these distributions to discuss the (probabilistic) matching between two protected groups of different sizes. We denote $\mathbb{E}$ and $\mathbb{E}_s$ as the expectation operators of $\mathbb{P}$ and $\mathbb{P}_s$, respectively. Let $\mathcal{X} = \{\mathbf{x}_i\}_{i=1}^n$, $\mathcal{X}_s = \{\mathbf{x}_i \in \mathcal{X} : s_i = s\}$, and $n_s := |\mathcal{X}_s|$ for $s \in \{0, 1\}$. Denote $\|\cdot\|^2$ as the $L_2$ norm.

We assume that the number of clusters, represented by $K \in \mathbb{N}$, is given a priori. The $K$-many cluster centers are denoted as $\boldsymbol{\mu} := \{\mu_1, \ldots, \mu_K\}$ where $\mu_k \in \mathbb{R}^d, \forall k \in [K] = \{1, \ldots, K\}$. Let $\mathcal{A} : \mathcal{X} \times \{0, 1\} \to \mathcal{S}^K$ be an assignment function that takes as input $(\mathbf{x}, s) \in \mathcal{X} \times \{0, 1\}$ and returns the assignment probabilities over clusters for the data point $\mathbf{x}$, where $\mathcal{S}^K$ is the $(K-1)$-dimensional simplex. We consider this probabilistic assignment function to ensure the existence of a perfectly fair clustering.

**Clustering objective function**   We first present the mathematical formulation of the clustering objective function. The objective of the standard (i.e., fair-unaware) $K$-means clustering is to minimize the clustering cost $C(\boldsymbol{\mu}, \mathcal{A}) := \frac{1}{n} \sum_{k=1}^K \sum_{(\mathbf{x}, s) \in \mathcal{D}} \mathcal{A}(\mathbf{x}, s)_k \|\mathbf{x} - \mu_k\|^2$, with respect to $\boldsymbol{\mu}$ and $\mathcal{A}$. Note that $C(\boldsymbol{\mu}, \mathcal{A})$ can be equivalently re-written as $\mathbb{E} \sum_{k=1}^K \mathcal{A}(\mathbf{X}, S)_k \|\mathbf{X} - \mu_k\|^2$. Furthermore, the optimal assignment function is deterministic, i.e., $\mathcal{A}(\mathbf{x}, s)_k = \mathbb{1}(\arg\min_{k' \in [K]} \|\mathbf{x} - \mu_{k'}^\diamond\|^2 = k)$ for a given $(\mathbf{x}, s) \in \mathcal{X}_s \times \{0, 1\}$, where $\mu_1^\diamond, \ldots, \mu_K^\diamond$ are the centers of the optimal clustering. Thus, $C(\boldsymbol{\mu}, \mathcal{A})$ becomes $\mathbb{E} \min_k \|\mathbf{X} - \mu_k\|^2$, and the optimal clustering is obtained by finding $\boldsymbol{\mu}$ minimizing $\mathbb{E} \min_k \|\mathbf{X} - \mu_k\|^2$.

**Definition of fair clustering**   An assignment function $\mathcal{A}$ is said to be *fair* in view of group (or proportional) fairness

---

[1] Implementation code is available at https://github.com/kwkimonline/FCA.

(Chierichetti et al., 2017), if it satisfies for all $k \in [K]$,

$$\frac{\sum_{\mathbf{x}_i \in \mathcal{X}_0} \mathcal{A}(\mathbf{x}_i, 0)_k}{n_0} \approx \frac{\sum_{\mathbf{x}_j \in \mathcal{X}_1} \mathcal{A}(\mathbf{x}_j, 1)_k}{n_1}. \quad (1)$$

This constraint ensures the proportion of data belonging to a cluster be balanced, resulting in fair clustering. That is, we find the cluster center $\boldsymbol{\mu}$ and the assignment function $\mathcal{A}$ that minimize $C(\boldsymbol{\mu}, \mathcal{A})$ among all $\boldsymbol{\mu}$ and fair assignment functions $\mathcal{A}$ satisfying eq. (1).

To assess the fairness level in clustering, *Balance* measure is widely used (Chierichetti et al., 2017; Bera et al., 2019; Backurs et al., 2019; Esmaeili et al., 2021; Ziko et al., 2021; Zeng et al., 2023), which is defined as

$$\min_{k \in [K]} \min \left( \frac{\sum_{\mathbf{x}_i \in \mathcal{X}_0} \mathcal{A}(\mathbf{x}_i, 0)_k}{\sum_{\mathbf{x}_j \in \mathcal{X}_1} \mathcal{A}(\mathbf{x}_j, 1)_k}, \frac{\sum_{\mathbf{x}_j \in \mathcal{X}_1} \mathcal{A}(\mathbf{x}_j, 1)_k}{\sum_{\mathbf{x}_i \in \mathcal{X}_0} \mathcal{A}(\mathbf{x}_i, 0)_k} \right).$$

Note that the higher balance is, the fairer the clustering is. Furthermore, the balance of any given perfectly fair clustering is $\min(n_0/n_1, n_1/n_0)$. The objective of FC is to minimize $C(\boldsymbol{\mu}, \mathcal{A})$ with respect to $\boldsymbol{\mu} \in \mathbb{R}^K$ and $\mathcal{A}$, under the fairness constraint (e.g., balance $\approx \min(n_0/n_1, n_1/n_0)$). We abbreviate $\mathcal{A}(\cdot, s) = \mathcal{A}_s(\cdot)$ and $C(\boldsymbol{\mu}, \mathcal{A}) = C(\boldsymbol{\mu}, \mathcal{A}_0, \mathcal{A}_1)$ when their meanings are clear.

For the case of perfect fairness, in Section 3, we prove that the FC objective can be decomposed into the sum of (i) the cost of transporting data from different protected groups to a common space (called the aligned space), and (ii) the clustering cost in the aligned space built by the transported data. Building on the decomposition, in Section 4.1, we introduce our proposed algorithm for perfect fairness. Then, in Section 4.2, we extend the algorithm to control the fairness level by relaxing the perfect fairness constraint.

## 3. Reformulation of fair clustering objective

This section presents our main theoretical contribution: we derive a novel decomposition of the perfectly fair (i.e., $\sum_{\mathbf{x}_i \in \mathcal{X}_0} \mathcal{A}_0(\mathbf{x}_i)_k / \sum_{\mathbf{x}_j \in \mathcal{X}_1} \mathcal{A}_1(\mathbf{x}_j)_k = n_0/n_1$ for all $k \in [K]$) clustering objective, which motivates our proposed algorithms in Section 4.

In Section 3.1, we introduce our idea through discussing the simplest case where the two protected groups are of equal size ($n_0 = n_1$). Then, in Section 3.2, we generalize to the unequal case ($n_0 \neq n_1$) by constructing the aligned space defined by a given joint distribution on $\mathcal{X}_0 \times \mathcal{X}_1$. We prove that there exists a joint distribution such that the objective function of perfectly fair clustering can be decomposed into the sum of the transport cost with respect to the joint distribution and the clustering cost with respect to the cluster centers on the aligned space. Full proofs of all the theoretical findings in this section are given in Appendix B.

**3.1. Case of equal sample sizes:** $n_0 = n_1$

Assume that the sizes of two protected groups are equal (i.e., $n_0 = n_1$). We consider deterministic assignment functions, i.e., $\mathcal{A}_s(\mathbf{x})_k \in \{0, 1\}$, since the optimal perfectly fair assignment function is deterministic when $n_0 = n_1$ (see Remark 3.4 in Section 3.2). The case of probabilistic assignment functions when $n_0 \neq n_1$ is discussed in Section 3.2.

The core idea of FCA is to *match two instances from different protected groups and assign them to the same cluster*. By doing so – matching all instances from $\mathcal{X}_0$ to $\mathcal{X}_1$ in a one-to-one fashion and assigning each pair to the same cluster – the resulting clustering becomes perfectly fair.

Conversely, suppose we are given a perfectly fair clustering constructed by a deterministic assignment function $\mathcal{A}$. Since $n_0 = n_1$ and $\mathcal{A}$ is deterministic, there exists a one-to-one matching between $\mathcal{X}_0$ and $\mathcal{X}_1$ such that two matched instances belong to the same cluster. Thus, we can decompose the clustering cost in terms of the one-to-one matching, as presented in Theorem 3.1.

**Theorem 3.1.** *For any given perfectly fair deterministic assignment function $\mathcal{A}$ and cluster centers $\boldsymbol{\mu}$, there exists a one-to-one matching map $\mathbf{T} : \mathcal{X}_s \to \mathcal{X}_{s'}$ such that, for any $s \in \{0, 1\}$, $C(\boldsymbol{\mu}, \mathcal{A}_0, \mathcal{A}_1) =$*

$$\mathbb{E}_s \sum_{k=1}^{K} \mathcal{A}_s(\mathbf{X})_k \left( \underbrace{\frac{\|\mathbf{X} - \mathbf{T}(\mathbf{X})\|^2}{4}}_{\text{Transport cost w.r.t. } \mathbf{T}} + \underbrace{\left\| \frac{\mathbf{X} + \mathbf{T}(\mathbf{X})}{2} - \mu_k \right\|^2}_{\text{Clustering cost w.r.t. } \boldsymbol{\mu} \text{ and } \mathbf{T}} \right).$$

(2)

The assignment function $\mathcal{A}$ that minimizes eq. (2) given $\boldsymbol{\mu}$ and $\mathbf{T}$ assigns both $\mathbf{x}$ and $\mathbf{T}(\mathbf{x})$ to cluster $k$, where $k = \arg\min_{k' \in [K]} \|\frac{\mathbf{x} + \mathbf{T}(\mathbf{x})}{2} - \mu_{k'}\|^2$. Hence, the optimal perfectly fair clustering can be found by minimizing $\mathbb{E}_s \left( \|\mathbf{X} - \mathbf{T}(\mathbf{X})\|^2 / 4 + \min_k \|(\mathbf{X} + \mathbf{T}(\mathbf{X}))/2 - \mu_k\|^2 \right)$ with respect to $\boldsymbol{\mu}$ and $\mathbf{T}$, instead of finding the optimal $\boldsymbol{\mu}$ and $\mathcal{A}$ minimizing $C(\boldsymbol{\mu}, \mathcal{A}_0, \mathcal{A}_1)$. We update $\boldsymbol{\mu}$ for a given $\mathbf{T}$ by applying a standard clustering algorithm to $\{\frac{\mathbf{x} + \mathbf{T}(\mathbf{x})}{2}, \mathbf{x} \in \mathcal{X}_s\}$, which is called the *aligned space*. To update $\mathbf{T}$, any algorithm for finding the optimal matching can be used, where the cost between two instances $\mathbf{x}_0 \in \mathcal{X}_0$ and $\mathbf{x}_1 \in \mathcal{X}_1$ is given by $\|\mathbf{x}_0 - \mathbf{x}_1\|^2 / 4 + \min_k \|(\mathbf{x}_0 + \mathbf{x}_1)/2 - \mu_k\|^2$ (see Section 4.1 for the specific algorithm we use). Note that there are no complex constraints in updating $\boldsymbol{\mu}$ and $\mathbf{T}$. Finally, we define $\mathcal{A}$ corresponding to $\mathbf{T}$, which assigns both $\mathbf{x}$ and $\mathbf{T}(\mathbf{x})$ to the same cluster on the aligned space $\{\frac{\mathbf{x} + \mathbf{T}(\mathbf{x})}{2}, \mathbf{x} \in \mathcal{X}_s\}$. See Appendix A.4 for a similar decomposition that extends to other general distance metrics (e.g., the $L_p$ norm for $p \geq 1$).

*Remark* 3.2 (Comparison to the fairlet-based methods). Although our idea of matching data from different protected groups may seem similar to the fairlet-based methods

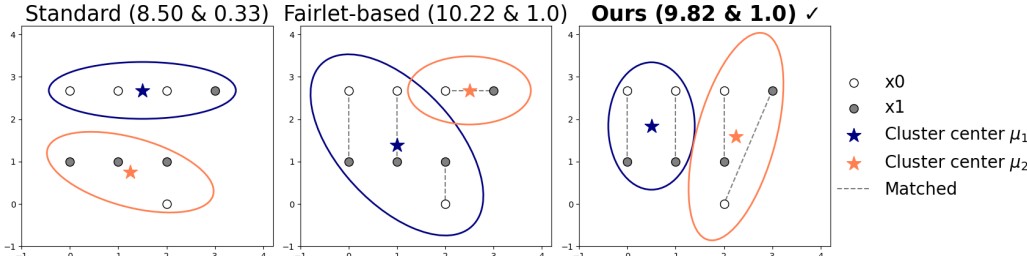

*Figure 1.* Comparison between the fairlet-based method and our approach with $n_0 = n_1 = 4$ and $K = 2$. The representative of each fairlet is set as the mean vector of the data within that fairlet, and the standard $K$-means algorithm is then applied to this set of representatives. The clustering results are visualized using contours. While both the fairlet-based method and ours result in perfectly fair clustering, i.e., balance (Bal) $= 1 = \min(n_0/n_1, n_1/n_0)$, our approach yields a lower cost ($9.82 < 10.22$), due to more efficient matchings.

(Chierichetti et al., 2017; Backurs et al., 2019), they fundamentally differ:

Our method is an *in-processing approach that directly minimizes the clustering cost with respect to both the matching map and cluster centers simultaneously*. In contrast, the fairlet-based method is a two-step pre-processing approach, which does not directly minimize the clustering cost; instead, it first searches for fairlets and then attempts to find the optimal cluster centers on the set of representatives for each fairlet. See Appendix A.1 for details about fairlets. As a result, in the fairlet-based method, the matchings and cluster centers are not jointly optimized, which may lead to suboptimal clustering utility.

Figure 1 illustrates how the fairlet-based method can produce suboptimal clustering, when compared to our approach. It implies that, more efficient matchings exist that yield higher clustering utility than the matchings of fairlets, and our approach is specifically designed to find these efficient matchings. We confirm this claim more comprehensively using real benchmark datasets in Section 5.2.

### 3.2. Case of unequal sample sizes: $n_0 \neq n_1$

To handle the case of $n_0 \neq n_1$, we follow a similar strategy to that for $n_0 = n_1$, but replace the matching map $\mathbf{T}$ with the joint distribution $\mathbb{Q}$ over $\mathbf{X}|S = 0$ and $\mathbf{X}|S = 1$. For each $s \in \{0, 1\}$, let $\mathbf{X}_s$ denote the random variable following the conditional distribution of $\mathbf{X}|S = s$, i.e., $\mathbb{P}_s$. We reformulate the perfectly fair clustering cost in terms of cluster centers $\boldsymbol{\mu}$ and joint distribution $\mathbb{Q}$ whose marginal distributions are $\mathbb{P}_s, s \in \{0, 1\}$. Note that $\mathbb{Q}$ serves as a smooth and stochastic version of $\mathbf{T}$.

Let $\mathcal{Q} = \{$all joint distributions $\mathbb{Q}$ on $\mathcal{X}_0 \times \mathcal{X}_1$ with $\mathbb{Q}_s = \mathbb{P}_s, s \in \{0, 1\}\}$, where $\mathbb{Q}_s$ is the marginal distribution of $\mathbb{Q}$ on $\mathbf{X}_s$. Theorem 3.3 below, which is the main theoretical result and motivation of our proposed algorithm, proves that the optimal perfectly fair clustering can be found by optimizing the joint distribution $\mathbb{Q}$ and cluster centers $\boldsymbol{\mu}$.

Let $\pi_s = n_s/(n_s + n_{s'})$ for $s \neq s' \in \{0, 1\}$. We then define

$$\mathbf{T}^A(\mathbf{x}_0, \mathbf{x}_1) := \pi_0 \mathbf{x}_0 + \pi_1 \mathbf{x}_1$$

as the *alignment map*.

**Theorem 3.3.** *Let $\boldsymbol{\mu}^* \in \mathbb{R}^d$ and $\mathbb{Q}^* \in \mathcal{Q}$ be the cluster centers and joint distribution minimizing*

$$\mathbb{E}_{\mathbb{Q}} \left( 2\pi_0 \pi_1 \|\mathbf{X}_0 - \mathbf{X}_1\|^2 + \min_k \|\mathbf{T}^A(\mathbf{X}_0, \mathbf{X}_1) - \mu_k\|^2 \right). \tag{3}$$

*Then, $(\boldsymbol{\mu}^*, \mathcal{A}_0^*, \mathcal{A}_1^*)$ is the solution of the perfectly fair $K$-means clustering, where $\mathcal{A}_0^*(\mathbf{x})_k := \mathbb{Q}^* \left(\arg\min_{k'} \|\mathbf{T}^A(\mathbf{x}, \mathbf{X}_1) - \mu_{k'}\|^2 = k | \mathbf{X}_0 = \mathbf{x}\right)$ and $\mathcal{A}_1^*(\mathbf{x})_k$ is defined similarly.*

This result implies that, by simultaneously minimizing the transport cost $\|\mathbf{X}_0 - \mathbf{X}_1\|^2$ and finding the cluster centers in the aligned space $\{\mathbf{T}^A(\mathbf{X}_0, \mathbf{X}_1) : (\mathbf{X}_0, \mathbf{X}_1) \sim \mathbb{Q}\}$, we obtain the optimal perfectly fair clustering. A notable observation is here that there is no explicit constraint in eq. (3). In fact, the constraint for perfect fairness (i.e., $\sum_{\mathbf{x}_i \in \mathcal{X}_0} \mathcal{A}(\mathbf{x}_i, 0)_k / \sum_{\mathbf{x}_j \in \mathcal{X}_1} \mathcal{A}(\mathbf{x}_j, 1)_k = n_0/n_1$ for all $k \in [K]$) is implicitly satisfied through the use of the alignment map $\mathbf{T}^A$ together with the assignment functions $\mathcal{A}_0^*$ and $\mathcal{A}_1^*$. In conclusion, solving eq. (3) with respect to $\boldsymbol{\mu}$ and $\mathbb{Q}$ yields the optimal perfectly fair clustering. The algorithm for solving eq. (3) is detailed in Section 4.

*Remark* 3.4 ($\mathcal{A}^*$ becomes deterministic when $n_0 = n_1$). Assume that $n_0 = n_1$. Then, for a given $\mathbf{x}_i \in \mathcal{X}_0$, note that $\gamma_{i,j}$ is positive for a unique $j \in [n_1]$, meaning that $\mathbb{Q}$ in Theorem 3.3 corresponds to the one-to-one matching map $\mathbf{T}$ in Theorem 3.1. In other words, finding $\mathbb{Q}$ becomes equivalent to optimizing the optimal permutation between $[n_0]$ and $[n_1]$. See Remark 2.4 in Peyré & Cuturi (2020) for the theoretical evidence. Then, we have $\mathcal{A}_0^*(\mathbf{x}_i)_k := \mathbb{Q}^* \left(\arg\min_{k'} \|\mathbf{T}^A(\mathbf{x}_i, \mathbf{X}_1) - \mu_{k'}\|^2 = k | \mathbf{X}_0 = \mathbf{x}_i\right) = \mathbb{1}(\arg\min_{k'} \|\frac{\mathbf{x}_i + \mathbf{T}(\mathbf{x}_i)}{2} - \mu_{k'}\|^2 = k)$, which is deterministic, provided that $\pi_0 = \pi_1 = 1/2$.

# 4. Proposed algorithms

## 4.1. FCA: Fair Clustering via Alignment

Based on Theorem 3.3, we propose an algorithm for finding the (approximately) optimal perfectly fair clustering, called **Fair Clustering via Alignment (FCA)**. FCA consists of two phases. Phase 1 finds the joint distribution $\mathbb{Q}$ with the cluster centers $\boldsymbol{\mu}$ being fixed. Phase 2 updates the cluster centers $\boldsymbol{\mu}$ with the joint distribution $\mathbb{Q}$ being fixed. Then, FCA iterates these two phases until cluster centers converge.

**Phase 1: Finding the joint distribution**   Phase 1 finds the optimal $\mathbb{Q}$ that minimizes the cost in eq. (3) given $\boldsymbol{\mu}$. For this goal, we modify the Kantorovich problem (Kantorovich, 2006; Villani, 2008), which finds the optimal coupling between two measures for a given cost matrix. Appendix A.2 covers details regarding the Kantorovich problem along with the optimal transport problem.

First, the transport cost matrix between the two (the first term of eq. (3)) is defined by $\mathbf{C} := [c_{i,j}] \in \mathbb{R}_+^{n_0 \times n_1}$ where $c_{i,j} = 2\pi_0\pi_1\|\mathbf{x}_i - \mathbf{x}_j\|^2$. The clustering cost matrix between the aligned data and their assigned centers (the second term of eq. (3)) is defined by $\mathbf{D} := [d_{i,j}] \in \mathbb{R}_+^{n_0 \times n_1}$ where $d_{i,j} = \min_{k \in [K]} \|\mathbf{T}^A(\mathbf{x}_i, \mathbf{x}_j) - \mu_k\|^2$. Then, we find the optimal coupling $\Gamma = [\gamma_{i,j}] \in \mathbb{R}_+^{n_0 \times n_1}$ solving

$$\min_\Gamma \|(\mathbf{C} + \mathbf{D}) \odot \Gamma\|_1 = \min_{\gamma_{i,j}}(c_{i,j} + d_{i,j})\gamma_{i,j}$$

$$\text{subject to} \sum_{i=1}^{n_0} \gamma_{i,j} = \frac{1}{n_1}, \sum_{j=1}^{n_1} \gamma_{i,j} = \frac{1}{n_0}, \gamma_{i,j} \geq 0. \quad (4)$$

Note that this problem becomes the original Kantorovich problem when $\mathbf{D} = \mathbf{0}$. Hence, this objective can be also efficiently solved using linear programming, similar to the Kantorovich problem (Villani, 2008).

Based on the optimal coupling $\Gamma$ that solves eq. (4), we define the joint distribution as $\mathbb{Q}(\{\mathbf{x}_i, \mathbf{x}_j\}) = \gamma_{i,j}$. That is, we have the measures (weights) $\{\gamma_{i,j}\}_{i\in[n_0], j\in[n_1]}$ for the aligned points in $\{\mathbf{T}^A(\mathbf{x}_i, \mathbf{x}_j)\}_{i\in[n_0], j\in[n_1]}$. As a result, we define the corresponding aligned space as the $n_0 \times n_1$ many pairs of the weight and the aligned points : $\{(\gamma_{i,j}, \mathbf{T}^A(\mathbf{x}_i, \mathbf{x}_j)\}_{i\in[n_0], j\in[n_1]}$.

**Phase 2: Optimizing cluster centers**   Phase 2 optimizes the cluster centers $\boldsymbol{\mu}$ on the aligned space $\{(\gamma_{i,j}, \mathbf{T}^A(\mathbf{x}_i, \mathbf{x}_j)\}_{i\in[n_0], j\in[n_1]}$ obtained from Phase 1, by solving $\min_{\boldsymbol{\mu}} \sum_{i=1}^{n_0} \sum_{j=1}^{n_1} \gamma_{i,j} \min_k \|\mathbf{T}^A(\mathbf{x}_i, \mathbf{x}_j) - \mu_k\|^2$. Standard clustering algorithms, such as the $K$-means++ algorithm (Arthur & Vassilvitskii, 2007) or a gradient descent-based algorithm, can be used to update $\boldsymbol{\mu}$. Note that in Section 5.4, we empirically show that FCA is stable regardless of the algorithm used to optimize $\boldsymbol{\mu}$. See Algorithm 1 for the overall algorithm of FCA.

---

**Algorithm 1** FCA algorithm

**input**  (i) Dataset $\mathcal{X}_0 \cup \mathcal{X}_1$. (ii) The number of clusters $K$.
 1: Initialize cluster centers $\boldsymbol{\mu} = \{\mu_k\}_{k=1}^K$.
 2: **while** $\boldsymbol{\mu}$ has not converged **do**
 3:   Update $\Gamma = [\gamma_{i,j}] \in \mathbb{R}_+^{n_0 \times n_1}$ by solving eq. (4)
     for a fixed $\boldsymbol{\mu}$.                    // Phase 1: update $\Gamma$
 4:   Update $\boldsymbol{\mu}$ by solving
     $\min_{\boldsymbol{\mu}} \sum_{i=1}^{n_0} \sum_{j=1}^{n_1} \gamma_{i,j} \min_k \|\mathbf{T}^A(\mathbf{x}_i, \mathbf{x}_j) - \mu_k\|^2$
     for a fixed $\Gamma$.                    // Phase 2: update $\boldsymbol{\mu}$
 5: **end while**
 6: Build fair assignments: for $\mathbf{x}_i \in \mathcal{X}_s$, define
    $\mathcal{A}_s(\mathbf{x}_i)_k := \sum_{\mathbf{x}_j \in \mathcal{X}_{s'}} n_s \gamma_{i,j} \mathbb{1}(\arg\min_{k'} \|\pi_s\mathbf{x}_i + \pi_{s'}\mathbf{x}_j - \mu_{k'}\|^2 = k), k \in [K]$.
**output**  (i) Cluster centers $\boldsymbol{\mu} = \{\mu_k\}_{k=1}^K$. (ii) Assignments $\mathcal{A}_0(\mathbf{x}_i), \mathbf{x}_i \in \mathcal{X}_0$ and $\mathcal{A}_1(\mathbf{x}_j), \mathbf{x}_j \in \mathcal{X}_1$.

---

In Appendix A.3, we introduce a practical extension of FCA to scenarios involving multiple protected groups and present experimental results on a real dataset that validate this extension (see Table 4).

## 4.2. FCA-C: control of fairness level

It is also important to find the optimal non-perfectly fair clustering. For this purpose, we introduce a feasible relaxation of FCA called **FCA-Control (FCA-C)**, a variant of FCA specifically designed for controlling the fairness level.

Let $\mathcal{W} \subset \mathcal{X}_0 \times \mathcal{X}_1$ be a given subset. The idea of our relaxation is to *apply FCA algorithm only to instances in $\mathcal{W}^c = \mathcal{X}_0 \times \mathcal{X}_1 \setminus \mathcal{W}$ and to apply the standard $K$-means algorithm to those in $\mathcal{W}$*. Note that $\mathcal{W} = \emptyset$ becomes FCA, while $\mathcal{W} = \mathcal{X}_0 \times \mathcal{X}_1$ results in standard (fair-unaware) clustering. For given $\boldsymbol{\mu}$ and $\mathcal{W}$, we define $C_{\text{FCA}}(\boldsymbol{\mu}, \mathcal{W}) := (2\pi_0\pi_1\|\mathbf{X}_0 - \mathbf{X}_1\|^2 + \min_k \|\mathbf{T}^A(\mathbf{X}_0, \mathbf{X}_1) - \mu_k\|^2) \cdot \mathbb{1}((\mathbf{X}_0, \mathbf{X}_1) \in \mathcal{W}^c)$ and $C_{K\text{-means}}(\boldsymbol{\mu}, \mathcal{W}) := (\min_k (\pi_0\|\mathbf{X}_0 - \mu_k\|^2) + \min_k (\pi_1\|\mathbf{X}_1 - \mu_k\|^2)) \cdot \mathbb{1}((\mathbf{X}_0, \mathbf{X}_1) \in \mathcal{W})$. Denote $\varepsilon > 0$ as a hyper-parameter that controls the fairness level. For a given $\varepsilon$, FCA-C algorithm minimizes

$$\mathbb{E}_{\mathbf{X}_0, \mathbf{X}_1 \sim \mathbb{Q}}(C_{\text{FCA}}(\boldsymbol{\mu}, \mathcal{W}) + C_{K\text{-means}}(\boldsymbol{\mu}, \mathcal{W})) \quad (5)$$

with respect to $\boldsymbol{\mu}, \mathbb{Q}$ and $\mathcal{W}$ satisfying $\mathbb{Q}((\mathbf{X}_0, \mathbf{X}_1) \in \mathcal{W}) \leq \varepsilon$. Figure 2 visualizes an example of $\mathcal{W}$ and $\mathbb{Q}$.

Then, we construct $\mathcal{A}$ as follows. For $\mathbf{x}_0 \in \mathcal{X}_0$, we define

$$\mathcal{A}_0(\mathbf{x}_0)_k =$$
$$\mathbb{P}_1(\arg\min_{k'} \|\mathbf{T}^A(\mathbf{x}_0, \mathbf{X}_1) - \mu_{k'}\|^2 = k, (\mathbf{x}_0, \mathbf{X}_1) \in \mathcal{W}^c)$$
$$+ \mathbb{1}(\arg\min_{k'} \|\mathbf{x}_0 - \mu_{k'}\|^2 = k) \cdot \mathbb{P}_1((\mathbf{x}_0, \mathbf{X}_1) \in \mathcal{W}),$$

and the assignment function $\mathcal{A}_1$ can be defined similarly.

$$(6)$$

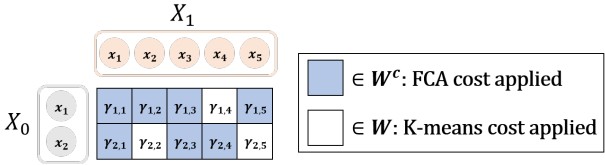

*Figure 2.* Example illustration of $\mathcal{W}$ and $\Gamma$ (or equivalently $\mathbb{Q}$) when $(n_0, n_1) = (2, 5)$ and $\varepsilon = \gamma_{1,4} + \gamma_{2,2} + \gamma_{2,5}$.

In summary, FCA-C comprises three phases: the first two mirror the Phases 1 and 2 of FCA, and the third updates $\mathcal{W}$. The overall algorithm for FCA-C is in Algorithm 2.

---

**Algorithm 2** FCA-C algorithm

---

**input** (i) Dataset $\mathcal{X}_0 \cup \mathcal{X}_1$. (ii) The number of clusters $K$. (iii) Fairness level $\varepsilon \in [0, 1]$.

1: Initialize cluster centers $\boldsymbol{\mu} = \{\mu_k\}_{k=1}^K$ and a subset $\mathcal{W} \subset \mathcal{X}_0 \times \mathcal{X}_1$ such that $\frac{1}{n_0 n_1} \sum_{i=1}^{n_0} \sum_{j=1}^{n_1} \mathbb{I}((\mathbf{x}_i, \mathbf{x}_j) \in \mathcal{W}) \le \varepsilon$.

2: **while** $\boldsymbol{\mu}$ has not converged **do**

3:     Calculate the costs $C_{K\text{-means}}$ for $(\mathbf{x}_i, \mathbf{x}_j) \in \mathcal{W}$ and $C_{\text{FCA}}$ for $(\mathbf{x}_i, \mathbf{x}_j) \in \mathcal{W}^c$.

4:     Update $\Gamma$ by minimizing eq. (5) for fixed $\boldsymbol{\mu}$ and $\mathcal{W}$.
                                 // Phase 1: update $\Gamma$

5:     Update $\boldsymbol{\mu}$ by minimizing eq. (5) for fixed $\Gamma$ and $\mathcal{W}$.
                                 // Phase 2: update $\boldsymbol{\mu}$

6:     For all $(\mathbf{x}_i, \mathbf{x}_j) \in \mathcal{X}_0 \times \mathcal{X}_1$, calculate $\eta(\mathbf{x}_i, \mathbf{x}_j) := 2\pi_0\pi_1 \|\mathbf{x}_i - \mathbf{x}_j\|^2 + \min_k \|\mathbf{T}^A(\mathbf{x}_i, \mathbf{x}_j) - \mu_k\|^2$. Let $\eta_\varepsilon$ be the $\varepsilon$th upper quantile. Update $\mathcal{W} = \{(\mathbf{x}_i, \mathbf{x}_j) \in \mathcal{X}_0 \times \mathcal{X}_1 : \eta(\mathbf{x}_i, \mathbf{x}_j) > \eta_\varepsilon\}$.
                                 // Phase 3: update $\mathcal{W}$

7: **end while**

8: Build fair assignment functions $\mathcal{A}_0$ and $\mathcal{A}_1$ following Equation (6).

**output** (i) Cluster centers $\boldsymbol{\mu} = \{\mu_k\}_{k=1}^K$. (ii) Assignments $\mathcal{A}_0(\mathbf{x}_i), \mathbf{x}_i \in \mathcal{X}_0$ and $\mathcal{A}_1(\mathbf{x}_j), \mathbf{x}_j \in \mathcal{X}_1$.

---

**Theoretical validity of FCA-C**   Notably, FCA-C algorithm achieves a solid theoretical guarantee for the (approximately) optimal trade-off between fairness level and clustering utility, for any given fairness level to be satisfied.

First, Theorem 4.1 shows that solving the objective of FCA-C algorithm in eq. (5) is equivalent to solve the clustering cost $C(\boldsymbol{\mu}, \mathcal{A})$ under a given fairness constraint, whose proof is given in Appendix B.3. For technical simplicity, assume that the densities of $\mathbb{P}_0$ and $\mathbb{P}_1$ exist (i.e., we consider the population version). See Remark B.1 in Appendix B when the densities do not exist. Recall that $\mathbb{E}_s(\mathcal{A}_s(\mathbf{X})_k) = \frac{1}{n_s} \sum_{\mathbf{x}_i \in \mathcal{X}_s} \mathcal{A}_s(\mathbf{x}_i)_k$. Let $\mathbf{A}_\varepsilon := \{(\mathcal{A}_0, \mathcal{A}_1) : \sum_{k=1}^K |\mathbb{E}_0(\mathcal{A}_0(\mathbf{X})_k) - \mathbb{E}_1(\mathcal{A}_1(\mathbf{X})_k)| \le \varepsilon\}$ represent a set of fair assignment functions. Let $\tilde{C}(\mathbb{Q}, \mathcal{W}, \boldsymbol{\mu})$ be the objective of FCA-C in eq. (5).

**Theorem 4.1** (Equivalence between $\tilde{C}$ and constrained $C$). *Minimizing FCA-C objective $\tilde{C}(\mathbb{Q}, \mathcal{W}, \boldsymbol{\mu})$ with the corresponding assignment function defined in eq. (6), is equivalent to minimizing $C(\boldsymbol{\mu}, \mathcal{A}_0, \mathcal{A}_1)$ subject to $(\mathcal{A}_0, \mathcal{A}_1) \in \mathbf{A}_\varepsilon$.*

Note that $\mathbf{A}_\varepsilon$ is based on the sum of unfairness for $k \in [K]$, whereas several previous works (e.g., Bera et al. (2019); Backurs et al. (2019)) focus on the proportions for all $k \in [K]$, which is more directly related to balance. However, $\mathbf{A}_\varepsilon$ is also closely connected to balance: (i) $\varepsilon = 0$ ensures perfect fairness (i.e., balance $= \min(n_0/n_1, n_1/n_0)$, and (ii) for all $k \in [K]$, the gap between $\sum_{\mathbf{x}_i \in \mathcal{X}_0} \mathcal{A}_0(\mathbf{x}_i)_k / \sum_{\mathbf{x}_i \in \mathcal{X}_1} \mathcal{A}_1(\mathbf{x}_i)_k$ and $n_0/n_1$ is bounded by $\varepsilon$, as shown in Proposition 4.2. Its proof is provided in Appendix B.4.

**Proposition 4.2** (Relationship between balance and $\varepsilon$). *For any assignment function $(\mathcal{A}_0, \mathcal{A}_1) \in \mathbf{A}_\varepsilon$, we have*

$$\max_{k \in [K]} \left| \frac{\sum_{\mathbf{x}_i \in \mathcal{X}_0} \mathcal{A}_0(\mathbf{x}_i)_k}{\sum_{\mathbf{x}_j \in \mathcal{X}_1} \mathcal{A}_1(\mathbf{x}_j)_k} - \frac{n_0}{n_1} \right| \le c\varepsilon, \qquad (7)$$

*where $c = \frac{n_0}{n_1} \max_{k \in [K]} \frac{1}{\mathbb{E}_1 \mathcal{A}_1(\mathbf{X})_k}$.*

Section 5.2 empirically validates Theorem 4.1 and Proposition 4.2, by showing that the trade-off between fairness (balance) and utility (cost) is effectively controlled by $\varepsilon$.

Lastly, we establish the approximation guarantee of FCA-C in Theorem 4.3, similar to Bera et al. (2019). The approximation error of a given algorithm is defined by an upper bound on the ratio by which the cost of the solution of the algorithm can exceed the optimal cost. Let $\tau$ be the approximation error of a standard clustering algorithm used to find initial cluster centers without the fairness constraint. Suppose that $\sup_{\mathbf{x} \in \mathcal{X}} \|\mathbf{x}\|^2 \le R$ for some $R > 0$.

**Theorem 4.3** (Approximation guarantee of FCA-C). *For any given $\varepsilon$, FCA-C algorithm returns an $(\tau + 2)$-approximate solution with a violation $3R\varepsilon$ for the optimal fair clustering, which is the solution of $\min_{\boldsymbol{\mu}, \mathcal{A}_0, \mathcal{A}_1} C(\boldsymbol{\mu}, \mathcal{A}_0, \mathcal{A}_1)$ subject to $(\mathcal{A}_0, \mathcal{A}_1) \in \mathbf{A}_\varepsilon$.*

A more formal statement is provided in Theorem B.2, with the proof in Appendix B.5. For example, if we use $K$-means++ initialization (Arthur & Vassilvitskii, 2007), then we have $\tau = \mathcal{O}(\log K)$. The violation term $3R\varepsilon$ suggests that FCA-C would be more efficient when $\varepsilon$ is small.

*Remark* 4.4 (Comparison of the approximation rate with existing approaches). The approximation rate $\tau + 2$ of FCA-C is comparable to that of existing approaches. For example, Bera et al. (2019) proposed an algorithm that achieves a $(\tau + 2)$-approximate solution with a violation of 3. FCA-C and the algorithm of Bera et al. (2019) provide the same rate of $\tau + 2$, but FCA-C can attain a smaller violation when $R = 1$, since $\varepsilon \in [0, 1]$. Furthermore, Schmidt et al. (2019) provided

a $(5.5\tau + 1)$-approximation algorithm, which exceeds $\tau + 2$ for $\tau > 2/9$.

## 4.3. Reducing computational complexity

As discussed in Section 4.1, finding the joint distribution $\mathbb{Q}$ is technically equivalent to solving the Kantorovich problem, which can be done by linear programming (Villani, 2008). Its computational complexity is approximately $\mathcal{O}(n^3)$ when $n_0 = n_1 = n$ (Bonneel et al., 2011), indicating a high computational cost when $n$ is large. To address this issue in practice, we randomly split each group into $L$ partitions of (approximately) same sizes: $\{\mathcal{X}_0^{(l)}\}_{l=1}^L$ of $\mathcal{X}_0$ and $\{\mathcal{X}_1^{(l)}\}_{l=1}^L$ of $\mathcal{X}_1$. We then calculate the optimal coupling $\Gamma^{(l)}$ between $\mathcal{X}_0^{(l)}$ and $\mathcal{X}_1^{(l)}$, by solving eq. (4). The (full) optimal coupling between $\mathcal{X}_0$ and $\mathcal{X}_1$ is estimated by $\hat{\Gamma} := \mathrm{diag}(\frac{1}{L}\Gamma^{(1)}, \ldots, \frac{1}{L}\Gamma^{(L)})$. Let $m = |\mathcal{X}_0^{(l)}| + |\mathcal{X}_1^{(l)}|$ be the partition size. This technique can theoretically reduce complexity by $\mathcal{O}(n/m) \times \mathcal{O}(m^3) = \mathcal{O}(nm^2)$. In our experiments, we apply this technique with $m = 1024$, which is significantly faster than using full datasets, while yielding similar performance (see Section 5.4 for the results).

## 5. Experiments

This section presents experiments showing that (i) FCA outperforms baseline methods in terms of the trade-off between fairness and utility; (ii) FCA-C effectively controls the fairness level; and (iii) FCA is numerically stable, efficient, and scalable. We further illustrate the applicability of FCA to visual clustering.

### 5.1. Settings

**Datasets and performance measures** We use three benchmark tabular datasets, ADULT (Becker & Kohavi, 1996), BANK (Moro et al., 2012), and CENSUS (Meek et al.), from the UCI Machine Learning Repository[2] (Dua & Graff, 2017). The sensitive attribute is defined by gender, marital status, and gender for ADULT, BANK, and CENSUS, respectively. The number of clusters $K$ is set to 10 for ADULT and BANK, and 20 for CENSUS, following Ziko et al. (2021). The features of data are scaled to have zero mean and unit variance. We then optionally apply $L_2$ normalization, as used in Ziko et al. (2021). Further details are given in Appendix C.1.

We consider two performance measures, Cost and Bal. The former assesses clustering utility, while the latter measures fairness level. For a given assignment function $\mathcal{A}$, let $\tilde{\mathcal{A}}_s(\mathbf{x})_k := \mathbb{1}(\arg\max_{k'} \mathcal{A}_s(\mathbf{x})_{k'} = k)$ be a deterministic version of $\mathcal{A}$. Note that we use this deterministic $\tilde{\mathcal{A}}$ instead of $\mathcal{A}$ itself for fair comparison with existing algo-

rithms. For a given $\mathbf{x}$, let $k(\mathbf{x}, s)$ be the index $k$ such that $\tilde{\mathcal{A}}_s(\mathbf{x})_k = 1$. Then, for given $\boldsymbol{\mu} = \{\mu_k\}_{k=1}^K$, Cost and Bal on $\mathcal{D}$ are defined as: Cost $= \frac{1}{n}\sum_{(\mathbf{x},s)\in\mathcal{D}} \|\mathbf{x} - \mu_{k(\mathbf{x},s)}\|^2$ and Bal $= \min_k \min(\tilde{r}_k, 1/\tilde{r}_k)$, where $\tilde{r}_k = \sum_{\mathbf{x}\in\mathcal{X}_0} \tilde{\mathcal{A}}_0(\mathbf{x})_k / \sum_{\mathbf{x}\in\mathcal{X}_1} \tilde{\mathcal{A}}_1(\mathbf{x})_k$. Let Bal$^\star := \min(n_0/n_1, n_1/n_0)$ be the balance of perfectly fair clustering for given $\mathcal{D}$.

**Baseline algorithms and implementation details** For the baseline algorithms compared with FCA, we consider four methods: a pre-processing (fairlet-based) method SFC from Backurs et al. (2019), two post-processing methods FCBC from Esmaeili et al. (2021) and FRAC from Gupta et al. (2023), and an in-processing method VFC from Ziko et al. (2021), which differs from the other baselines as it is specifically designed to control the trade-off between fairness level Bal and clustering utility Cost. For details of these baselines, refer to Appendix C.2.

When solving the linear program (i.e., finding the coupling matrix $\Gamma$), we use the POT library (Flamary et al., 2021). For finding cluster centers, we adopt the scikit-learn library (Pedregosa et al., 2011) to run the $K$-means algorithm. We specifically choose the $K$-means++ initialization (Arthur & Vassilvitskii, 2007) with Lloyd's algorithm (Lloyd, 1982). An ablation study comparing the Lloyd's algorithm and a gradient descent-based algorithm (i.e., Adam (Kingma & Ba, 2014)) for finding cluster centers is provided in Section 5.4. The maximum number of iterations is set to 100, and we select the best iteration when Cost is minimized.

### 5.2. Performance comparison results

**Trade-off: fairness level vs. clustering utility** First, we compare FC algorithms in terms of their ability to achieve reasonable trade-off between fairness level (Bal) and clustering utility (Cost). Specifically, we compare FCA with three baselines: SFC, FCBC, and FRAC, all of which are designed to achieve perfect fairness. Table 1 presents Bal and Cost of the four methods, where FCA consistently attains the lowest Cost (i.e., the highest utility).

Notably, FCA outperforms SFC by achieving higher Bal and lower Cost, highlighting the effectiveness of in-processing approach for finding better matchings. While FCA requires slightly more iterations until convergence (see Table 21 in Appendix C.3.8), it remains superior even with a smaller number of iterations similar to SFC (see Table 22 in Appendix C.3.8). Moreover, FCA also outperforms SFC in $K$-median clustering (see Table 23 in Appendix C.3.9), which is the original clustering objective of SFC. Table 8 in Appendix C.3.1 additionally shows that FCA also outperforms the first fairlet-based method introduced by Chierichetti et al. (2017). Furthermore, FCBC and

---

[2] https://archive.ics.uci.edu/datasets

*Table 1.* Comparison of the trade-off between `Bal` and `Cost` on ADULT, BANK, and CENSUS datasets. We underline `Bal` values for the cases of near-perfect fairness (i.e., `Bal` ≈ `Bal`$^\star$) and use **bold face for the lowest** `Cost` value among those cases. Similar results when data are not $L_2$-normalized are presented in Table 6 in Appendix C.3.1.

| Dataset / `Bal`$^\star$ | ADULT / 0.494 | | BANK / 0.649 | | CENSUS / 0.969 | |
|---|---|---|---|---|---|---|
| With $L_2$ normalization | `Cost` (↓) | `Bal` (↑) | `Cost` (↓) | `Bal` (↑) | `Cost` (↓) | `Bal` (↑) |
| Standard (fair-unaware) | 0.295 | 0.223 | 0.208 | 0.325 | 0.403 | 0.024 |
| FCBC (Esmaeili et al., 2021) | 0.314 | 0.443 | 0.685 | 0.615 | 1.006 | 0.926 |
| SFC (Backurs et al., 2019) | 0.534 | 0.489 | 0.410 | 0.632 | 1.015 | 0.937 |
| FRAC (Gupta et al., 2023) | 0.340 | 0.490 | 0.307 | 0.642 | 0.537 | 0.954 |
| FCA ✓ | **0.328** | 0.493 | **0.264** | 0.645 | **0.477** | 0.962 |

FRAC offer lower `Bal` and higher `Cost` than FCA in most cases. Specifically for a fixed `Cost` ≈ 0.314, FCA achieves a higher `Bal` than FCBC (see Table 7 in Appendix C.3.1). These results suggest that FCA is the most effective algorithm for maximizing utility under perfect fairness.

*Table 2.* Comparison of the two in-processing algorithms, VFC and FCA, in terms of numerical stability when achieving the maximum `Bal`, on ADULT, BANK, and CENSUS datasets with $L_2$ normalization. **Bold**-faced results are the highest `Bal`. Similar results without $L_2$ normalization are in Table 9 in Appendix C.3.1.

| | `Bal` (`Cost`) | |
|---|---|---|
| Dataset / `Bal`$^\star$ | VFC (Ziko et al., 2021) | FCA ✓ |
| ADULT / 0.494 | 0.437 (0.310) | **0.493** (0.328) |
| BANK / 0.649 | 0.568 (0.221) | **0.645** (0.264) |
| CENSUS / 0.969 | 0.749 (0.432) | **0.962** (0.477) |

**Numerical stability** Second, we compare the two in-processing algorithms, FCA and VFC, in terms of their ability to achieve a high (near-perfect) fairness level (i.e., `Bal` ≈ `Bal`$^\star$) without numerical instability. To do so, we obtain the maximum `Bal` that each algorithm can attain. We also evaluate robustness to data pre-processing, i.e., the impact of $L_2$ normalization to numerical stability.

The results presented in Table 2 show that VFC achieves `Bal` values no higher than 90% of `Bal`$^\star$. While FCA is explicitly designed to achieve perfect fairness, VFC is designed to control `Bal` using a hyper-parameter. However, even with a large hyper-parameter (see Appendix C.2 for the chosen hyper-parameter), VFC fails to achieve perfect fairness (i.e., `Bal` remains significantly lower than `Bal`$^\star$). Moreover, the performance gap between FCA and VFC becomes greater without $L_2$ normalization than with it, and VFC further fails on CENSUS dataset due to an overflow (see Appendix C.3.1 for details on this issue).

**Control of fairness level** In addition to the previous analysis for the scenario of perfect fairness (where `Bal` ≈ `Bal`$^\star$), we also compare FCA-C and VFC in terms of controlling `Bal` lower than `Bal`$^\star$. To this end, we assess the ability to

achieve reasonable `Cost` while controlling `Bal`.

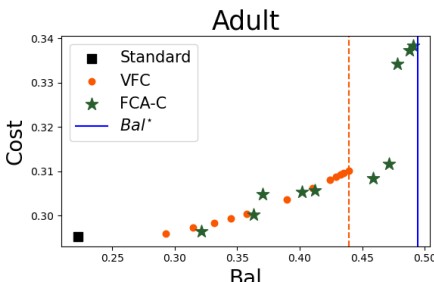

*Figure 3.* `Bal` vs. `Cost` trade-offs on ADULT dataset. Black square (■) is from the standard clustering, orange circle (●) is from VFC, green star (★) is from FCA-C, orange dashed line (- -) is the maximum of `Bal` that VFC can achieve, and blue line (–) is the maximum achievable balance `Bal`$^\star$.

Figure 3 displays the performance of FCA-C and VFC across various fairness levels, on ADULT dataset. It shows that FCA-C achieves favorable trade-off for many values of `Bal` (by controlling $\varepsilon$), with `Cost` similar to that of VFC. Similar results on other datasets or without $L_2$ normalization are given in Figures 4 and 5 in Appendix C.3.1. Furthermore, FCA-C can attain `Bal` beyond the orange dashed vertical line, which is the maximum achievable `Bal` by VFC. Refer to Appendix C.2 for details on selecting $\varepsilon$ for FCA and the hyper-parameters used in VFC.

Furthermore, the computation times of FCA-C (when $\varepsilon > 0$) and FCA (i.e., FCA-C with $\varepsilon = 0$) are compared in Table 20 in Appendix C.3.8, indicating that finding $\mathcal{W}$ in FCA-C does not require significant additional time.

### 5.3. Applicability to visual clustering

We further evaluate the applicability of FCA to visual clustering using two image datasets, which are commonly used in recent visual clustering algorithms (Li et al., 2020; Zeng et al., 2023): (i) RMNIST, a mixture of the original MNIST and a color-reversed version, and (ii) OFFICE-31, consisting of images from two domains.

We compare FCA with SFC, VFC and two visual FC baselines: DFC (Li et al., 2020) and FCMI (Zeng et al., 2023).

Note that DFC and FCMI are end-to-end algorithms that simultaneously perform clustering and learn latent space using autoencoder with fairness constraints. In contrast, FCA, SFC, and VFC are applied to a latent space pre-trained by autoencoder without any fairness constraints. The clustering utility is evaluated using ACC (accuracy based on assignment indices and ground-truth labels) and NMI (normalized mutual information between assigned cluster distribution and ground-truth label distribution), as in Zeng et al. (2023).

*Table 3.* Comparison of clustering utility (ACC) and fairness level (Bal) on two image datasets. 'Standard (fair-unaware)' indicates autoencoder + $K$-means. The first-place values are **bold**, and second-place values are underlined.

| Dataset
Bal⋆ | RMNIST
1.000 | | OFFICE-31
0.282 | |
|---|---|---|---|---|
| A = ACC, B = Bal | A (↑) | B (↑) | A (↑) | B (↑) |
| Standard (fair-unaware) | 41.0 | 0.000 | 63.8 | 0.192 |
| SFC (Backurs et al., 2019) | 51.3 | **1.000** | 61.6 | 0.267 |
| VFC (Ziko et al., 2021) | 38.1 | 0.000 | 64.8 | 0.212 |
| DFC (Li et al., 2020) | 49.9 | 0.800 | **69.0** | 0.165 |
| FCMI (Zeng et al., 2023) | 88.4 | 0.995 | **70.0** | 0.226 |
| FCA ✓ | **89.0** | **1.000** | 67.6 | **0.270** |

Table 3 compares FCA with the baseline methods, showing its performance is comparable to the state-of-the-art FCMI and generally better than SFC, VFC, and DFC. While DFC achieves slightly higher ACC on the OFFICE-31 dataset, its Bal is significantly lower (0.165 vs. 0.270 of FCA). Notably, while FCA operates as a two-step approach leveraging the pre-trained latent space, while DFC and FCMI are end-to-end methods. Moreover, FCA offers practical benefits: (i) requiring fewer hyper-parameters, reducing burden of hyper-parameter tuning in the absence of ground-truth labels, and (ii) adaptability with any pre-trained latent space.

Further details are provided in Appendix C.3.2, where Table 12 presents the full results of Table 3, which includes the similar superiority of FCA in terms of NMI values. Moreover, Table 13 highlights FCA's additional benefits over the fairlet-based method (SFC), in terms of matching efficiency.

### 5.4. Ablation studies

**(1) Selection of the partition size** $m$  We empirically confirm the efficiency of the partitioning technique in Section 4.3, by investigating the convergence of Bal and Cost with respect to the partition size $m$. In Appendix C.3.3, Figure 6 suggests that $m = 1024$ yields reasonable results, while Table 14 shows a significant reduction in computation time.

**(2) Optimization and initialization of cluster centers** $\mu$**:** We also analyze the stability with respect to (i) the choice of optimization algorithm for finding cluster centers, and (ii)

the initialization of cluster centers. Appendix C.3.4 shows that FCA is robust to both factors.

**(3) Consistent outperformance across various** $K$**:**  We confirm that FCA consistently outperforms baseline methods for various values of $K$. That is, as shown in Figure 8 in Appendix C.3.5, FCA outperforms the baseline methods across all $K \in \{5, 10, 20, 40\}$.

**(4) Scalability for large-scale data:**  We further investigate the adaptivity of FCA to large-scale data. To do so, we compare FCA and VFC using a large-scale dataset ($n = 10^6$), and observe that FCA still outperforms VFC (see Appendix C.3.6).

**(5) Linear program vs. Sinkhorn for optimizing** $\mathbb{Q}$  In our main experiments, we solve the objective in eq. (4) with the linear programming. To explore alternative approaches, we compare (i) FCA with the linear programming and (ii) FCA with the Sinkhorn algorithm, in terms of Cost, Bal, and runtime. Table 19 in Appendix C.3.7 shows that their performances are comparable; however, the Sinkhorn algorithm requires tuning a regularization parameter, suggesting that solving the linear program remains a practical and reliable option.

## 6. Conclusion and discussion

This paper has proposed FCA, an in-processing algorithm for fair clustering, inspired by the theoretical equivalence between optimal perfectly fair clustering and finding cluster centers in the aligned space. FCA algorithm is based on well-known algorithms, the $K$-means algorithm and linear programming, making it transparent and stable. Furthermore, we have developed FCA-C (a variant of FCA) to control fairness level, and established its approximation guarantee. Experimental results show FCA achieves near-perfect fairness, superior clustering utility, and robustness to data pre-processing, optimization methods, and initialization.

A promising direction for future work is to apply FCA to other clustering algorithms such as model-based clustering, e.g., Gaussian mixture, which we will pursue in near future.

## Acknowledgements

This work was partly supported by the National Research Foundation of Korea(NRF) grant funded by the Korea government(MSIT) (No. 2022R1A5A7083908), the National Research Foundation of Korea(NRF) grant funded by the Korea government(MSIT) (RS-2025-00556079), Next-generation Intelligence semiconductor R&D Program through the National Research Foundation of Korea(NRF) funded by the Korea government(MSIT)(RS-2024-00431718), and by Institute of Information & communications Technology Planning & Evaluation (IITP) grant funded by the Korea government(MSIT) [NO.RS-2021-II211343, Artificial Intelligence Graduate School Program (Seoul National University)].

## Impact Statement

This study aims to advance the field of algorithmic fairness in machine learning. In particular, given the growing application of clustering algorithms across diverse domains, we believe the proposed framework offers a promising approach to fair clustering. For example, it can ensure that privileged and unprivileged groups are clustered fairly, leading to unbiased downstream decision-making. Thus, this study is expected to yield a potentially positive impact rather than significant negative societal consequences.

Furthermore, incorporating the joint distribution in the proposed algorithm enables the identification of specific data that are matched and assigned together. This feature can enhance interpretability of fair clustering through the lens of implicit matching.

While this work focuses on proportional (group) fairness, we acknowledge that such an approach may overlook other fairness notions, such as individual fairness or social fairness (e.g., balancing clustering costs across groups), which are not explicitly addressed within the proportional fairness framework. We therefore encourage readers to interpret our results with this scope in mind: our goal is to improve the trade-off between proportional fairness level and clustering cost, but care must be taken when considering the broader implications for other fairness notions. In conclusion, we anticipate future studies that integrate fair clustering research with these additional notions of fairness.

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

# A. Supplementary discussion

## A.1. Fairlet-based methods

Fairlet decomposition was first introduced in Chierichetti et al. (2017), providing a pre-processing method for fair clustering. A fairlet is defined as a subset (as small as possible) of data in which the proportion of the sensitive attribute is balanced. The dataset is then partitioned into disjoint fairlets. It is important to note that the fairlets are not built arbitrarily; instead, the sum of distances among instances within each fairlet is minimized (i.e., each fairlet consists of similar instances). Finding such fairlets can be done by solving the minimum cost flow problem.

Notably, when $n_0 = n_1$, building fairlets is equivalent to finding the optimal coupling $\Gamma$ in the Kantorovich problem, which can be understood as a special case of minimum cost problem (Peyré & Cuturi, 2020; Chen et al., 2022). Hence, we can say that the fairlet-based methods find the coupling only once, then apply the standard clustering algorithm.

After building the fairlets, each fairlet is then deterministically assigned to a cluster. Since fairness is implicitly guaranteed within each fairlet, the resulting clustering directly becomes also fair. The representatives of each fairlet can be arbitrarily chosen when $n_0 = n_1$, or chosen by the medoids (or possibly the centroids) when $n_0 \neq n_1$, as suggested by Chierichetti et al. (2017). Then, a standard clustering algorithm is applied to the set of these chosen representatives.

On the other hand, the computational cost is high, with most of the time spent finding the fairlets due to the quadratic complexity of the minimum cost flow problem. To address this issue, SFC (Backurs et al., 2019) proposed a scalable algorithm for fairlet decomposition by using a reduction approach using metric embedding and trees. See Appendix C.2.3 for details on the SFC algorithm.

## A.2. Optimal transport problem

The notion of Optimal Transport (OT) provides a geometric view of the discrepancy between two probability measures. For two given probability measures $\mathcal{P}_1$ and $\mathcal{P}_2$, a map $\mathbf{T} : \mathrm{Supp}(\mathcal{P}_1) \to \mathrm{Supp}(\mathcal{P}_2)$ is defined as 'transport map' if $\mathbf{T}_{\#}\mathcal{P}_1 = \mathcal{P}_2$, where $\mathbf{T}_{\#}\mathcal{P}_1(A) = \mathcal{P}_1(\mathbf{T}^{-1}(A)), \forall A$, is the push-forward measure. The OT map is the minimizer of transport cost among all transport maps. That is, the OT map from $\mathcal{P}_1$ to $\mathcal{P}_2$ is the solution of $\min_{\mathbf{T}:\mathbf{T}_{\#}\mathcal{P}_1=\mathcal{P}_2} \mathbb{E}_{\mathbf{X}\sim\mathcal{P}_1} \left( c\left(\mathbf{X}, \mathbf{T}(\mathbf{X})\right)\right)$ for a pre-specified cost function $c$ (e.g., $L_2$ distance), which is so-called the Monge problem.

Kantorovich relaxed the Monge problem by seeking the optimal coupling (joint distribution) between two distributions. The Kantorovich problem is mathematically formulated as $\inf_{\pi \in \Pi(\mathcal{P}_1, \mathcal{P}_2)} \mathbb{E}_{\mathbf{X},\mathbf{Y}\sim\pi} \left( c(\mathbf{X}, \mathbf{Y}) \right)$ where $\Pi(\mathcal{P}_1, \mathcal{P}_2)$ is the set of all joint measures of $\mathcal{P}_1$ and $\mathcal{P}_2$. For two empirical measures, this problem can be solved by the use of linear programming as follows. For given two empirical distributions on $\mathcal{X}_0 = \{\mathbf{x}_i\}_{i=1}^{n_0}$ and $\mathcal{X}_1 = \{\mathbf{x}_j\}_{j=1}^{n_1}$, the cost matrix between the two is defined by $\mathbf{C} := [c_{i,j}] \in \mathbb{R}_+^{n_0 \times n_1}$ where $c_{i,j} = \|\mathbf{x}_i - \mathbf{x}_j\|^2$. Then, the optimal coupling (joint distribution) is defined by the matrix $\Gamma = [\gamma_{i,j}] \in \mathbb{R}_+^{n_0 \times n_1}$ that solves the following objective:

$$\min_{\Gamma} \|\mathbf{C} \odot \Gamma\|_1 = \min_{\gamma_{i,j}} c_{i,j}\gamma_{i,j} \text{ s.t. } \sum_{i=1}^{n_0} \gamma_{i,j} = \frac{1}{n_1}, \sum_{j=1}^{n_1} \gamma_{i,j} = \frac{1}{n_0}, \gamma_{i,j} \geq 0.$$

This problem can be solved by the use of linear programming. For the case of large $n$ with $n_0 \neq n_1$, various feasible estimators have been developed (Cuturi, 2013; Genevay et al., 2016), e.g., the Sinkhorn algorithm (Cuturi, 2013). Note only practical implementations, but theoretical aspects such as minimax estimation have also discussed deeply (Seguy et al., 2018; Yang & Uhler, 2019; Deb et al., 2021; Hütter & Rigollet, 2021). Recently, the OT map is utilized in diverse domains, for example, economics (Galichon, 2016; Chiappori et al., 2010), domain adaptation (Damodaran et al., 2018; Forrow et al., 2019), and computer vision (Su et al., 2015; Salimans et al., 2018). Several studies, including Jiang et al. (2020); Chzhen et al. (2020); Gordaliza et al. (2019); Kim et al. (2025), also employed the OT map in the field of algorithmic fairness for supervised learning.

## A.3. Extension to multiple protected groups

In this paper, we mainly focus on the case of two protected groups, for ease of discussion. However, it is possible to extend FCA to handle multiple (more than two) protected groups. The idea is equivalent to the case of two protected groups: matching multiple individuals from different protected groups.

**The case of equal sample sizes** Let $G$ be the number of protected groups, and denote $s^* \in \{0, \ldots, G-1\}$ as a fixed sensitive attribute as an anchor (a reference) attribute. For the case of $n_0 = \ldots = n_{G-1}$, we can similarly decompose the objective function in terms of matching map, as follows. First, we decompose the clustering objective similar to the proof of Theorem 3.1:

$$C(\boldsymbol{\mu}, \mathcal{A}_0, \ldots, \mathcal{A}_{G-1}) = \mathbb{E} \sum_{k=1}^K \mathcal{A}_S(\mathbf{X}) \|\mathbf{X} - \mu_k\|^2$$

$$= \frac{1}{G} \mathbb{E}_{s^*} \sum_{k=1}^K \mathcal{A}_{s^*}(\mathbf{X})_k \left( \|\mathbf{X} - \mu_k\|^2 + \sum_{s' \neq s^*} \|\mathbf{T}_{s'}(\mathbf{X}) - \mu_k\|^2 \right),$$

where $\mathbf{T}_{s'}$ is the one-to-one matching map from $\mathcal{X}_{s^*}$ to $\mathcal{X}_{s'}$ for $s' \in \{0, \ldots, G-1\}$. Note that $\mathbf{T}_{s'}$ then becomes the identity map from $\mathcal{X}_{s^*}$ to $\mathcal{X}_{s^*}$ for $s' = s^*$. Then for any given $\mathbf{x}$ and $\mu_k$, we can decompose $\|\mathbf{x} - \mu_k\|^2$ by

$$\left\| \frac{\sum_{s' \neq s^*} (\mathbf{x} - \mathbf{T}_{s'}(\mathbf{x}))}{G} \right\|^2 + \left\| \frac{\mathbf{x} + \sum_{s' \neq s^*} \mathbf{T}_{s'}(\mathbf{x})}{G} - \mu_k \right\|^2$$

$$+ 2 \left\langle \frac{\sum_{s' \neq s^*} (\mathbf{x} - \mathbf{T}_{s'}(\mathbf{x}))}{G}, \frac{\mathbf{x} + \sum_{s \neq s^*} \mathbf{T}_{s'}(\mathbf{x})}{G} - \mu_k \right\rangle.$$

We also similarly decompose $\|\mathbf{T}_{s'}(\mathbf{x}) - \mu_k\|^2$ by

$$\left\| \frac{\sum_{s'' \neq s', s'' \in \{0, \ldots, G-1\}} (\mathbf{T}_{s'}(\mathbf{x}) - \mathbf{T}_{s''}(\mathbf{x}))}{G} \right\|^2 + \left\| \frac{\mathbf{x} + \sum_{s' \neq s^*} \mathbf{T}_{s'}(\mathbf{x})}{G} - \mu_k \right\|^2$$

$$+ 2 \left\langle \frac{\sum_{s'' \neq s', s'' \in \{0, \ldots, G-1\}} (\mathbf{T}_{s'}(\mathbf{x}) - \mathbf{T}_{s''}(\mathbf{x}))}{G}, \frac{\mathbf{x} + \sum_{s' \neq s^*} \mathbf{T}_{s'}(\mathbf{x})}{G} - \mu_k \right\rangle.$$

By summing up the above $G$ many terms, we have the final result that

$$C(\boldsymbol{\mu}, \mathcal{A}_0, \ldots, \mathcal{A}_{G-1})$$

$$= \frac{1}{G} \mathbb{E}_{s^*} \sum_{k=1}^K \mathcal{A}_{s^*}(\mathbf{X})_k \left( \sum_{s=0}^{G-1} \left\| \frac{\sum_{s' \neq s} (\mathbf{T}_s(\mathbf{X}) - \mathbf{T}_{s'}(\mathbf{X}))}{G} \right\|^2 + \left\| \frac{\mathbf{X} + \sum_{s' \neq s^*} \mathbf{T}_{s'}(\mathbf{X})}{G} - \mu_k \right\|^2 \right).$$

Note that this result holds for any anchor $s^* \in \{0, \ldots, G-1\}$.

**The case of unequal sample sizes** For the case of unequal sample sizes, we also derive a similar decomposition (i.e., eq. (8)). We first define the alignment map as: $\mathbf{T}^A(\mathbf{x}_0, \ldots, \mathbf{x}_{G-1}) := \pi_0 \mathbf{x}_0 + \ldots + \pi_{G-1} \mathbf{x}_{G-1}$, where $\pi_s = n_s/n$ for $s \in \{0, \ldots, G-1\}$. Then, we find the joint distribution and cluster centers by minimizing the following objective:

$$\mathbb{E}_{(\mathbf{X}_0, \ldots, \mathbf{X}_{G-1}) \sim \mathbb{Q}} \left( \sum_{s=0}^{G-1} G \pi^G \left\| \sum_{s' \neq s} (\mathbf{X}_s - \mathbf{X}_{s'}) \right\|^2 + \min_k \|\mathbf{T}^A(\mathbf{X}_0, \ldots, \mathbf{X}_{G-1}) - \mu_k\|^2 \right), \tag{8}$$

where $\pi^G := \prod_{s=0}^{G-1} \pi_s$.

The proof idea is similar to the case of two protected groups, in Theorem 3.3. We first recall the original $K$-means objective: $C(\boldsymbol{\mu}, \mathcal{A}_0, \ldots, \mathcal{A}_{G-1}) := \mathbb{E} \sum_{k=1}^K \mathcal{A}_S(\mathbf{X})_k \|\mathbf{X} - \mu_k\|^2 = \sum_{s=0}^{G-1} \pi_s \mathbb{E}_s \sum_{k=1}^K \mathcal{A}_s(\mathbf{X}_s)_k \|\mathbf{X}_s - \mu_k\|^2$. Consider a set of joint distributions of $\mathbf{X}_0, \ldots, \mathbf{X}_{G-1}$ given $\boldsymbol{\mu}$ as $\mathcal{Q} := \{\mathbb{Q}(\{\mathbf{x}_0, \ldots, \mathbf{x}_{G-1}\}|\boldsymbol{\mu}) : \mathbf{x}_s \in \mathcal{X}_s, s \in \{0, \ldots, G-1\}\}$. Then, we can show that there exists a $\mathbb{Q} = \mathbb{Q}(\{\mathbf{x}_0, \ldots, \mathbf{x}_{G-1}\}|\boldsymbol{\mu}) \in \mathcal{Q}$ satisfying

$$C(\boldsymbol{\mu}, \mathcal{A}_0, \ldots, \mathcal{A}_{G-1}) = \sum_{s=0}^{G-1} \pi_s \mathbb{E}_s \sum_{k=1}^K \mathcal{A}_s(\mathbf{X}_s)_k \|\mathbf{X}_s - \mu_k\|^2$$

$$= \mathbb{E}_{(\mathbf{X}_0, \ldots, \mathbf{X}_{G-1}) \sim \mathbb{Q}} \left( \sum_{s=0}^{G-1} \pi_s \|\mathbf{X}_s - \mu_k\|^2 \right),$$

by using the same logic in the proof of Theorem 3.3. Furthermore, we can reformulate it as

$$\mathbb{E}_{(\mathbf{X}_0,\ldots,\mathbf{X}_{G-1})\sim\mathbb{Q}}\left(\sum_{s=0}^{G-1}G\pi^G\left\|\sum_{s'\neq s}(\mathbf{X}_s-\mathbf{X}_{s'})\right\|^2+\min_k\|\mathbf{T}^{\mathrm{A}}(\mathbf{X}_0,\ldots,\mathbf{X}_{G-1})-\mu_k\|^2\right),$$

with the assignment functions for fair clustering given as

$$\mathcal{A}_s(\mathbf{x}_s)_k:=\mathbb{Q}\left(\arg\min_{k'}\|\pi_s\mathbf{x}_s+\sum_{s'\neq s}\pi_{s'}\mathbf{X}_{s'}-\mu_{k'}\|^2=k\,\Big|\,\mathbf{X}_s=\mathbf{x}_s\right),\forall s\in\{0,\ldots,G-1\}.$$

Furthermore, since finding the joint distribution for multiple protected groups is technically similar to the case of two protected groups, the optimization of the objective in eq. (8) can be solved using a linear program.

**Experiments**  To empirically confirm the validity of the extension, we use BANK dataset with three protected groups defined by dividing marital status as single/married/divorced (directly following the approach of VFC (Ziko et al., 2021)). We then compare the performance of FCA with VFC, which also can handle multiple protected groups. The result is presented in Table 4 below, showing the superior performance of FCA: achieving a lower cost ($0.222 < 0.228$) along with a higher balance ($0.182 > 0.172$).

*Table 4.* Comparison of VFC and FCA in terms of the trade-off between `Bal` and `Cost`, on BANK dataset with three protected groups.

| BANK | 3 groups (single/married/divorced) | |
| --- | --- | --- |
| Bal$^\star$ = 0.185 | Cost ($\downarrow$) | Bal ($\uparrow$) |
| VFC | 0.228 | 0.172 |
| FCA ✓ | **0.222** | **0.182** |

### A.4. Extension to $K$-clustering for general distance metrics

Let $D:\mathbb{R}^d\times\mathbb{R}^d\to\mathbb{R}_+$ be a given distance satisfying (i) $D(\mathbf{v},\mathbf{w})\leq D(\mathbf{v},\mathbf{u})+D(\mathbf{u},\mathbf{w})$, (ii) $D(\mathbf{v},\mathbf{w})=D(\mathbf{v}+\mathbf{u},\mathbf{w}+\mathbf{u})$, and (iii) $D(\lambda\mathbf{v},\lambda\mathbf{w})=|\lambda|\,D(\mathbf{v},\mathbf{w})$, for all $\mathbf{v},\mathbf{w},\mathbf{u}\in\mathbb{R}^d$ and $\lambda\in\mathbb{R}$. For example, $D$ can be the $L_p$ norm on $\mathbb{R}^d$ for any $p\geq 1$. Then, for the balanced case $n_0=n_1$, we can replicate the argument of Theorem 3.1 as follows:

$$\mathbb{E}\sum_{k=1}^K\mathcal{A}_S(\mathbf{X})_kD(\mathbf{X},\mu_k)\leq\mathbb{E}_s\sum_{k=1}^K\mathcal{A}_S(\mathbf{X})_k\left(D\left(\frac{\mathbf{X}}{2},\frac{\mathbf{T}(\mathbf{X})}{2}\right)+D\left(\frac{\mathbf{X}+\mathbf{T}(\mathbf{X})}{2},\mu_k\right)\right).$$

Minimizing this upper bound yields a fair $K$-clustering procedure for any distance satisfying the above conditions. Note that using the $L_1$ norm for $D$ corresponds to the $K$-median clustering.

See Appendix C.3.9 for experiments based on this approach, showing the outperformance FCA over SFC in view of the $K$-median clustering.

# B. Proofs of the theorems

## B.1. Proof of Theorem 3.1

**Theorem 3.1.** *For any given perfectly fair deterministic assignment function $\mathcal{A}$ and cluster centers $\boldsymbol{\mu}$, there exists a one-to-one matching map $\mathbf{T} : \mathcal{X}_s \to \mathcal{X}_{s'}$ such that, for any $s \in \{0, 1\}$,*

$$C(\boldsymbol{\mu}, \mathcal{A}_0, \mathcal{A}_1) = \mathbb{E}_s \sum_{k=1}^{K} \mathcal{A}_s(\mathbf{X})_k \left( \frac{\|\mathbf{X} - \mathbf{T}(\mathbf{X})\|^2}{4} + \left\| \frac{\mathbf{X} + \mathbf{T}(\mathbf{X})}{2} - \mu_k \right\|^2 \right). \tag{9}$$

*Proof of Theorem 3.1.* Without loss of generality, let $s = 0$. First, it is clear that we can construct a one-to-one map $\mathbf{T}$ that maps each $\mathbf{x} \in \mathcal{X}^{k,0} := \{\mathbf{x} \in \mathcal{X}_0 : \mathcal{A}_0(\mathbf{x})_k = 1\}$ to a unique $\mathbf{x}' \in \mathcal{X}^{k,1} := \{\mathbf{x}' \in \mathcal{X}_1 : \mathcal{A}_1(\mathbf{x}')_k = 1\}$ for all $k \in [K]$. That is, $\{\mathbf{T}(\mathbf{x}) : \mathbf{x} \in \mathcal{X}^{k,0}\} = \mathcal{X}^{k,1}, \forall k \in [K]$ and $\{\mathbf{T}(\mathbf{x}) : \mathbf{x} \in \mathcal{X}_0\} = \mathcal{X}_1$.

Then, for the given $\mathbf{T}$, we can rewrite the clustering cost as

$$C(\boldsymbol{\mu}, \mathcal{A}_0, \mathcal{A}_1) = \mathbb{E} \sum_{k=1}^{K} \mathcal{A}_S(\mathbf{X})_k \|\mathbf{X} - \mu_k\|^2 = \frac{1}{2} \mathbb{E}_0 \sum_{k=1}^{K} \mathcal{A}_0(\mathbf{X})_k \left( \|\mathbf{X} - \mu_k\|^2 + \|\mathbf{T}(\mathbf{X}) - \mu_k\|^2 \right). \tag{10}$$

For any given $\mathbf{x}$ and $\mu_k$, we decompose $\|\mathbf{x} - \mu_k\|^2$ as:

$$\|\mathbf{x} - \mu_k\|^2 = \left\| \mathbf{x} - \frac{\mathbf{x} + \mathbf{T}(\mathbf{x})}{2} + \frac{\mathbf{x} + \mathbf{T}(\mathbf{x})}{2} - \mu_k \right\|^2$$

$$= \frac{\|\mathbf{x} - \mathbf{T}(\mathbf{x})\|^2}{4} + \left\| \frac{\mathbf{x} + \mathbf{T}(\mathbf{x})}{2} - \mu_k \right\|^2 + 2 \left\langle \mathbf{x} - \frac{\mathbf{x} + \mathbf{T}(\mathbf{x})}{2}, \frac{\mathbf{x} + \mathbf{T}(\mathbf{x})}{2} - \mu_k \right\rangle.$$

We similarly decompose $\|\mathbf{T}(\mathbf{x}) - \mu_k\|^2$ as:

$$\|\mathbf{T}(\mathbf{x}) - \mu_k\|^2 = \left\| \mathbf{T}(\mathbf{x}) - \frac{\mathbf{x} + \mathbf{T}(\mathbf{x})}{2} + \frac{\mathbf{x} + \mathbf{T}(\mathbf{x})}{2} - \mu_k \right\|^2$$

$$= \frac{\|\mathbf{x} - \mathbf{T}(\mathbf{x})\|^2}{4} + \left\| \frac{\mathbf{x} + \mathbf{T}(\mathbf{x})}{2} - \mu_k \right\|^2 + 2 \left\langle \mathbf{T}(\mathbf{x}) - \frac{\mathbf{x} + \mathbf{T}(\mathbf{x})}{2}, \frac{\mathbf{x} + \mathbf{T}(\mathbf{x})}{2} - \mu_k \right\rangle.$$

Adding the two terms, we have

$$2 \left( \frac{\|\mathbf{x} - \mathbf{T}(\mathbf{x})\|^2}{4} + \left\| \frac{\mathbf{x} + \mathbf{T}(\mathbf{x})}{2} - \mu_k \right\|^2 \right).$$

Finally, we conclude that

$$\frac{1}{2} \mathbb{E}_0 \sum_{k=1}^{K} \mathcal{A}_0(\mathbf{X})_k \left( \|\mathbf{X} - \mu_k\|^2 + \|\mathbf{T}(\mathbf{X}) - \mu_k\|^2 \right)$$

$$= \frac{1}{2} \mathbb{E}_0 \sum_{k=1}^{K} \mathcal{A}_0(\mathbf{X})_k \cdot 2 \left( \frac{\|\mathbf{X} - \mathbf{T}(\mathbf{X})\|^2}{4} + \left\| \frac{\mathbf{X} + \mathbf{T}(\mathbf{X})}{2} - \mu_k \right\|^2 \right). \tag{11}$$

$\square$

## B.2. Proof of Theorem 3.3

**Theorem 3.3.** *Let $\boldsymbol{\mu}^*$ and $\mathbb{Q}^*$ be the cluster centers and joint distribution minimizing*

$$\min_{\boldsymbol{\mu}, \mathbb{Q} \in \mathcal{Q}} \mathbb{E}_{(\mathbf{X}_0, \mathbf{X}_1) \sim \mathbb{Q}} \left( 2\pi_0 \pi_1 \|\mathbf{X}_0 - \mathbf{X}_1\|^2 + \min_k \|\mathbf{T}^{\mathrm{A}}(\mathbf{X}_0, \mathbf{X}_1) - \mu_k\|^2 \right). \tag{12}$$

*Then, $(\boldsymbol{\mu}^*, \mathcal{A}_0^*, \mathcal{A}_1^*)$ is the solution of the perfectly fair $K$-means clustering, where $\mathcal{A}_0^*(\mathbf{x})_k := \mathbb{Q}^* \left( \arg\min_{k'} \|\mathbf{T}^{\mathrm{A}}(\mathbf{x}, \mathbf{X}_1) - \mu_{k'}\|^2 = k | \mathbf{X}_0 = \mathbf{x} \right)$ and $\mathcal{A}_1^*(\mathbf{x})_k$ is defined similarly.*

*Proof of Theorem 3.3.* Let $\mathbf{X}_s = \mathbf{X}|S = s, s \in \{0, 1\}$. For given $(\boldsymbol{\mu}, \mathcal{A}_0, \mathcal{A}_1)$, recall the original $K$-means objective:
$C(\boldsymbol{\mu}, \mathcal{A}_0, \mathcal{A}_1) := \mathbb{E}\sum_{k=1}^{K} \mathcal{A}_S(\mathbf{X})_k\|\mathbf{X}-\mu_k\|^2 = \pi_0\mathbb{E}_0\sum_{k=1}^{K}\mathcal{A}_0(\mathbf{X}_0)_k\|\mathbf{X}_0-\mu_k\|^2 + \pi_1\mathbb{E}_1\sum_{k=1}^{K}\mathcal{A}_1(\mathbf{X}_1)_k\|\mathbf{X}_1-\mu_k\|^2.$

Consider a set of joint distributions of $\mathbf{X}_0, \mathbf{X}_1$ given $\boldsymbol{\mu}$ as $\mathcal{Q} := \{\mathbb{Q}(\{\mathbf{x}_0, \mathbf{x}_1\}|\boldsymbol{\mu}) : \mathbf{x}_0 \in \mathcal{X}_0, \mathbf{x}_1 \in \mathcal{X}_1\}$. We show that there exists a $\mathbb{Q} = \mathbb{Q}(\{\mathbf{x}_0, \mathbf{x}_1\}|\boldsymbol{\mu}) \in \mathcal{Q}$ satisfying

$$C(\boldsymbol{\mu}, \mathcal{A}_0, \mathcal{A}_1) = \pi_0\mathbb{E}_0 \sum_{k=1}^{K} \mathcal{A}_0(\mathbf{X}_0)_k\|\mathbf{X}_0 - \mu_k\|^2 + \pi_1\mathbb{E}_1 \sum_{k=1}^{K} \mathcal{A}_1(\mathbf{X}_1)_k\|\mathbf{X}_1 - \mu_k\|^2$$
$$= \mathbb{E}_{(\mathbf{X}_0, \mathbf{X}_1) \sim \mathbb{Q}}\left(\pi_0\|\mathbf{X}_0 - \mu_k\|^2 + \pi_1\|\mathbf{X}_1 - \mu_k\|^2\right). \tag{13}$$

Let

$$\mathbb{Q}(\{\mathbf{x}_0, \mathbf{x}_1\}|\boldsymbol{\mu}) = \sum_{k=1}^{K} \frac{\mathcal{A}_0(\mathbf{x}_0)_k\mathcal{A}_1(\mathbf{x}_1)_k}{\mathcal{C}_k}\mathbb{P}_0(\{\mathbf{x}_0\})\mathbb{P}_1(\{\mathbf{x}_1\}), \tag{14}$$

where $\mathcal{C}_k := \mathbb{E}\mathcal{A}_S(\mathbf{X})_k = \mathbb{E}_0\mathcal{A}_0(\mathbf{X}_0)_k = \mathbb{E}_1\mathcal{A}_1(\mathbf{X}_1)_k$. Then,

$$\mathbb{E}_{(\mathbf{X}_0, \mathbf{X}_1) \sim \mathbb{Q}}\left(\pi_0\|\mathbf{X}_0 - \mu_k\|^2 + \pi_1\|\mathbf{X}_1 - \mu_k\|^2\right)$$
$$= \sum_{k=1}^{K}\left(\sum_{\mathbf{x}_0}\mathcal{A}_0(\mathbf{x}_0)_k\pi_0\|\mathbf{x}_0 - \mu_k\|^2\mathbb{P}_0(\{\mathbf{x}_0\}) + \sum_{\mathbf{x}_1}\mathcal{A}_1(\mathbf{x}_1)_k\pi_1\|\mathbf{x}_1 - \mu_k\|^2\mathbb{P}_1(\{\mathbf{x}_1\})\right) \tag{15}$$
$$= \sum_{k=1}^{K}\left(\mathbb{E}_0\pi_0\mathcal{A}_0(\mathbf{X}_0)_k\|\mathbf{X}_0 - \mu_k\|^2 + \mathbb{E}_1\pi_1\mathcal{A}_1(\mathbf{X}_1)_k\|\mathbf{X}_1 - \mu_k\|^2\right) = C(\boldsymbol{\mu}, \mathcal{A}_0, \mathcal{A}_1),$$

which concludes eq. (13). Our original aim is to find $(\boldsymbol{\mu}, \mathcal{A}_0, \mathcal{A}_1)$ minimizing $C(\boldsymbol{\mu}, \mathcal{A}_0, \mathcal{A}_1)$. Let $\mu(\mathbf{x}) = \mu_{k^*}$, where $k^* = \arg\min_k \|\mathbf{x} - \mu_k\|^2$. Let $\tilde{\mu}(\mathbf{x}_0, \mathbf{x}_1) := \mu_{k'}$, where $k' = \arg\min_k \|\pi_0\mathbf{x}_0 + \pi_1\mathbf{x}_1 - \mu_k\|^2$. For given $\mathbb{Q}$ defined in eq. (14), similar to Theorem 3.1, we can reformulate

$$\mathbb{E}_{(\mathbf{X}_0, \mathbf{X}_1) \sim \mathbb{Q}}\left(\pi_0\|\mathbf{X}_0 - \mu(\mathbf{X}_0)\|^2 + \pi_1\|\mathbf{X}_1 - \mu(\mathbf{X}_1)\|^2\right)$$
$$= \mathbb{E}_{(\mathbf{X}_0, \mathbf{X}_1) \sim \mathbb{Q}}\left(2\pi_0\pi_1\|\mathbf{X}_0 - \mathbf{X}_1\|^2 + \|\pi_0\mathbf{X}_0 + \pi_1\mathbf{X}_1 - \tilde{\mu}(\mathbf{X}_0, \mathbf{X}_1)\|^2\right). \tag{16}$$

In turn, the assignment functions are given as

$$\mathcal{A}_0(\mathbf{x}_0)_k := \mathbb{Q}\left(\arg\min_{k'}\|\pi_0\mathbf{x}_0 + \pi_1\mathbf{X}_1 - \mu_{k'}\|^2 = k\Big|\mathbf{X}_0 = \mathbf{x}_0\right) \tag{17}$$

and

$$\mathcal{A}_1(\mathbf{x}_1)_k := \mathbb{Q}\left(\arg\min_{k'}\|\pi_0\mathbf{X}_0 + \pi_1\mathbf{x}_1 - \mu_{k'}\|^2 = k\Big|\mathbf{X}_1 = \mathbf{x}_1\right). \tag{18}$$

Hence, $C(\boldsymbol{\mu}, \mathcal{A}_0, \mathcal{A}_1) = \mathbb{E}_{(\mathbf{X}_0, \mathbf{X}_1) \sim \mathbb{Q}}\left(2\pi_0\pi_1\|\mathbf{X}_0 - \mathbf{X}_1\|^2 + \min_k\|\pi_0\mathbf{X}_0 + \pi_1\mathbf{X}_1 - \mu_k\|^2\right).$ $\qquad\square$

### B.3. Proof of Theorem 4.1

**Theorem 4.1.** *Minimizing FCA-C objective $\tilde{C}(\mathbb{Q}, \mathcal{W}, \boldsymbol{\mu})$ with the corresponding assignment function defined in eq. (6), is equivalent to minimizing $C(\boldsymbol{\mu}, \mathcal{A}_0, \mathcal{A}_1)$ subject to $(\mathcal{A}_0, \mathcal{A}_1) \in \mathbf{A}_\varepsilon$.*

*Proof of Theorem 4.1.* It suffices to show the followings: A and B.

A. For given $\mathbb{Q}, \mathcal{W}, \boldsymbol{\mu}$, let $\mathcal{A}_0$ and $\mathcal{A}_1$ be the constructed assignment functions, defined in eq. (6). Then, we have $\tilde{C}(\mathbb{Q}, \mathcal{W}, \boldsymbol{\mu}) = C(\boldsymbol{\mu}, \mathcal{A}_0, \mathcal{A}_1)$ and $(\mathcal{A}_0, \mathcal{A}_1) \in \mathbf{A}_\varepsilon$.

B. (i) For given $\boldsymbol{\mu}$ and $(\mathcal{A}_0, \mathcal{A}_1) \in \mathbf{A}_\varepsilon$, there exist $\mathbb{Q}$ and $\mathcal{W}$ s.t. $\tilde{C}(\mathbb{Q}, \mathcal{W}, \boldsymbol{\mu}) \leq C(\boldsymbol{\mu}, \mathcal{A}_0, \mathcal{A}_1)$.

• *Proof of A.*

For given $\mathbb{Q}, \mathcal{W}, \boldsymbol{\mu}$, we construct the assignment functions as in eq. (6). In other words, we define

$$\mathcal{A}_0(\mathbf{x}_0)_k = \mathbb{P}_1(\{\arg\min_{k'} \|\mathbf{T}^A(\mathbf{x}_0, \mathbf{X}_1) - \mu_{k'}\|^2 = k, (\mathbf{x}_0, \mathbf{X}_1) \in \mathcal{W}^c\})$$
$$+ \mathbb{1}\left(\arg\min_{k'} \|\mathbf{x}_0 - \mu_{k'}\|^2 = k\right) \cdot \mathbb{P}_1(\{(\mathbf{x}_0, \mathbf{X}_1) \in \mathcal{W}\}),$$

and the assignment function $\mathcal{A}_1$ is defined similarly. Then, we have $\tilde{C}(\mathbb{Q}, \mathcal{W}, \boldsymbol{\mu}) = C(\boldsymbol{\mu}, \mathcal{A}_0, \mathcal{A}_1)$ by its definition.

Furthermore,

$$\sum_{k=1}^{K} |\mathbb{E}_0 \mathcal{A}_0(\mathbf{X})_k - \mathbb{E}_1 \mathcal{A}_1(\mathbf{X})_k|$$

$$= \sum_{k=1}^{K} \left| \mathbb{E}_0 \mathbb{Q}_1 \left( \arg\min_{k'} \|\mathbf{T}^A(\mathbf{X}_0, \mathbf{X}_1) - \mu_{k'}\|^2 = k, (\mathbf{X}_0, \mathbf{X}_1) \in \mathcal{W}^c \right) \right.$$

$$- \mathbb{E}_1 \mathbb{Q}_0 \left( \arg\min_{k'} \|\mathbf{T}^A(\mathbf{X}_0, \mathbf{X}_1) - \mu_{k'}\|^2 = k, (\mathbf{X}_0, \mathbf{X}_1) \in \mathcal{W}^c \right)$$

$$+ \mathbb{E}_0 \mathbb{Q}_1 ((\mathbf{X}_0, \mathbf{X}_1) \in \mathcal{W}) \mathbb{1}(\arg\min_{k'} \|\mathbf{X}_0 - \mu_{k'}\|^2 = k)$$

$$\left. - \mathbb{E}_1 \mathbb{Q}_0 ((\mathbf{X}_0, \mathbf{X}_1) \in \mathcal{W}) \mathbb{1}(\arg\min_{k'} \|\mathbf{X}_1 - \mu_{k'}\|^2 = k) \right|$$

$$= \sum_{k=1}^{K} \left| \mathbb{E}_0 \mathbb{Q}_1 ((\mathbf{X}_0, \mathbf{X}_1) \in \mathcal{W}) \mathbb{1}(\arg\min_{k'} \|\mathbf{X}_0 - \mu_{k'}\|^2 = k) \right. \tag{19}$$

$$\left. - \mathbb{E}_1 \mathbb{Q}_0 ((\mathbf{X}_0, \mathbf{X}_1) \in \mathcal{W}) \mathbb{1}(\arg\min_{k'} \|\mathbf{X}_1 - \mu_{k'}\|^2 = k) \right|$$

$$= \sum_{k=1}^{K} \left| \mathbb{E}_{\mathbf{X}_0, \mathbf{X}_1} \mathbb{1}((\mathbf{X}_0, \mathbf{X}_1) \in \mathcal{W}) \cdot \left( \mathbb{1}(\arg\min_{k'} \|\mathbf{X}_0 - \mu_{k'}\|^2 = k) - \mathbb{1}(\arg\min_{k'} \|\mathbf{X}_1 - \mu_{k'}\|^2 = k) \right) \right|$$

$$\leq \sum_{k=1}^{K} \mathbb{E}_{\mathbf{X}_0, \mathbf{X}_1} \left( \left| \mathbb{1}(\arg\min_{k'} \|\mathbf{X}_0 - \mu_{k'}\|^2 = k) - \mathbb{1}(\arg\min_{k'} \|\mathbf{X}_1 - \mu_{k'}\|^2 = k) \right| |(\mathbf{X}_0, \mathbf{X}_1) \in \mathcal{W} \right)$$

$$\cdot \mathbb{P}_{\mathbf{X}_0, \mathbf{X}_1} \left| \mathbb{1}((\mathbf{X}_0, \mathbf{X}_1) \in \mathcal{W}) \right|$$

$$\leq \varepsilon,$$

which implies $(\mathcal{A}_0, \mathcal{A}_1) \in \mathbf{A}_\varepsilon$.

• *Proof of B.*

For each $k \in [K]$, let $\delta_k = \min\{\mathbb{E}_0(\mathcal{A}_0(\mathbf{X}))_k, \mathbb{E}_1(\mathcal{A}_1(\mathbf{X}))_k\}$. We decompose $\mathcal{A}_s(\cdot)_k = \tilde{\mathcal{A}}_s(\cdot)_k + \mathcal{A}_s^c(\cdot)_k$, where $\tilde{\mathcal{A}}_s(\cdot)_k = \delta_k \mathcal{A}_s(\cdot)_k / \mathbb{E}_s(\mathcal{A}_s(\mathbf{X}))_k$. Then, $\mathbb{E}_0 \tilde{\mathcal{A}}_0(\mathbf{X})_k = \mathbb{E}_1 \tilde{\mathcal{A}}_1(\mathbf{X})_k$ for all $k \in [K]$. Define $\tilde{\mathcal{A}}_s(\mathbf{X})_{K+1} := \sum_{k=1}^{K} \mathcal{A}_s^c(\mathbf{X})_k$. Note that $\mathbb{E}_s \tilde{\mathcal{A}}_s(\mathbf{X})_{K+1} \leq \varepsilon$.

Now, for given $\boldsymbol{\mu}$, we define a probability measure on $\mathcal{X}_0 \times \mathcal{X}_1$ by

$$\mathbb{Q}(d\mathbf{x}_0, d\mathbf{x}_1|\boldsymbol{\mu}) = \sum_{k=1}^{K} \frac{\tilde{\mathcal{A}}_0(\mathbf{x}_0)_k \tilde{\mathcal{A}}_1(\mathbf{x}_1)_k}{\delta_k} p_0(\mathbf{x}_0) p_1(\mathbf{x}_1) + \frac{\tilde{\mathcal{A}}_0(\mathbf{x}_0)_{K+1} \tilde{\mathcal{A}}_1(\mathbf{x}_1)_{K+1}}{\delta_{K+1}} p_0(\mathbf{x}_0) p_1(\mathbf{x}_1),$$

where $\delta_{K+1} = 1 - \sum_{k=1}^{K} \delta_k$ and $p_0, p_1$ are the densities of $\mathbb{P}_0, \mathbb{P}_1$, respectively.

Note that $\delta_{K+1} = \mathbb{E}_s \tilde{\mathcal{A}}_s(\mathbf{X})_{K+1}$ and thus $\delta_{K+1} \leq \varepsilon$. In addition, for given $(\mathbf{x}_0, \mathbf{x}_1) \in \mathcal{X}_0 \times \mathcal{X}_1$, we define a binary random variable $R(\mathbf{x}_0, \mathbf{x}_1)$ such that

$$\Pr(R(\mathbf{x}_0, \mathbf{x}_1) = 1) = \frac{\frac{\tilde{\mathcal{A}}_0(\mathbf{x}_0)_{K+1} \tilde{\mathcal{A}}_1(\mathbf{x}_1)_{K+1}}{\delta_{K+1}} p_0(\mathbf{x}_0) p_1(\mathbf{x}_1)}{\mathbb{Q}(d\mathbf{x}_0, d\mathbf{x}_1|\boldsymbol{\mu})}.$$

For given $\mathbf{x} \in \mathcal{X}_s$, let $\mu(\mathbf{x}) = \mu_{k^*}$, where $k^* = \operatorname{argmin}_k \|\mathbf{x} - \mu_k\|^2$. For given $(\mathbf{x}_0, \mathbf{x}_1) \in \mathcal{X}_0 \times \mathcal{X}_1$, let $\tilde{\mu}(\mathbf{x}_0, \mathbf{x}_1) = \mu_{k'}$, where $k' = \arg\min_k \|\pi_0 \mathbf{x}_0 + \pi_1 \mathbf{x}_1 - \mu_k\|^2$. Then, it holds that $\tilde{C}(\mathbb{Q}, R, \boldsymbol{\mu}) \leq C(\boldsymbol{\mu}, \mathcal{A}_0, \mathcal{A}_1)$, where

$$\tilde{C}(\mathbb{Q}, R, \boldsymbol{\mu}) := \mathbb{E}_{\mathbb{Q},R}\left[\left\{2\pi_0\pi_1\|\mathbf{X}_0 - \mathbf{X}_1\|^2 + \|\pi_0\mathbf{X}_0 + \pi_1\mathbf{X}_1 - \tilde{\mu}(\mathbf{X}_0, \mathbf{X}_1)\|^2\right\}(1 - R(\mathbf{X}_0, \mathbf{X}_1))\right]$$
$$+ \mathbb{E}_{\mathbb{Q},R}\left[\left\{\pi_0\|\mathbf{X}_0 - \mu(\mathbf{X}_0)\|^2 + \pi_1\|\mathbf{X}_1 - \mu(\mathbf{X}_1)\|^2\right\}R(\mathbf{X}_0, \mathbf{X}_1)\right].$$

The final mission is to find $\mathcal{W} \subset \mathcal{X}_0 \times \mathcal{X}_1$ such that $R(\mathbf{x}_0, \mathbf{x}_1) = \mathbb{1}((\mathbf{x}_0, \mathbf{x}_1) \in \mathcal{W})$. For this, define $\eta(\mathbf{x}_0, \mathbf{x}_1) = 2\pi_0\pi_1\|\mathbf{x}_0 - \mathbf{x}_1\|^2 + \min_k \|\mathbf{T}^A(\mathbf{x}_0, \mathbf{x}_1) - \mu_k\|^2$. Let $\eta_\varepsilon$ be the $\varepsilon$th upper quantile of $\eta(\mathbf{X}_0, \mathbf{X}_1)$ and let

$$\tilde{\mathcal{W}} = \{(\mathbf{x}_0, \mathbf{x}_1) : \eta(\mathbf{x}_0, \mathbf{x}_1) > \eta_\varepsilon\}. \tag{20}$$

Then, we can find $\mathcal{W} \supset \tilde{\mathcal{W}}$ such that $\tilde{C}(\mathbb{Q}, \mathcal{W}, \boldsymbol{\mu}) \leq \tilde{C}(\mathbb{Q}, R, \boldsymbol{\mu})$ and $\mathbb{Q}(\mathcal{W}) = \varepsilon$ (see Remark B.1 below), which completes the proof.

$\square$

*Remark* B.1 (The case when the densities do not exist). When the distribution of $\eta(\mathbf{X}_0, \mathbf{X}_1)$ is strictly increasing, we have $\mathbb{Q}(\mathcal{W}) = \varepsilon$. If not, we can find $\mathcal{W}$ such that $\mathcal{W} \supset \{(\mathbf{x}_0, \mathbf{x}_1) : \eta(\mathbf{x}_0, \mathbf{x}_1) \geq \varepsilon\}$ with $\mathbb{Q}(\mathcal{W}) = \varepsilon$ when $\mathbb{Q}$ has its density.

When $\mathbb{Q}$ is discrete, the situation is tricky. When $n_0 = n_1$, the measure $\mathbb{Q}$ minimizing $\tilde{C}(\mathbb{Q}, \mathcal{W}, \boldsymbol{\mu})$ has masses $1/n_0$ on $n_0$ many pairs of $(\mathbf{x}_0, \mathbf{x}_1)$ among $\mathcal{X}_0 \times \mathcal{X}_1$. In this case, whenever $n_0\varepsilon$ is an integer, we can find $\mathcal{W}$ such that $\mathbb{Q}(\mathcal{W}) = \varepsilon$.

Otherwise, we could consider a random assignment. Let $F_\eta$ be the distribution of $\eta(\mathbf{X}_0, \mathbf{X}_1)$. Suppose that $F_\eta$ has a jump at $\eta_\varepsilon$. In that case, $\mathbb{Q}(\mathcal{W}) < \varepsilon$. Let $(\mathbf{x}_0^*, \mathbf{x}_1^*)$ be the element such that $\eta(\mathbf{x}_0^*, \mathbf{x}_1^*) = \eta_\varepsilon$. Then, we can let $R(\mathbf{x}_0^*, \mathbf{x}_1^*) = 1$ with probability $(\varepsilon - \mathbb{Q}(\mathcal{W}))/\mathbb{Q}(\{\mathbf{x}_0^*, \mathbf{x}_1^*\})$, $R(\mathbf{x}_0, \mathbf{x}_1) = 1$ for $(\mathbf{x}_0, \mathbf{x}_1) \in \mathcal{W}$ and $R(\mathbf{x}_0, \mathbf{x}_1) = 0$ for $(\mathbf{x}_0, \mathbf{x}_1) \in (\mathcal{W} \cup \{\mathbf{x}_0^*, \mathbf{x}_1^*\})^c$. The current FCA-C algorithm can be modified easily for this random assignment.

### B.4. Proof of Proposition 4.2

**Proposition 4.2.** *For any assignment function* $(\mathcal{A}_0, \mathcal{A}_1) \in \mathbf{A}_\varepsilon$, *we have*

$$\max_{k \in [K]} \left| \frac{\sum_{\mathbf{x}_i \in \mathcal{X}_0} \mathcal{A}_0(\mathbf{x}_i)_k}{\sum_{\mathbf{x}_i \in \mathcal{X}_1} \mathcal{A}_1(\mathbf{x}_i)_k} - \frac{n_0}{n_1} \right| \leq c\varepsilon \tag{21}$$

*where* $c = \frac{n_0}{n_1} \max_{k \in [K]} \frac{1}{\mathbb{E}_1 \mathcal{A}_1(\mathbf{X})_k}$.

*Proof of Proposition 4.2.* On the other hand, by the definition of $\mathbf{A}_\varepsilon$, any $(\mathcal{A}_0, \mathcal{A}_1) \in \mathbf{A}_\varepsilon$ satisfies

$$\sum_{k=1}^{K} \left| \frac{1}{n_0} \sum_{\mathbf{x}_i \in \mathcal{X}_0} \mathcal{A}_0(\mathbf{x}_i)_k - \frac{1}{n_1} \sum_{\mathbf{x}_j \in \mathcal{X}_1} \mathcal{A}_1(\mathbf{x}_j)_k \right| \leq \varepsilon,$$

which implies

$$\left| \frac{1}{n_0} \sum_{\mathbf{x}_i \in \mathcal{X}_0} \mathcal{A}_0(\mathbf{x}_i)_k - \frac{1}{n_1} \sum_{\mathbf{x}_j \in \mathcal{X}_1} \mathcal{A}_1(\mathbf{x}_j)_k \right| = |\mathbb{E}_0 \mathcal{A}_0(\mathbf{X})_k - \mathbb{E}_1 \mathcal{A}_1(\mathbf{X})_k| \leq \varepsilon$$

for all $k \in [K]$. Then, we have

$$\left| \frac{\sum_{\mathbf{x}_i \in \mathcal{X}_0} \mathcal{A}_0(\mathbf{x}_i)_k}{\sum_{\mathbf{x}_i \in \mathcal{X}_1} \mathcal{A}_1(\mathbf{x}_i)_k} - \frac{n_0}{n_1} \right| = \frac{n_0}{n_1} \left| \frac{\sum_{\mathbf{x}_i \in \mathcal{X}_0} \mathcal{A}_0(\mathbf{x}_i)_k / n_0}{\sum_{\mathbf{x}_i \in \mathcal{X}_1} \mathcal{A}_1(\mathbf{x}_i)_k / n_1} - 1 \right| = \frac{n_0}{n_1} \left| \frac{\mathbb{E}_0 \mathcal{A}_0(\mathbf{X})_k}{\mathbb{E}_1 \mathcal{A}_1(\mathbf{X})_k} - 1 \right|$$

$$= \frac{n_0}{n_1} \frac{1}{\mathbb{E}_1 \mathcal{A}_1(\mathbf{X})_k} |\mathbb{E}_0 \mathcal{A}_0(\mathbf{X})_k - \mathbb{E}_1 \mathcal{A}_1(\mathbf{X})_k| \leq \frac{n_0}{n_1} \frac{1}{\mathbb{E}_1 \mathcal{A}_1(\mathbf{X})_k} \varepsilon, \tag{22}$$

for all $k \in [K]$. Letting $c := \frac{n_0}{n_1} \max_{k \in [K]} \frac{1}{\mathbb{E}_1 \mathcal{A}_1(\mathbf{X})_k}$ concludes the proof.

$\square$

## B.5. Proof of Theorem 4.3

We establish the approximation guarantee of our proposed algorithm: The cost of the solution obtained by our FCA-C algorithm in Section 4.2 is at most $\tau + 2$ times of the cost of the global optimal fair clustering solution (where $\tau$ is the approximation error of the algorithm used for finding initial cluster centers without the fairness constraint), with an additional violation of $\mathcal{O}(\varepsilon)$. In other words, for a given fairness level $\varepsilon$, FCA-C has an approximation error of $\tau + 2$ with an additional violation of $\mathcal{O}(\varepsilon)$.

First, for a given fairness level $\varepsilon$, we define several notations to be used as follows.

- $\boldsymbol{\mu}^{\triangle}, \mathcal{A}_0^{\triangle}, \mathcal{A}_1^{\triangle}$: the solution of FCA-C algorithm for fairness level $\varepsilon$, given initial cluster centers obtained by a $\tau$-approximation algorithm for $K$-means clustering.

- $\tilde{\boldsymbol{\mu}}, \tilde{\mathcal{A}}_0, \tilde{\mathcal{A}}_1$: the optimal fair clustering solution of $\min_{\boldsymbol{\mu}, \mathcal{A}_0, \mathcal{A}_1} C(\boldsymbol{\mu}, \mathcal{A}_0, \mathcal{A}_1)$ subject to $(\mathcal{A}_0, \mathcal{A}_1) \in \mathbf{A}_\varepsilon$, for a given fairness level $\varepsilon$.

- $\boldsymbol{\mu}^{\diamond}, \mathcal{A}_0^{\diamond}, \mathcal{A}_1^{\diamond}$: the (approximated) clustering solution, given a $\tau$-approximation algorithm for (standard) $K$-means clustering.

Then, we can prove Theorem B.2 below, which is a formal version of Theorem 4.3, showing the approximation error of FCA-C algorithm.

**Theorem B.2** (A formal version of Theorem 4.3). *Suppose that $\sup_{\mathbf{x} \in \mathcal{X}} \|\mathbf{x}\|^2 \leq R$ for some $R > 0$. Given initial cluster centers obtained by a $\tau$-approximation algorithm, FCA-C returns a $(\tau + 2)$-approximate solution with a violation $3R\varepsilon$ for optimal fair clustering, i.e., $C(\boldsymbol{\mu}^{\triangle}, \mathcal{A}_0^{\triangle}, \mathcal{A}_1^{\triangle}) \leq (\tau + 2)C(\tilde{\boldsymbol{\mu}}, \tilde{\mathcal{A}}_0, \tilde{\mathcal{A}}_1) + 3R\varepsilon$.*

*Proof of Theorem B.2.* Let $\mathbb{Q}', \mathcal{W}' = \arg\min_{\mathbb{Q}, \mathcal{W}} \tilde{C}(\mathbb{Q}, \mathcal{W}, \boldsymbol{\mu}^{\diamond})$, and $\mathcal{A}_0', \mathcal{A}_1'$ are the assignment functions corresponding to $\mathbb{Q}'$ and $\mathcal{W}'$.

It suffices to show the following two claims:

1. (*Claim A*): Given initial cluster centers obtained by a $\tau$-approximation algorithm for $K$-means clustering, we have $C(\boldsymbol{\mu}^{\diamond}, \mathcal{A}_0', \mathcal{A}_1') \leq (\tau + 2)C(\tilde{\boldsymbol{\mu}}, \tilde{\mathcal{A}}_0, \tilde{\mathcal{A}}_1)$.

2. (*Claim B*): There exist $\mathbb{Q}^{\star}, \mathcal{W}^{\star}$ such that $C(\boldsymbol{\mu}^{\diamond}, \mathcal{A}_0^{\star}, \mathcal{A}_1^{\star}) \leq C(\boldsymbol{\mu}^{\diamond}, \mathcal{A}_0', \mathcal{A}_1') + 3R\varepsilon$, where $\mathcal{A}_0^{\star}$ and $\mathcal{A}_1^{\star}$ are the fair assignment functions corresponding to $\mathbb{Q}^{\star}$ and $\mathcal{W}^{\star}$.

Then, by combining Claim A and B, we have that $C(\boldsymbol{\mu}^{\diamond}, \mathcal{A}_0^{\star}, \mathcal{A}_1^{\star}) \leq (\tau + 2)C(\tilde{\boldsymbol{\mu}}, \tilde{\mathcal{A}}_0, \tilde{\mathcal{A}}_1) + 3R\varepsilon$. Note that $\mathcal{A}_0^{\star}$ and $\mathcal{A}_1^{\star}$ are the feasible solution of FCA-C, while $\mathcal{A}_0'$ and $\mathcal{A}_1'$ are the global optimal fair assignment function given $\boldsymbol{\mu}^{\diamond}$. Consequently, we can clearly get the same approximation error for $(\boldsymbol{\mu}^{\triangle}, \mathcal{A}_0^{\triangle}, \mathcal{A}_1^{\triangle})$, by iterating the process (finding new cluster centers minimizing the cost and updating the assignment functions), which completes the proof. See Remark B.3 for details.

- *Proof of Claim A*: Let $C^* = \min_{\boldsymbol{\mu}, \mathcal{A}_0, \mathcal{A}_1} C(\boldsymbol{\mu}, \mathcal{A}_0, \mathcal{A}_1)$ represent the global optimal clustering cost without fairness constraint. Then, we have $C(\boldsymbol{\mu}^{\diamond}, \mathcal{A}_0^{\diamond}, \mathcal{A}_1^{\diamond}) \leq \tau C^*$.

  We show that for given initial cluster centers $\boldsymbol{\mu}^{\diamond}$, there exist $\mathcal{A}_0^+$ and $\mathcal{A}_1^+$ that satisfy the following conditions: (i) $(\mathcal{A}_0^+, \mathcal{A}_1^+) \in \mathbf{A}_\varepsilon$, (ii) $C(\boldsymbol{\mu}^{\diamond}, \mathcal{A}_0^+, \mathcal{A}_1^+) \leq (\tau + 2)C(\tilde{\boldsymbol{\mu}}, \tilde{\mathcal{A}}_0, \tilde{\mathcal{A}}_1)$, and (iii) $C(\boldsymbol{\mu}^{\diamond}, \mathcal{A}_0', \mathcal{A}_1') \leq C(\boldsymbol{\mu}^{\diamond}, \mathcal{A}_0^+, \mathcal{A}_1^+)$, where $\mathcal{A}_0', \mathcal{A}_1'$ are the fair assignment functions corresponding to $\mathbb{Q}', \mathcal{W}' = \arg\min_{\mathbb{Q}, \mathcal{W}} \tilde{C}(\mathbb{Q}, \mathcal{W}, \boldsymbol{\mu}^{\diamond})$.

  (i) Recall that $\tilde{\boldsymbol{\mu}} = \{\tilde{\mu}_k\}_{k=1}^K$ and $\boldsymbol{\mu}^{\diamond} = \{\mu_k^{\diamond}\}_{k=1}^K$. For all $k \in [K]$, we define the set of nearest neighbors of $\mu_k^{\diamond}$ as $N(\mu_k^{\diamond}) := \{\tilde{\mu}_k \in \tilde{\boldsymbol{\mu}} : \arg\min_{\mu_{k'}^{\diamond} \in \boldsymbol{\mu}^{\diamond}} \|\tilde{\mu}_k - \mu_{k'}^{\diamond}\|^2 = \mu_k^{\diamond}\}$. For $\mathbf{x} \in \mathcal{X}$, we define $\mathcal{A}_s^+(\mathbf{x})_k := \sum_{k': \tilde{\mu}_{k'} \in N(\mu_k^{\diamond})} \tilde{\mathcal{A}}_s(\mathbf{x})_{k'}$. Then, $\mathcal{A}_s^+$ also becomes an assignment function since $\sum_{k=1}^K \mathcal{A}_s^+(\mathbf{x}) = 1$ for all $\mathbf{x} \in \mathcal{X}$.

  Since $(\tilde{\mathcal{A}}_0, \tilde{\mathcal{A}}_1) \in \mathbf{A}_\varepsilon$, we have the fact that $\sum_{k=1}^K \left| \frac{1}{n_0} \sum_{i=1}^{n_0} \tilde{\mathcal{A}}_0(\mathbf{x}_i)_k - \frac{1}{n_1} \sum_{j=1}^{n_1} \tilde{\mathcal{A}}_1(\mathbf{x}_j)_k \right| \leq \varepsilon$. Therefore, we obtain

$$\sum_{k=1}^K \left| \frac{1}{n_0} \sum_{i=1}^{n_0} \mathcal{A}_0^+(\mathbf{x}_i)_k - \frac{1}{n_1} \sum_{j=1}^{n_1} \mathcal{A}_1^+(\mathbf{x}_j)_k \right| = \sum_{k=1}^K \left| \sum_{k': \tilde{\mu}_{k'} \in N(\mu_k^{\diamond})} \left( \frac{1}{n_0} \sum_{i=1}^{n_0} \tilde{\mathcal{A}}_0(\mathbf{x}_i)_{k'} - \frac{1}{n_1} \sum_{j=1}^{n_1} \tilde{\mathcal{A}}_1(\mathbf{x}_j)_{k'} \right) \right|$$

$$\leq \sum_{k=1}^{K} \sum_{k':\tilde{\mu}_{k'}\in N(\mu_k^\diamond)} \left| \frac{1}{n_0} \sum_{i=1}^{n_0} \tilde{\mathcal{A}}_0(\mathbf{x}_i)_{k'} - \frac{1}{n_1} \sum_{j=1}^{n_1} \tilde{\mathcal{A}}_1(\mathbf{x}_j)_{k'} \right| = \sum_{k'=1}^{K} \left| \frac{1}{n_0} \sum_{i=1}^{n_0} \tilde{\mathcal{A}}_0(\mathbf{x}_i)_{k'} - \frac{1}{n_1} \sum_{j=1}^{n_1} \tilde{\mathcal{A}}_1(\mathbf{x}_j)_{k'} \right| \leq \varepsilon,$$

where the last equality holds because $\cup_{k=1}^{K} \cup_{k':\tilde{\mu}_{k'}\in N(\mu_k^\diamond)} \{k'\} = \{k'\}_{k'=1}^{K}$. Thus, the condition (i) is satisfied.

(ii) For a given $\mathbf{x}\in\mathcal{X}$, let $\mu_k^\diamond(\mathbf{x}) := \arg\min_{\mu_k^\diamond\in\mu^\diamond} \|\mathbf{x} - \mu_k^\diamond\|^2$ be the initial cluster center closest to $\mathbf{x}$. We then have:

$$
\begin{aligned}
\sum_{k=1}^{K} \mathcal{A}_s^+(\mathbf{x})_k \|\mathbf{x} - \mu_k^\diamond\|^2 &= \sum_{k=1}^{K} \sum_{k':\tilde{\mu}_{k'}\in N(\mu_k^\diamond)} \tilde{\mathcal{A}}_s(\mathbf{x})_{k'} \|\mathbf{x} - \mu_k^\diamond\|^2 \\
&\leq \sum_{k=1}^{K} \sum_{k':\tilde{\mu}_{k'}\in N(\mu_k^\diamond)} \tilde{\mathcal{A}}_s(\mathbf{x})_{k'} \left( \|\mathbf{x} - \tilde{\mu}_{k'}\|^2 + \|\tilde{\mu}_{k'} - \mu_k^\diamond\|^2 \right) \\
&\leq \sum_{k=1}^{K} \sum_{k':\tilde{\mu}_{k'}\in N(\mu_k^\diamond)} \tilde{\mathcal{A}}_s(\mathbf{x})_{k'} \left( \|\mathbf{x} - \tilde{\mu}_{k'}\|^2 + \|\tilde{\mu}_{k'} - \mu_k^\diamond(\mathbf{x})\|^2 \right) \\
&\leq \sum_{k=1}^{K} \sum_{k':\tilde{\mu}_{k'}\in N(\mu_k^\diamond)} \tilde{\mathcal{A}}_s(\mathbf{x})_{k'} \left( 2\|\mathbf{x} - \tilde{\mu}_{k'}\|^2 + \|\mathbf{x} - \mu_k^\diamond(\mathbf{x})\|^2 \right) \\
&= 2 \sum_{k'=1}^{K} \tilde{\mathcal{A}}_s(\mathbf{x})_{k'} \|\mathbf{x} - \tilde{\mu}_{k'}\|^2 + \sum_{k=1}^{K} \sum_{k':\tilde{\mu}_{k'}\in N(\mu_k^\diamond)} \tilde{\mathcal{A}}_s(\mathbf{x})_{k'} \|\mathbf{x} - \mu_k^\diamond(\mathbf{x})\|^2 \\
&= 2 \sum_{k'=1}^{K} \tilde{\mathcal{A}}_s(\mathbf{x})_{k'} \|\mathbf{x} - \tilde{\mu}_{k'}\|^2 + \|\mathbf{x} - \mu_k^\diamond(\mathbf{x})\|^2.
\end{aligned}
\tag{23}
$$

Summing over all $\mathbf{x}$ and dividing by $n$, we obtain:

$$C(\boldsymbol{\mu}^\diamond, \mathcal{A}_0^+, \mathcal{A}_1^+) \leq 2C(\tilde{\boldsymbol{\mu}}, \tilde{\mathcal{A}}_0, \tilde{\mathcal{A}}_1) + \frac{1}{n} \sum_{\mathbf{x}\in\mathcal{X}} \min_k \|\mathbf{x} - \mu_k^\diamond\|^2 = 2C(\tilde{\boldsymbol{\mu}}, \tilde{\mathcal{A}}_0, \tilde{\mathcal{A}}_1) + C(\boldsymbol{\mu}^\diamond, \mathcal{A}_0^\diamond, \mathcal{A}_1^\diamond)$$

$$\leq 2C(\tilde{\boldsymbol{\mu}}, \tilde{\mathcal{A}}_0, \tilde{\mathcal{A}}_1) + \tau C^* \leq 2C(\tilde{\boldsymbol{\mu}}, \tilde{\mathcal{A}}_0, \tilde{\mathcal{A}}_1) + \tau C(\tilde{\boldsymbol{\mu}}, \tilde{\mathcal{A}}_0, \tilde{\mathcal{A}}_1) = (\tau + 2)C(\tilde{\boldsymbol{\mu}}, \tilde{\mathcal{A}}_0, \tilde{\mathcal{A}}_1),$$

which concludes (ii).

(iii) It is clear that $C(\boldsymbol{\mu}^\diamond, \mathcal{A}_0', \mathcal{A}_1') \leq C(\boldsymbol{\mu}^\diamond, \mathcal{A}_0^+, \mathcal{A}_1^+)$, since $\mathcal{A}_0'$ and $\mathcal{A}_1'$ are the minimizers of $C(\boldsymbol{\mu}, \mathcal{A}_0, \mathcal{A}_1)$ subject to $(\mathcal{A}_0, \mathcal{A}_1) \in \mathbf{A}_\varepsilon$ given $\boldsymbol{\mu} = \boldsymbol{\mu}^\diamond$ (by Theorem 4.1). Thus, the condition (iii) is satisfied.

Let $\mu_k' := \sum_{i=1}^{n} \mathcal{A}_s'(\mathbf{x}_i)_k \mathbf{x}_i / \sum_{i=1}^{n} \mathcal{A}_s'(\mathbf{x}_i)_k$, which is the minimizer of $\min_{\boldsymbol{\mu}} C(\boldsymbol{\mu}, \mathcal{A}_0', \mathcal{A}_1')$ given $\mathcal{A}_0'$ and $\mathcal{A}_1'$. Then, it is clear that $C(\boldsymbol{\mu}', \mathcal{A}_0', \mathcal{A}_1') \leq C(\boldsymbol{\mu}^\diamond, \mathcal{A}_0', \mathcal{A}_1') \leq (\tau + 2)C(\tilde{\boldsymbol{\mu}}, \tilde{\mathcal{A}}_0, \tilde{\mathcal{A}}_1)$.

- *Proof of Claim B*: Note that we can rewrite

$$
\begin{aligned}
\tilde{C}(\mathbb{Q}, \mathcal{W}, \boldsymbol{\mu}) = {}& \mathbb{E}_{\mathbf{X}_0, \mathbf{X}_1 \sim \mathbb{Q}} \big( \text{FCA cost}(\mathbf{X}_0, \mathbf{X}_1, \boldsymbol{\mu}) \big) \\
& - \mathbb{E}_{\mathbf{X}_0, \mathbf{X}_1 \sim \mathbb{Q}} \big( \text{FCA cost}(\mathbf{X}_0, \mathbf{X}_1, \boldsymbol{\mu}) - \text{K-means cost}(\mathbf{X}_0, \mathbf{X}_1, \boldsymbol{\mu}) \big) \mathbb{1} \left( (\mathbf{X}_0, \mathbf{X}_1) \in \mathcal{W} \right),
\end{aligned}
\tag{24}
$$

where $\text{FCA cost}(\mathbf{X}_0, \mathbf{X}_1, \boldsymbol{\mu}) = \big( 2\pi_0\pi_1 \|\mathbf{X}_0 - \mathbf{X}_1\|^2 + \min_k \|\mathbf{T}^A(\mathbf{X}_0, \mathbf{X}_1) - \mu_k\|^2 \big)$ and $\text{K-means cost}(\mathbf{X}_0, \mathbf{X}_1, \boldsymbol{\mu}) = \min_k \big( \pi_0 \|\mathbf{X}_0 - \mu_k\|^2 \big) + \min_k \big( \pi_1 \|\mathbf{X}_1 - \mu_k\|^2 \big)$, which are defined in eq. (5).

(i): For given $\boldsymbol{\mu}^\diamond$, define $\mathbb{Q}^*$ as the solution of $\min_{\mathbb{Q}\in\mathcal{Q}} \mathbb{E}_{\mathbf{X}_0, \mathbf{X}_1 \sim \mathbb{Q}} (\text{FCA cost}(\mathbf{X}_0, \mathbf{X}_1, \boldsymbol{\mu}^\diamond))$, which can be found by solving the Kantorovich problem. Then, we have $\mathbb{E}_{\mathbf{X}_0, \mathbf{X}_1 \sim \mathbb{Q}^*} \big( \text{FCA cost}(\mathbf{X}_0, \mathbf{X}_1, \boldsymbol{\mu}^\diamond) \big) \leq \mathbb{E}_{\mathbf{X}_0, \mathbf{X}_1 \sim \mathbb{Q}'} \big( \text{FCA cost}(\mathbf{X}_0, \mathbf{X}_1, \boldsymbol{\mu}^\diamond) \big)$.

(ii): Let $\eta(\mathbf{x}_0, \mathbf{x}_1) := \text{FCA cost}(\mathbf{x}_0, \mathbf{x}_1, \boldsymbol{\mu}^\diamond) - \text{K-means cost}(\mathbf{x}_0, \mathbf{x}_1, \boldsymbol{\mu}^\diamond)$ and $\eta_\varepsilon$ be the $\varepsilon$th upper quantile. Define

$\mathcal{W}^\star = \{(\mathbf{x}_0, \mathbf{x}_1) \in \mathcal{X}_0 \times \mathcal{X}_1 : \eta(\mathbf{x}_0, \mathbf{x}_1) > \eta_\varepsilon\}$. Then, using (i), we have

$$
\begin{aligned}
\tilde{C}(\mathbb{Q}^\star, \mathcal{W}^\star, \boldsymbol{\mu}^\diamond) &\leq \tilde{C}(\mathbb{Q}^\star, \emptyset, \boldsymbol{\mu}^\diamond) = \mathbb{E}_{\mathbf{X}_0, \mathbf{X}_1 \sim \mathbb{Q}^\star}(\text{FCA-cost}(\mathbf{X}_0, \mathbf{X}_1, \boldsymbol{\mu}^\diamond)) \\
&= \mathbb{E}_{\mathbf{X}_0, \mathbf{X}_1 \sim \mathbb{Q}^\star}(\text{FCA-cost}(\mathbf{X}_0, \mathbf{X}_1, \boldsymbol{\mu}^\diamond)) \\
&\quad - \sup_{\mathbb{Q}, \mathcal{W}: \mathbb{Q}(\mathcal{W}) \leq \varepsilon} \mathbb{E}_{\mathbf{X}_0, \mathbf{X}_1 \sim \mathbb{Q}}\big(\text{FCA cost}(\mathbf{X}_0, \mathbf{X}_1, \boldsymbol{\mu}) - \text{K-means cost}(\mathbf{X}_0, \mathbf{X}_1, \boldsymbol{\mu})\big) \mathbb{1}\left((\mathbf{X}_0, \mathbf{X}_1) \in \mathcal{W}\right) \\
&\quad + \sup_{\mathbb{Q}, \mathcal{W}: \mathbb{Q}(\mathcal{W}) \leq \varepsilon} \mathbb{E}_{\mathbf{X}_0, \mathbf{X}_1 \sim \mathbb{Q}}\big(\text{FCA cost}(\mathbf{X}_0, \mathbf{X}_1, \boldsymbol{\mu}) - \text{K-means cost}(\mathbf{X}_0, \mathbf{X}_1, \boldsymbol{\mu})\big) \mathbb{1}\left((\mathbf{X}_0, \mathbf{X}_1) \in \mathcal{W}\right) \\
&\leq \mathbb{E}_{\mathbf{X}_0, \mathbf{X}_1 \sim \mathbb{Q}'}(\text{FCA-cost}(\mathbf{X}_0, \mathbf{X}_1, \boldsymbol{\mu}^\diamond)) \\
&\quad - \mathbb{E}_{\mathbf{X}_0, \mathbf{X}_1 \sim \mathbb{Q}'}\big(\text{FCA cost}(\mathbf{X}_0, \mathbf{X}_1, \boldsymbol{\mu}) - \text{K-means cost}(\mathbf{X}_0, \mathbf{X}_1, \boldsymbol{\mu})\big) \mathbb{1}\left((\mathbf{X}_0, \mathbf{X}_1) \in \mathcal{W}'\right) \\
&\quad + \sup_{\mathbb{Q}, \mathcal{W}: \mathbb{Q}(\mathcal{W}) \leq \varepsilon} \mathbb{E}_{\mathbf{X}_0, \mathbf{X}_1 \sim \mathbb{Q}}\big(\text{FCA cost}(\mathbf{X}_0, \mathbf{X}_1, \boldsymbol{\mu}) - \text{K-means cost}(\mathbf{X}_0, \mathbf{X}_1, \boldsymbol{\mu})\big) \mathbb{1}\left((\mathbf{X}_0, \mathbf{X}_1) \in \mathcal{W}\right) \\
&= \tilde{C}(\mathbb{Q}', \mathcal{W}', \boldsymbol{\mu}^\diamond) \\
&\quad + \sup_{\mathbb{Q}, \mathcal{W}: \mathbb{Q}(\mathcal{W}) \leq \varepsilon} \mathbb{E}_{\mathbf{X}_0, \mathbf{X}_1 \sim \mathbb{Q}}\big(\text{FCA cost}(\mathbf{X}_0, \mathbf{X}_1, \boldsymbol{\mu}) - \text{K-means cost}(\mathbf{X}_0, \mathbf{X}_1, \boldsymbol{\mu})\big) \mathbb{1}\left((\mathbf{X}_0, \mathbf{X}_1) \in \mathcal{W}\right).
\end{aligned}
\tag{25}
$$

(iii) The last term of the right-hand-side can be bounded as follows:

First, let $\mu^\diamond(\mathbf{x}) := \arg\min_{\mu_k^\diamond \in \boldsymbol{\mu}^\diamond} \|\mathbf{x} - \mu_k^\diamond\|^2$ for $\mathbf{x} \in \mathcal{X}$ and $\mu^\diamond(\mathbf{x}_0, \mathbf{x}_1) := \arg\min_{\mu_k^\diamond \in \boldsymbol{\mu}^\diamond} \|\mathbf{T}^A(\mathbf{x}_0, \mathbf{x}_1) - \mu_k^\diamond\|^2$ for $(\mathbf{x}_0, \mathbf{x}_1) \in \mathcal{X}_0 \times \mathcal{X}_1$. Then, $\forall (\mathbf{x}_0, \mathbf{x}_1) \in \mathcal{X}_0 \times \mathcal{X}_1$, it holds that

$$
\begin{aligned}
&\text{FCA cost}(\mathbf{x}_0, \mathbf{x}_1, \boldsymbol{\mu}^\diamond) - \text{K-means cost}(\mathbf{x}_0, \mathbf{x}_1, \boldsymbol{\mu}^\diamond) \\
&= 2\pi_0 \pi_1 \|\mathbf{x}_0 - \mathbf{x}_1\|^2 + \|\mathbf{T}^A(\mathbf{x}_0, \mathbf{x}_1) - \mu^\diamond(\mathbf{x}_0, \mathbf{x}_1)\|^2 - \left(\pi_0 \|\mathbf{x}_0 - \mu^\diamond(\mathbf{x}_0)\|^2\right) - \left(\pi_1 \|\mathbf{x}_1 - \mu^\diamond(\mathbf{x}_1)\|^2\right) \\
&\leq 2\pi_0 \pi_1 \|\mathbf{x}_0 - \mathbf{x}_1\|^2 + \|\mathbf{T}^A(\mathbf{x}_0, \mathbf{x}_1) - \mu^\diamond(\mathbf{x}_0)\|^2 - \left(\pi_0 \|\mathbf{x}_0 - \mu^\diamond(\mathbf{x}_0)\|^2\right) - \left(\pi_1 \|\mathbf{x}_1 - \mu^\diamond(\mathbf{x}_1)\|^2\right) \\
&\leq 2\pi_0 \pi_1 \|\mathbf{x}_0 - \mathbf{x}_1\|^2 + \pi_0 \|\mathbf{x}_0 - \mu^\diamond(\mathbf{x}_0)\|^2 + \pi_1 \|\mathbf{x}_1 - \mu^\diamond(\mathbf{x}_0)\|^2 \\
&\quad - \left(\pi_0 \|\mathbf{x}_0 - \mu^\diamond(\mathbf{x}_0)\|^2\right) - \left(\pi_1 \|\mathbf{x}_1 - \mu^\diamond(\mathbf{x}_1)\|^2\right) \\
&= 2\pi_0 \pi_1 \|\mathbf{x}_0 - \mathbf{x}_1\|^2 + \pi_1 (\|\mathbf{x}_1 - \mu^\diamond(\mathbf{x}_0)\|^2 - \|\mathbf{x}_1 - \mu^\diamond(\mathbf{x}_1)\|^2) \\
&\leq 2\pi_0 \pi_1 \|\mathbf{x}_0 - \mathbf{x}_1\|^2 + \pi_1 \|\mathbf{x}_1 - \mu^\diamond(\mathbf{x}_0)\|^2 \\
&\leq \frac{1}{2} \|\mathbf{x}_0 - \mathbf{x}_1\|^2 + \|\mathbf{x}_1 - \mu^\diamond(\mathbf{x}_0)\|^2 \leq \frac{1}{2} 2R + 2R = 3R.
\end{aligned}
\tag{26}
$$

Hence, we conclude that

$$
\sup_{\mathbb{Q}, \mathcal{W}: \mathbb{Q}(\mathcal{W}) \leq \varepsilon} \mathbb{E}_{\mathbf{X}_0, \mathbf{X}_1 \sim \mathbb{Q}}\big(\text{FCA cost}(\mathbf{X}_0, \mathbf{X}_1, \boldsymbol{\mu}) - \text{K-means cost}(\mathbf{X}_0, \mathbf{X}_1, \boldsymbol{\mu})\big) \mathbb{1}\left((\mathbf{X}_0, \mathbf{X}_1) \in \mathcal{W}\right)
$$

$$
\leq 3R \sup_{\mathbb{Q}, \mathcal{W}: \mathbb{Q}(\mathcal{W}) = \varepsilon} \mathbb{E}_{\mathbf{X}_0, \mathbf{X}_1 \sim \mathbb{Q}}(\mathbb{1}\left((\mathbf{X}_0, \mathbf{X}_1) \in \mathcal{W}\right)) = 3R\mathbb{Q}(\mathcal{W}) = 3R\varepsilon.
$$

Finally, combining (ii) and (iii), we have $\tilde{C}(\mathbb{Q}^\star, \mathcal{W}^\star, \boldsymbol{\mu}^\diamond) \leq \tilde{C}(\mathbb{Q}', \mathcal{W}', \boldsymbol{\mu}^\diamond) + 3R\varepsilon$, which implies $C(\boldsymbol{\mu}^\diamond, \mathcal{A}_0^\star, \mathcal{A}_1^\star) \leq C(\boldsymbol{\mu}^\diamond, \mathcal{A}_0', \mathcal{A}_1') + 3R\varepsilon$, where $\mathcal{A}_0^\star$ and $\mathcal{A}_1^\star$ are the fair assignment functions corresponding to $\mathbb{Q}^\star$ and $\mathcal{W}^\star$.

*Remark* B.3. Finally, FCA-C algorithm iterates the above process. Using FCA-C, we find $\mathcal{A}_0''$ and $\mathcal{A}_1''$, which minimizes the cost given $\boldsymbol{\mu}'$, where $\mu_k' := \sum_{i=1}^n \mathcal{A}_s'(\mathbf{x}_i)_k \mathbf{x}_i / \sum_{i=1}^n \mathcal{A}_s'(\mathbf{x}_i)_k$, i.e., the minimizer of $\min_{\boldsymbol{\mu}} C(\boldsymbol{\mu}, \mathcal{A}_0', \mathcal{A}_1')$ given $\mathcal{A}_0'$ and $\mathcal{A}_1'$. Again, let $\mu_k'' := \sum_{i=1}^n \mathcal{A}_s''(\mathbf{x}_i)_k \mathbf{x}_i / \sum_{i=1}^n \mathcal{A}_s''(\mathbf{x}_i)_k$, which is the minimizer of $\min_{\boldsymbol{\mu}} C(\boldsymbol{\mu}, \mathcal{A}_0'', \mathcal{A}_1'')$ given $\mathcal{A}_0''$ and $\mathcal{A}_1''$.

This iteration process results in the following inequality: $C(\boldsymbol{\mu}'', \mathcal{A}_0'', \mathcal{A}_1'') \leq C(\boldsymbol{\mu}', \mathcal{A}_0'', \mathcal{A}_1'') \leq C(\boldsymbol{\mu}', \mathcal{A}_0', \mathcal{A}_1') \leq (\tau + 2)C(\tilde{\boldsymbol{\mu}}, \tilde{\mathcal{A}}_0, \tilde{\mathcal{A}}_1) + 3R\varepsilon$. Hence, iterating this process until convergence guarantees the approximation error of $\tau + 2$ with additional violation of $3R\varepsilon$, which concludes that $C(\boldsymbol{\mu}^\triangle, \mathcal{A}_0^\triangle, \mathcal{A}_1^\triangle) \leq (\tau + 2)C(\tilde{\boldsymbol{\mu}}, \tilde{\mathcal{A}}_0, \tilde{\mathcal{A}}_1) + 3R\varepsilon$.

$\square$

# C. Experiments

## C.1. Datasets

The three datasets used in our experiments can be found in the UCI Machine Learning Repository[3].

- ADULT dataset (Becker & Kohavi, 1996) can be downloaded from `https://archive.is.uci/ml/datasets/adult`. We use 5 continuous variables (`age`, `fnlwgt`, `education`, `capital-gain`, and `hours-per-week`). The total sample size is 32,561. The sample size for $S = 0$ and $S = 1$ (resp. female and male) are 10,771 and 21,790, respectively.

- BANK dataset (Moro et al., 2012) can be downloaded from `https://archive.ics.uci.edu/ml/datasets/Bank+Marketing`. We use 6 continuous variables (`age`, `duration`, `euribor3m`, `nr.employed`, `cons.price.idx`, and `campaign`). The total sample size is 41,108. The sample size for $S = 0$ and $S = 1$ (resp. not married and married) are 16,180 and 24,928, respectively.

- CENSUS dataset (Meek et al.) can be downloaded from `https://archive.ics.uci.edu/ml/datasets/US+Census+Data+(1990)`. We use 66 continuous variables (`dAncstry1`, `dAncstry2`, `iAvail`, `iCitizen`, `iClass`, `dDepart`, `iDisabl1`, `iDisabl2`, `iEnglish`, `iFeb55`, `iFertil`, `dHispanic`, `dHour89`, `dHours`, `iImmigr`, `dIncome1`, `dIncome2`, `dIncome3`, `dIncome4`, `dIncome5`, `dIncome6`, `dIncome7`, `dIncome8`, `dIndustry`, `iKorean`, `iLang1`, `iLooking`, `iMarital`, `iMay75880`, `iMeans`, `iMilitary`, `iMobility`, `iMobillim`, `dOccup`, `iOthrserv`, `iPerscare`, `dPOB`, `dPoverty`, `dPwgt1`, `iRagechld`, `dRearning`, `iRelat1`, `iRelat2`, `iRemplpar`, `iRiders`, `iRlabor`, `iRownchld`, `dRpincome`, `iRPOB`, `iRrelchld`, `iRspouse`, `iRvetserv`, `iSchool`, `iSept80`, `iSubfam1`, `iSubfam2`, `iTmpabsnt`, `dTravtime`, `iVietnam`, `dWeek89`, `iWork89`, `iWorklwk`, `iWWII`, `iYearsch`, `iYearwrk`, and `dYrsserv`). We subsample 20,000 instances, as done in Chierichetti et al. (2017); Bera et al. (2019); Esmaeili et al. (2021). The sample size for $S = 0$ and $S = 1$ (not married and married, respectively) are 9,844 and 10,156 respectively.

Note that, for each dataset, we use only the continuous (numerical) variables as introduced above, consistent with previous studies, e.g., Backurs et al. (2019); Bera et al. (2019); Esmaeili et al. (2021); Ziko et al. (2021), to name a few. Particularly, the features we consider in this paper are the same as those selected in a baseline method VFC (Ziko et al., 2021). The variables of all datasets are scaled with zero mean and unit variance.

## C.2. Implementation details

### C.2.1. COMPUTING RESOURCES

The computation is performed on several Intel Xeon Silver CPU cores and an additional RTX 4090 GPU processor.

### C.2.2. PROPOSED ALGORITHMS

**FCA** We use the `POT` library (Flamary et al., 2021) to find the optimal joint distribution $\mathbb{Q}$ (i.e., $\Gamma$). For updating the cluster centers, we adopt the $K$-means++ algorithm (Arthur & Vassilvitskii, 2007) from the implementation of `scikit-learn` package Pedregosa et al. (2011). The iterative process of updating the cluster centers and the joint distribution is run for 100 iterations, with the result where `Cost` is minimized being selected.

**FCA-C** To control `Bal`, we vary $\varepsilon$, i.e., the size of $\mathcal{W}$. The value of $\varepsilon$ is swept in increments of 0.05, ranging from 0.1 to 0.9. We also use a similar partitioning technique in Section 4.3 for FCA-C with $m = 2048$. For stability, at the first iteration step, we find $\mathcal{W}$ based on fixed cluster centers $\boldsymbol{\mu}$, initialized using the $K$-means++ algorithm.

### C.2.3. BASELINE ALGORITHMS

**Scalable Fair Clustering (SFC)** (Backurs et al., 2019) We directly use the official source code of SFC, available on the authors' GitHub[4]. SFC provides a fast and scalable algorithm for fairlet decomposition, which builds the fairlets in nearly

---

[3] `https://archive.ics.uci.edu/datasets`
[4] `https://github.com/talwagner/fair_clustering`

linear time. The given data are first embedded to a tree structure (hierarchically well-separated tree; HST) using probabilistic metric embedding, where we seek for optimal edges (as well as their nodes) to be activated that satisfy `Bal ≈ Bal⋆`. Then, the linked nodes are then aggregated as a fairlet. This process can build fairlets in nearly linear time while minimizing the cost of building them. After building these fairlets using SFC, we apply the standard $K$-means algorithm on the fairlet space (i.e., the set of representatives of the obtained fairlets).

**Fair Clustering under a Bounded Cost (FCBC) (Esmaeili et al., 2021)**  We use the official source code of FCBC, available on the authors' GitHub[5]. FCBC maximizes `Bal` under a cost constraint, where the cost constraint is defined by the Price of Fairness (PoF) = 'cost of fair clustering (the solution) / cost of standard clustering'. However, since the authors mentioned the constrained optimization problem is NP-hard, they reduced the problem to a post-processing approach (i.e., fairly assigning data under the cost constraint, with pre-specified centers opened by a standard clustering algorithm). We set the value of PoF to 1.2 to achieve `Bal ≈ Bal*`.

**Fair Round-Robin Algorithm for Clustering (FRAC) (Gupta et al., 2023)**  We directly follow the official source code of FRAC, available on the authors' GitHub[6]. FRAC provides an in-loop post-processing approach to fairly assign data from each protected group to given cluster centers found by a standard clustering algorithm. In other words, the fair assignment problem is solved at each iteration of standard clustering algorithm.

**Variational Fair Clustering (VFC) (Ziko et al., 2021)**  We use the official source code of VFC, available on the authors' GitHub[7]. The overall objective of VFC is $\mathbb{E}\min_k \|\mathbf{X} - \mu_k\|^2 + \lambda \cdot \sum_{k=1}^{K} \text{KL}\left([\pi_0, \pi_1]\| \left[\frac{\pi_0 \mathbb{E}_0 \mathcal{A}_0(\mathbf{X})_k}{\mathbb{E}\mathcal{A}_S(\mathbf{X})_k}, \frac{\pi_1 \mathbb{E}_1 \mathcal{A}_1(\mathbf{X})_k}{\mathbb{E}\mathcal{A}_S(\mathbf{X})_k}\right]\right)$, where $\lambda$ is a hyper-parameter to control `Bal`. A higher $\lambda$ results in higher `Bal`, with KL $= 0 \iff$ `Bal` $= 1$. We run the code with multiple trials, by varying the values of $\lambda$. For Table 2, we select the best $\lambda$ that achieves the highest `Bal` and report the corresponding performance for the chosen hyper-parameter, as the authors have done. For Figure 3, we present all the results obtained using the various values of $\lambda$. Table 5 below provides the values of VFC's hyper-parameters used in our experiments.

*Table 5.* The hyper-parameters used in VFC for searching a clustering with maximum achievable `Bal` for each dataset. The **bold** faces are the recommended ones by the authors. The underlined values are the ones we use for maximum achievable `Bal`.

| Dataset | $L_2$ normalization | Hyper-parameters |
|---|---|---|
| ADULT | O | $\{5000, 7000, \mathbf{9000}, 10000, 11000, 12000, 13000, 13600, \underline{14200}\}$ |
|  | X | $\{5000, 7000, 9000, 10000, 11000, 12000, 13000, 15000, 20000, 22000, \underline{23000}\}$ |
| BANK | O | $\{5000, \mathbf{6000}, 7000, 9000, 10000, 11000, 12000, \underline{12300}\}$ |
|  | X | $\{10000, 12000, 13000, 15000, 17000, 19000, 25200, \underline{26000}\}$ |
| CENSUS | O | $\{100, 200, 500, 700, 1000, 1500, \underline{2000}\}$ |
|  | X | *Failed* |

Note that VFC's superior performance over other two well-known FC algorithms from Bera et al. (2019); Kleindessner et al. (2019) was already shown in Ziko et al. (2021), which is why we omit these two methods as baselines in our experiments.

---

[5] https://github.com/Seyed2357/Fair-Clustering-Under-Bounded-Cost
[6] https://github.com/shivi98g/Fair-k-means-Clustering-via-Algorithmic-Fairness
[7] https://github.com/imtiazziko/Variational-Fair-Clustering

## C.3. Omitted experimental results

### C.3.1. PERFORMANCE COMPARISON RESULTS (SECTION 5.2)

**Trade-off: fairness level vs. clustering utility**  Table 6 presents the comparison results for the trade-off between `Bal` and `Cost` without $L_2$ normalization, which is similar to Table 1.

*Table 6.* Comparison of the trade-off between `Bal` and `Cost` on ADULT, BANK, and CENSUS datasets, when data are not $L_2$-normalized. We underline `Bal` values for the cases of near-perfect fairness (i.e., `Bal` $\approx$ `Bal`$^\star$) and use **bold face for the lowest** `Cost` value among those cases.

| Dataset / `Bal`$^\star$ | ADULT / 0.494 | | BANK / 0.649 | | CENSUS / 0.969 | |
|---|---|---|---|---|---|---|
| Without $L_2$ normalization | Cost (↓) | Bal (↑) | Cost (↓) | Bal (↑) | Cost (↓) | Bal (↑) |
| Standard (fair-unaware) | 1.620 | 0.206 | 1.510 | 0.391 | 28.809 | 0.030 |
| FCBC (Esmaeili et al., 2021) | 1.851 | 0.460 | 2.013 | 0.610 | 59.988 | 0.925 |
| SFC (Backurs et al., 2019) | 3.399 | 0.471 | 3.236 | 0.622 | 69.437 | 0.940 |
| FRAC (Gupta et al., 2023) | 2.900 | 0.488 | 2.716 | 0.646 | 38.430 | 0.962 |
| FCA ✓ | **1.875** | 0.492 | **1.859** | 0.647 | **33.472** | 0.959 |

On the other hand, as shown in Table 1, FCBC attains a slightly lower `Cost` (0.314) than FCA, whereas FCA yields a higher fairness level. To fairly compare the two methods at equal `Cost`, we run FCA-C targeting `Cost` $\approx 0.314$, i.e., the `Cost` of FCBC. Under this constraint, FCA-C achieves a `Balance` of 0.473, compared to 0.443 of FCBC, demonstrating that FCA offers a superior trade-off between `Bal` and `Cost`.

*Table 7.* Comparison of FCBC and FCA-C in terms of the trade-off between `Bal` and `Cost` on ADULT dataset, when data are $L_2$-normalized. `Cost` is fixed near 0.314 for a fair comparison.

| ADULT | | |
|---|---|---|
| `Bal`$^\star = 0.494$ | Cost (↓) | Bal (↑) |
| FCBC (Esmaeili et al., 2021) | 0.314 | 0.443 |
| FCA-C ✓ | 0.313 | **0.473** |

For our main experiments in Section 5, we mainly compare FCA with the scalable fairlet-based method of Backurs et al. (2019). Here, we additionally compare FCA with the original fairlet-based approach introduced by Chierichetti et al. (2017). Table 8 shows that FCA outperforms the fairlet-based method of Chierichetti et al. (2017) as well as SFC.

*Table 8.* Comparison of Chierichetti et al. (2017), SFC and FCA in terms of the trade-off between `Bal` and `Cost` on ADULT, BANK, and CENSUS datasets, when data are $L_2$-normalized. The **bold**-faced results indicate the bests.

| Dataset / `Bal`$^\star$ | ADULT / 0.494 | | BANK / 0.649 | | CENSUS / 0.969 | |
|---|---|---|---|---|---|---|
| With $L_2$ normalization | Cost (↓) | Bal (↑) | Cost (↓) | Bal (↑) | Cost (↓) | Bal (↑) |
| Chietichetti et al. (2017) | 0.507 | 0.488 | 0.378 | 0.639 | 1.124 | 0.941 |
| SFC | 0.534 | 0.489 | 0.410 | 0.632 | 1.015 | 0.937 |
| FCA ✓ | **0.328** | **0.493** | **0.264** | **0.645** | **0.477** | **0.962** |

**Numerical stability** Table 9 presents the comparison results of FCA and VFC in terms of numerical stability without $L_2$ normalization, which is similar to Table 2. In specific, on CENSUS dataset without $L_2$ normalization, VFC fails, i.e., it does not achieve higher Bal than the standard clustering for any $\lambda$. We find that the failure of VFC on CENSUS dataset (without $L_2$ normalization) is due to explosion of an exponential term calculated in its algorithm. There exists an exponential term with respect to the clustering cost in the calculation of the optimal assignment vector in the VFC algorithm, and thus when the clustering cost becomes too large, VFC fails due to an overflow. Note that the input dimension of CENSUS dataset is 66, while those of ADULT and BANK are 5 and 6, respectively. While the clustering cost with $L_2$ normalization is bounded regardless of the dimension, the clustering cost without $L_2$ normalization is proportional to the input dimension. This is why VFC fails only for CENSUS dataset without $L_2$ normalization. In contrast, as FCA does not fail at all regardless of the $L_2$ normalization because there is no exponential term in the algorithm, meaning that it is numerically more stable than VFC.

*Table 9.* Comparison of the two in-processing algorithms, VFC (Ziko et al., 2021) and FCA, in terms of numerical stability when achieving the maximum Bal, on ADULT, BANK, and CENSUS datasets without $L_2$ normalization. **Bold**-faced results are the highest values of Bal.

| Dataset / Bal* | Bal(Cost) | |
|---|---|---|
| | VFC (Ziko et al., 2021) | FCA ✓ |
| ADULT / 0.494 | 0.310 (1.688) | **0.493** (1.875) |
| BANK / 0.649 | 0.505 (1.549) | **0.647** (1.859) |
| CENSUS / 0.969 | Failed | **0.959** (33.472) |

**Control of fairness level** We also provide the full results (i.e., Figures 4 and 5 where the data are $L_2$-normalized and not $L_2$-normalized, respectively) showing the trade-off of VFC and FCA-C for various fairness levels, on ADULT, BANK and CENSUS datasets. FCA and VFC show similar trade-offs, while VFC cannot achieve sufficiently high values of Bal (the orange dashed lines are the limit values of balance that VFC can achieve).

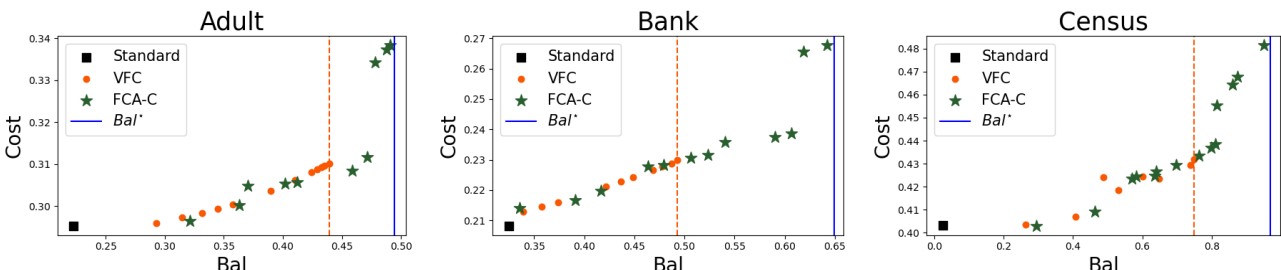

*Figure 4.* Bal vs. Cost trade-offs on (left) ADULT, (center) BANK and (right) CENSUS datasets. Black square (■) is from the standard clustering, orange circle (●) is from VFC, green star (★) is from FCA-C, orange dashed line (- -) is the maximum of Bal that VFC can achieve, and blue line (–) is the maximum achievable balance Bal*.

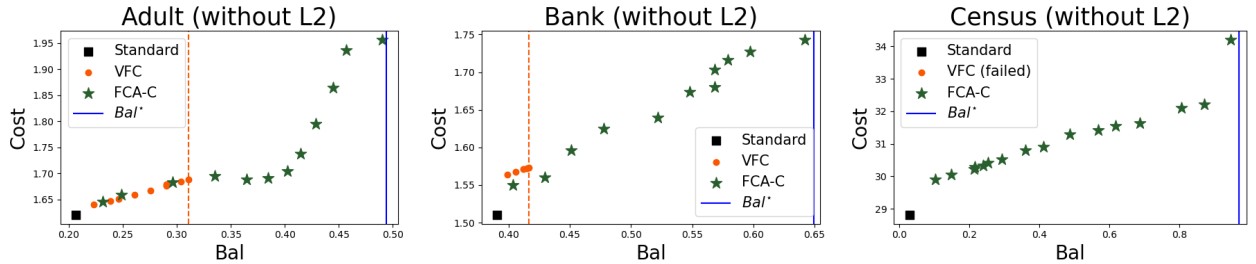

*Figure 5.* Bal vs. Cost trade-offs on (left) ADULT, (center) BANK and (right) CENSUS datasets. Black square (■) is from the standard clustering, orange circle (●) is from VFC, green star (★) is from FCA-C, orange dashed line (- -) is the maximum of Bal that VFC can achieve, and blue line (–) is the maximum achievable balance Bal*. The data are not $L_2$-normalized.

**Comparison in terms of the silhouette score**  As measuring `Cost` alone may not fully capture the clustering quality, we additionally consider another measure, called the silhouette score. The silhouette score (we abbreviate by `Silhouette`) is computed as the average of $(d_{\text{near}} - d_{\text{intra}})/\max(d_{\text{intra}}, d_{\text{near}})$ over all data points, where $d_{\text{intra}}$ denotes the intra-cluster distance and $d_{\text{near}}$ represents the average distance to the nearest neighboring cluster. Notably, the results in Table 10 show that FCA is also superior or competitive to baselines in terms of the silhouette score.

*Table 10.* Comparison of the `Silhouette` and `Bal` on ADULT dataset, when the data are $L_2$-normalized.

| Dataset / `Bal`$^\star$ | ADULT / 0.494 | |
| --- | --- | --- |
| With $L_2$ normalization | `Silhouette` (↑) | `Bal` (↑) |
| Standard (fair-unaware) | 0.227 | 0.223 |
| FCBC | 0.173 | 0.443 |
| SFC | 0.071 | 0.489 |
| FRAC | 0.156 | 0.490 |
| FCA ✓ | **0.176** | **0.493** |

**Analysis on an additional dataset**  We additionally conduct an analysis on CREDITCARD dataset ($n = 30000$) from Yeh & Lien (2009), which was also used in Bera et al. (2019); Harb & Lam (2020). We directly follow the data preprocessing of Bera et al. (2019). We use gender as the sensitive attribute and set $K = 10$. The results are provided in Table 11 below, where we can observe that FCA outperforms the baseline methods on CREDITCARD dataset as well.

*Table 11.* Comparison of the `Cost` and `Bal` on CREDITCARD dataset, when the data are $L_2$-normalized.

| Dataset / `Bal`$^\star$ | CREDITCARD / 0.656 | |
| --- | --- | --- |
| With $L_2$ normalization | `Cost` (↓) | `Bal` (↑) |
| Standard (fair-unaware) | 0.392 | 0.506 |
| FCBC | 0.492 | 0.629 |
| SFC | 0.682 | **0.653** |
| FRAC | 0.510 | 0.649 |
| FCA ✓ | **0.402** | **0.653** |

## C.3.2. APPLICABILITY TO VISUAL CLUSTERING (SECTION 5.3)

**Settings: datasets, baselines, and measures**    RMNIST is a mixture of two image digit datasets: the original MNIST and a color-reversed version (where black and white are swapped). OFFICE-31 is a mixture of two datasets from two domains (amazon and webcam) with 31 classes. Both datasets are used in state-of-the-art visual FC methods (Li et al., 2020; Zeng et al., 2023).

For the baseline, we consider a state-of-the-art FC method in vision domain called FCMI from Zeng et al. (2023). FCMI learns a fair autoencoder with two additional loss terms: (i) clustering loss on the latent space and (ii) mutual information between latent vector and color. While FCMI is an end-to-end method that learns the fair latent vector and perform clustering on the fair latent space simultaneously, FCA is applied to a pre-trained latent space obtained by learning an autoencoder with the reconstruction loss only. We also report the performances of DFC (Li et al., 2020) which was the main baseline in the FCMI paper, along with SFC (Backurs et al., 2019) and VFC (Ziko et al., 2021), even though these two methods are not specifically designed for the vision domain.

The clustering performance for the two image datasets is evaluated by two classification measures `ACC` (accuracy calculated based on assigned cluster indices and ground-truth classification labels) and `NMI` (normalized mutual information between ground-truth label distribution and assigned cluster distribution), which are consistently used in Li et al. (2020); Zeng et al. (2023), as datasets involve ground-truth classification labels (e.g., $\{0, 1, \dots, 9\}$ for RMNIST and the 31 classes for OFFICE-31). The fairness level is also evaluated by `Bal`.

**Results**    Table 12 shows that FCA performs similar to FCMI, which is the state-of-the-art, while outperforming the other baselines with large margins in terms of `Bal`. Note that SFC, VFC, and FCA are two-step methods, i.e., they find fair clustering on the pre-trained (fair-unaware) latent space, and FCA is the best among those. Furthermore, on the other hand, DFC and FCMI are end-to-end methods so it is noteworthy that FCA outperforms DFC and performs similarly to FCMI.

*Table 12.* Comparison of clustering utility (`ACC` and `NMI`) and fairness level (`Bal`) on two image datasets. 'Standard (fair-unaware)' indicates autoencoder + $K$-means (Vincent et al., 2010). First-place values are **bold**, and second-place values are underlined. The performances of the baselines reflects the better results between our re-implementation and the one reported by Zeng et al. (2023).

| Dataset / `Bal`* | RMNIST / 1.000 | | | OFFICE-31 / 0.282 | | |
|---|---|---|---|---|---|---|
| Performance | ACC (↑) | NMI (↑) | Bal (↑) | ACC (↑) | NMI (↑) | Bal (↑) |
| Standard (fair-unaware) | 41.0 | 52.8 | 0.000 | 63.8 | 66.8 | 0.192 |
| SFC (Backurs et al., 2019) | 51.3 | 49.1 | **1.000** | 61.6 | 61.2 | 0.267 |
| VFC (Ziko et al., 2021) | 38.1 | 42.7 | 0.000 | 64.8 | 70.4 | 0.212 |
| DFC (Li et al., 2020) | 49.9 | 68.9 | 0.800 | 69.0 | 70.9 | 0.165 |
| FCMI (Zeng et al., 2023) | 88.4 | **86.4** | 0.995 | **70.0** | **71.2** | 0.226 |
| FCA ✓ | **89.0** | 79.0 | **1.000** | 67.6 | 70.5 | **0.270** |

**Further comparison between the fairlet-based method and FCA in visual clustering**    We compare the fairlet-based method and FCA using OFFICE-31 dataset, considering (i) not only the overall clustering utility, (ii) but also the similarity of matched features. For a clear comparison, we sample a balanced subset (with respect to the label and sensitive attribute) from the original dataset, which is imbalanced. That is, we ensure that the number of samples with the same label is equal across the two protected groups (i.e., the two domains). As a result, we obtain 795 images from both the amazon and webcam domains. We then find fairlets or apply FCA on this balanced dataset. Note that finding fairlets when $n_0 = n_1$ is equivalent to finding the optimal transport map (Villani, 2008; Chierichetti et al., 2017; Peyré & Cuturi, 2020).

*Table 13.* Comparison of the fairlet-based method and FCA. 'Matching cost' is defined by the average distance between two matched features. 'Matching = Label' is defined by the average ratio of images with the same label being matched. **Bold**-faced values indicate the best performance.

| Matching method | Matching performance | | Clustering performance | | |
|---|---|---|---|---|---|
| | Matching cost | Matching = Label (↑) | Cost (↓) | ACC (↑) | NMI (↑) |
| Fairlet-based | **0.211** | 0.595 | 0.278 | 65.8 | 71.0 |
| FCA ✓ | 0.241 | **0.631** | **0.269** | **69.3** | **72.2** |

Table 13 presents the comparison results using various measures, including performance measures with respect to matching (the matching cost and how much images with the same label are matched) and clustering (`Cost`, `ACC`, and `NMI`). While the fairlets tend to match more similar features (i.e., the matching cost is lower) as expected by the definition of fairlets, FCA exhibits a better ability to collect images with same labels into clusters (i.e., lower `Cost`, higher `ACC`, and higher `NMI`). Moreover, in visual clustering, matching similar features in the pre-trained latent space does not always guarantee that images with the same label are matched (FCA achieves a higher proportion of matchings where images share the same label compared to fairlets). These results suggest that while fairlets provide optimal matchings in terms of feature similarity, however, they may be suboptimal in terms of label similarity and overall clustering utility.

### C.3.3. Ablation study: (1) selection of the partition size $m$ (Section 5.4)

This section provided empirical evidence for the partitioning technique introduced in Section 4.3. First, Figure 6 indicates that using a partition size of around 1000 yields reasonable results. Specifically, using $m$ values greater than 1000 shows similar performance compared to those obtained with $m = 1024$.

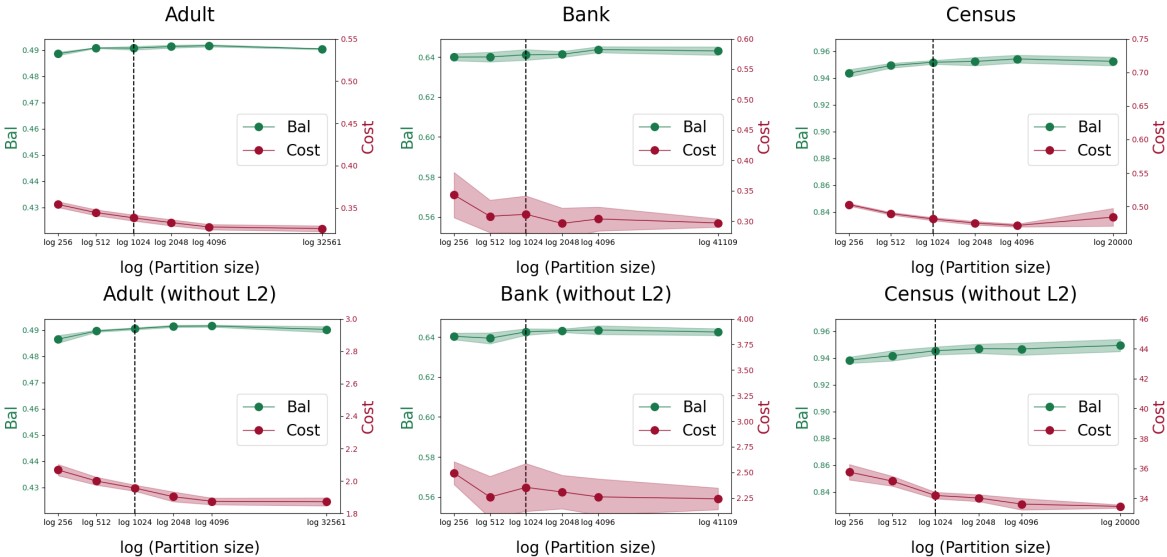

*Figure 6.* Variations of `Cost` and `Bal` with respect to the partition size. (Left, Center, Right) = (Adult, Bank, Census). (Top, Bottom) = (With $L_2$ normalization, Without $L_2$ normalization).

We further provide the elapsed computation time for various partition sizes, up to using the full dataset. Using $m = 1024$ as the baseline, we calculate the relative computation time (%) for other partition sizes. The comparison, presented in Table 14 below, shows that using $m = 1024$ leads to a significant reduction in computation time.

*Table 14.* Comparison of computation time with different partition sizes, up to using the full dataset. For each partition size and dataset, we provide the averaged relative elapsed time per iteration, when compared to computation time for $m = 1024$.

| (Relative) elapsed time per iteration | Partition size $m$ | | | | | |
| --- | --- | --- | --- | --- | --- | --- |
| | 256 | 512 | 1024 ✓ | 2048 | 4096 | Full |
| Adult ($n = 32561, d = 5$) | 17% | 58% | 100% | 140% | 344% | 3,184% |
| Bank ($n = 41108, d = 6$) | 14% | 43% | 100% | 161% | 288% | 3,308% |
| Census ($n = 20000, d = 66$) | 23% | 52% | 100% | 176% | 375% | 1,064% |

Additionally, we observe that computation time is linear in $m^2$, as shown in Figure 7 below. This numerical result can support the theoretical computational complexity $\mathcal{O}(nm^2)$ described in Section 4.3. See Table 18 in Appendix C.3.6 for another result showing that the computation time is also linear in $n$.

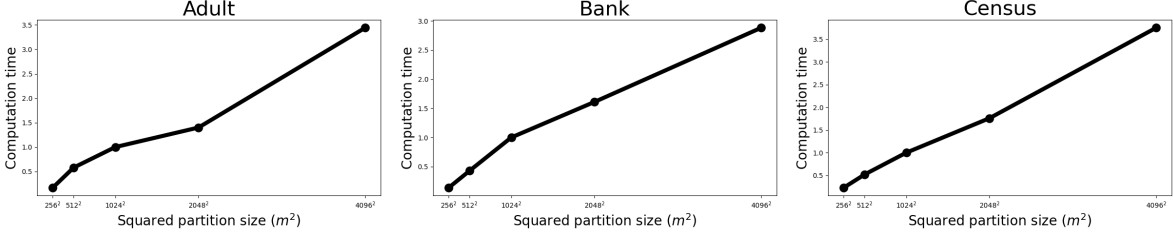

*Figure 7.* Squared partition size $m^2$ vs. Relative computation time. (Left, Center, Right) = (Adult, Bank, Census).

C.3.4. ABLATION STUDY: (2) OPTIMIZATION AND INITIALIZATION OF CLUSTER CENTERS (SECTION 5.4)

**Optimization algorithm of cluster centers**    We analyze how the performance of FCA varies depending on the optimization algorithm for finding cluster centers.

(i) For the $K$-means algorithm whose results are reported in the main body, we use the $K$-means++ initialization (Arthur & Vassilvitskii, 2007) from the implementation of `scikit-learn` package (Pedregosa et al., 2011). Note that we use the algorithm of Lloyd (1982) for the $K$-means algorithm.

(ii) We additionally consider random initialization of cluster centers with Lloyd's algorithm.

(iii) For the gradient-based algorithm, we use Adam optimizer (Kingma & Ba, 2014). We set a learning rate of 0.005 for CENSUS dataset with $L_2$ normalization, and 0.05 for all other cases. To accelerate convergence, 20 gradient steps of updating the centers are performed per iteration.

Table 15 presents the results comparing these three approaches, showing that FCA is robust to the choice of the optimization algorithm for finding cluster centers. Note that, while the gradient-based algorithm is also effective and accurate, it requires additional practical considerations such as selections of the learning rate and optimizer. Furthermore, the slight outperformance of $K$-means++ initialization over random initialization suggests that an efficient initialization algorithm can enhance the final performance of FCA. This is also theoretically examined through the approximation error in Appendix B.5, which depends on the approximation error of the standard algorithm for finding cluster centers without fairness constraints. However, since the margins are small, FCA can be considered empirically robust to the initialization.

*Table 15.* Comparison of performance with respect to optimization algorithms for finding cluster centers with $L_2$ normalization (top) and without $L_2$ normalization (bottom). '$K$-means++' indicates that the initial centers are set according to the $K$-means++ initialization in the first iteration, then apply the algorithm of Lloyd (1982). '$K$-means random' indicates that the initial centers are set randomly at the first iteration, then apply the algorithm from Lloyd (1982). 'Gradient-based' indicates that the initial centers are set randomly, and the centers are subsequently updated using the Adam optimizer.

| Dataset / `Bal`* | | ADULT / 0.494 | | BANK / 0.649 | | CENSUS / 0.969 | |
|---|---|---|---|---|---|---|---|
| With $L_2$ normalization | | Cost | Bal | Cost | Bal | Cost | Bal |
| FCA ($K$-means++) | | 0.328 | 0.493 | 0.264 | 0.645 | 0.477 | 0.962 |
| FCA ($K$-means random) | | 0.331 | 0.490 | 0.275 | 0.646 | 0.477 | 0.955 |
| FCA (Gradient-based) | | 0.339 | 0.492 | 0.254 | 0.640 | 0.478 | 0.957 |

| Dataset / `Bal`* | | ADULT / 0.494 | | BANK / 0.649 | | CENSUS / 0.969 | |
|---|---|---|---|---|---|---|---|
| Without $L_2$ normalization | | Cost | Bal | Cost | Bal | Cost | Bal |
| FCA ($K$-means++) | | 1.875 | 0.492 | 1.859 | 0.647 | 33.472 | 0.959 |
| FCA ($K$-means random) | | 1.882 | 0.489 | 1.864 | 0.644 | 32.913 | 0.960 |
| FCA (Gradient-based) | | 1.943 | 0.490 | 1.967 | 0.646 | 34.121 | 0.962 |

**Initialization of cluster centers** Moreover, we empirically assess the stability of FCA and FCA-C with respect to the different initial cluster centers (while keeping the initialization algorithm fixed to $K$-means++), and compare them with the standard $K$-means++ algorithm. For FCA and the standard $K$-means++ algorithm, we run each algorithm five times with five different random initial centers. For FCA-C, we use five random initial centers for each $\varepsilon \in \{0.1, 0.15, \dots, 0.85, 0.9\}$ and compute the averages as well as standard deviations. Then, we divide the standard deviation by average 17 times (corresponding to 17 $\varepsilon$s), and take average.

Table 16 below reports the coefficient of variation (= standard deviation ÷ average) of Cost and Bal. The results show that the variations of all the three algorithms are similar, indicating that FCA and FCA-C are as stable as the standard $K$-means++ with respect to the choice of initial cluster centers. That is, aligning data (i.e., finding the optimal coupling matrix $\Gamma$) to build fair clustering does not affect the stability of the overall algorithm.

*Table 16.* Standard deviations divided by averages (i.e., coefficient of variation) with respect to five random different choices of initial centers.

| FCA | ADULT | | BANK | | CENSUS | |
|---|---|---|---|---|---|---|
| | Cost | Bal | Cost | Bal | Cost | Bal |
| with $L_2$ | 0.012 | 0.001 | 0.093 | 0.006 | 0.004 | 0.001 |
| without $L_2$ | 0.010 | 0.001 | 0.081 | 0.003 | 0.006 | 0.003 |

| FCA-C | ADULT | | BANK | | CENSUS | |
|---|---|---|---|---|---|---|
| | Cost | Bal | Cost | Bal | Cost | Bal |
| with $L_2$ | 0.015 | 0.001 | 0.083 | 0.007 | 0.011 | 0.004 |
| without $L_2$ | 0.011 | 0.001 | 0.088 | 0.002 | 0.010 | 0.002 |

| $K$-means++ | ADULT | | BANK | | CENSUS | |
|---|---|---|---|---|---|---|
| | Cost | Bal | Cost | Bal | Cost | Bal |
| with $L_2$ | 0.009 | 0.001 | 0.078 | 0.005 | 0.008 | 0.002 |
| without $L_2$ | 0.011 | 0.000 | 0.090 | 0.004 | 0.010 | 0.002 |

### C.3.5. ABLATION STUDY: (3) CONSISTENT OUTPERFORMANCE ACROSS VARIOUS $K$ (SECTION 5.4)

We analyze the impact of $K$ to the performance of four FC algorithms (FCBC, SFC, FRAC, and FCA). On ADULT dataset, we evaluate the FC algorithms with $K \in \{5, 10, 20, 40\}$. The results are presented in Figure 8 below, which show that FCA outperforms existing FC algorithms across all values of $K$. Specifically, FCA achieves lower values of Cost than baselines for most values of $K$, while maintaining the highest fairness level: Bal $\approx$ Bal$^\star$.

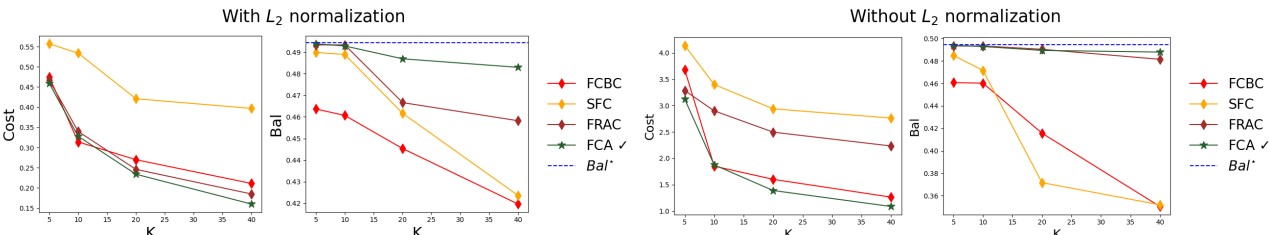

*Figure 8.* Performance comparison of FC algorithms in terms of Cost and Bal for $K \in \{5, 10, 20, 40\}$. (Left two, Right two) = (With $L_2$ normalization, Without $L_2$ normalization).

C.3.6. ABLATION STUDY: (4) SCALABILITY FOR LARGE-SCALE DATA (SECTION 5.4)

In this section, we evaluate the scalability of FCA for larger datasets, while the main experiments in Section 5 are conducted on real datasets with sample sizes of around $20,000$ to $40,000$. In specific, we apply FCA on a synthetic dataset with an extremely large sample size (around a million).

**Large-scale dataset generation**    We generate a large ($n = 10^6$) synthetic dataset in $\mathbb{R}^d$ using a $J$-component Gaussian mixture, as follows:

1. (Mean vectors) We sample $J$ many $d$-dimensional vectors $m_j, j \in \{1, \ldots, J\}$ from a uniform distribution Unif$(-20, 20)$. To ensure diversity, the distance between any two vectors is constrained to be at least 1. These vectors are used as the mean vectors for the Gaussian components.

2. (Covariance matrices) Each $j$th Gaussian component is assigned a covariance matrix $\Sigma_j = \sigma_j^2 \mathbb{I}$, where $\sigma_j \sim$ Unif$(1, 3)$.

3. (Weights) Component weights, denoted as $\phi_j, j \in \{1, \ldots, J\}$, are sampled from a Dirichlet distribution Dirichlet$(\alpha_1, \ldots, \alpha_J)$ for given parameters $\alpha_1, \ldots, \alpha_J$.

4. (Completion) The Gaussian mixture model is completed as $\sum_{j=1}^{J} \phi_j \mathcal{N}(m_j, \Sigma_j)$. We set $J$ as an even number, and sample data for $S = 0$ from $J/2$ components and for $S = 1$ from the remaining $J/2$ components.

Using this procedure, we construct a dataset with $n = 10^6, d = 2, J = 20$, and $\alpha_j = 1, \forall j \in \{1, \ldots, J\}$. The resulting generated dataset contains 320,988 samples for $S = 0$ and 679,012 samples for $S = 1$. The features are then scaled to have zero mean and unit variance. Figure 9 provides a visualization of this synthetically generated dataset.

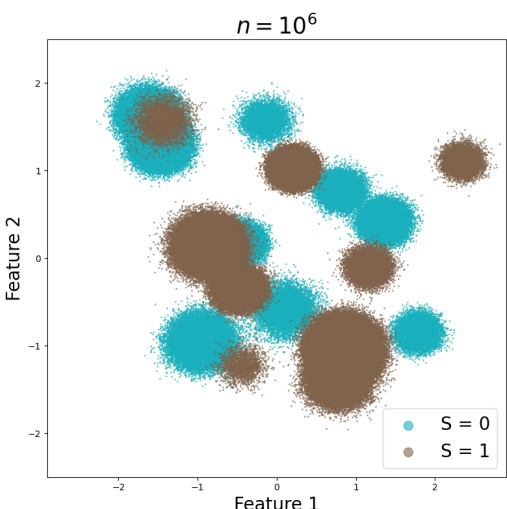

*Figure 9.* The large synthetic dataset generated with $n = 10^6, d = 2, J = 20$, and $\alpha_j = 1, \forall j \in \{1, \ldots, J\}$.

**Results**   We fix the number of clusters to $K = 10$. In this analysis, we compare only FCA and VFC, as other baselines incur extremely high computational costs for this dataset. Table 17 presents the results, demonstrating that FCA is easily scaled-up and remains a favorable FC algorithm for this large-scale dataset. That is, FCA successfully achieves near-perfect fairness (i.e., `Bal` $= 0.472 \approx 0.473$). In contrast, VFC fails to achieve near-perfect fairness, with a limit of `Bal` $0.114$. Meanwhile, FCA-C achieves a lower `Cost` than VFC ($0.107 < 0.111$), while attaining the maximum achievable `Bal` for VFC ($\approx 0.114$).

Table 17. Comparison of `Cost` and `Bal` between FCA (or FCA-C) and VFC on the large synthetic dataset in Figure 9.

| $\text{Bal}^\star = 0.473$ | Cost ($\downarrow$) | Bal ($\uparrow$) |
|---|---|---|
| Standard (fair-unaware) | 0.058 | 0.000 |
| VFC (Ziko et al., 2021) ($\lambda = 51000$) | 0.111 | 0.114 |
| FCA-C ($\varepsilon = 0.65$) ✓ | 0.107 | 0.115 |
| FCA ✓ | 0.669 | 0.472 |

We further analyze the computation times of FCA on four datasets with different $n$s, including this large-scale dataset. Table 18 shows that the computation time scales linearly with $n$ (i.e., Average time / $n$ is nearly constant). This observation aligns with our discussion on the computational complexity of $\mathcal{O}(nm^2)$ when applying the partitioning technique with a fixed $m = 1024$ (see Section 4.3). These results can further highlight the empirical efficiency of the partitioning technique.

Table 18. Comparison of total computation times (seconds) of FCA, on four different datasets with different $n$s. The reported results are averages and standard deviations, based on five runs.

| | ADULT | BANK | CENSUS | Large synthetic (Figure 9) |
|---|---|---|---|---|
| Partition size $m$ | 1024 | 1024 | 1024 | 1024 |
| $n$ | 32,561 | 41,108 | 20,000 | 1,000,000 |
| Average time (Standard deviation) | 56.7 (3.9) | 73.1 (7.8) | 32.2 (5.4) | 1410.1 (115.8) |
| Average time / $n$ | $1.7 \times 10^{-3}$ | $1.8 \times 10^{-3}$ | $1.6 \times 10^{-3}$ | $1.4 \times 10^{-3}$ |

### C.3.7. ABLATION STUDY: (5) LINEAR PROGRAM VS. SINKHORN FOR OPTIMIZING $\mathbb{Q}$ (SECTION 5.4)

We evaluate an alternative algorithm for finding the coupling matrix $\Gamma$. Specifically, the Sinkhorn algorithm of Cuturi (2013) optimizes $\mathbf{C} + \lambda \operatorname{ent}(\Gamma)$, where $\mathbf{C}$ is the cost matrix defined in Phase 1 of Section 4.1, $\lambda$ is a regularization parameter, and $\operatorname{ent}(\Gamma)$ denotes the entropy of $\Gamma$. Note that, as $\lambda$ increases, $\Gamma$ approaches a uniform matrix. It is well-known that the Sinkhorn algorithm generally yields a more relaxed solution with reduced runtime compared to solving the Kantorovich problem via linear programming.

We compare (i) the original FCA (using linear programming) and (ii) FCA with the Sinkhorn algorithm for $\lambda \in \{0.01, 0.1, 1.0\}$. Table 19 shows the results, suggesting: (i) A small regularization ($\lambda = 0.01$) achieves performance comparable to linear programming with a slight runtime reduction ($\approx 2\%$). (ii) A large regularization ($\lambda = 1.0$) significantly degrades performance while reducing runtime ($\approx 10\%$). Overall, careful tuning of $\lambda$ is critical when using the Sinkhorn algorithm, while the runtime reduction may not be significant in practice. Therefore, we recommend solving the linear program for FCA.

Table 19. Comparison of using the Sinkhorn algorithm and solving the linear program, in terms of `Cost`, `Bal`, and runtime per iteration.

| ADULT | | | |
|---|---|---|---|
| $\text{Bal}^\star = 0.494$ | Cost ($\downarrow$) | Bal ($\uparrow$) | Runtime / iteration (sec) |
| FCA (Sinkhorn, $\lambda = 1.0$) | 0.350 | 0.271 | 4.98 |
| FCA (Sinkhorn, $\lambda = 0.1$) | 0.315 | 0.463 | 5.12 |
| FCA (Sinkhorn, $\lambda = 0.01$) | 0.330 | 0.491 | 5.55 |
| FCA (Linear program) | 0.328 | 0.493 | 5.67 |

## C.3.8. Comparisons of computation time

**FCA versus FCA-C** We compare the computation times of FCA and FCA-C, as FCA-C technically involves an additional step of optimizing $\mathcal{W}$. Table 20 below shows that while FCA-C requires slightly more computation time than FCA, the increase is not substantial (a maximum of 3.7%).

*Table 20.* Comparison of computation times (seconds) of FCA and FCA-C per each iteration. The averages and standard deviations are calculated based on five runs. The data are $L_2$-normalized and the batch size is fixed as 1024.

| Average (Standard deviation) | ADULT | BANK | CENSUS |
|---|---|---|---|
| FCA | 5.67 (0.39) | 7.31 (0.78) | 16.10 (2.70) |
| FCA-C | 5.72 (0.47) | 7.58 (0.32) | 16.46 (1.23) |
| Increase (FCA-C / FCA) | 0.5% ↑ | 3.7% ↑ | 2.2% ↑ |

**FCA versus SFC** We here additionally consider two more scenarios to compare FCA and SFC. In this analysis, the data are $L_2$-normalized and the batch size is fixed as 1024 for FCA. Note that FCA consists of an outer iteration (updating the cluster centers and joint distribution alternately), and an inner iteration when applying the $K$-means algorithm in the aligned space.

1. We compare the number of iterations until convergence. For SFC, we calculate the number of iterations in the $K$-means algorithm (after finding fairlets). For FCA, we calculate the sum of the number of iterations (inner iteration) in the $K$-means algorithm for each outer iteration (updating the cluster centers and joint distribution). In this analysis, note that we report the elapsed time for FCA with 10 iterations of the outer iteration, because the performance of FCA with 10 iterations of the outer iteration is not significantly different from FCA with 100 iterations of the outer iteration.

    As a result, SFC requires smaller number of iterations, compared to FCA (see Table 21), primarily because FCA involves the outer iterations. On the other hand, in FCA, the number of total iterations is almost linear with respect to the number of iterations in each inner loop.

*Table 21.* Comparison of computational costs between FCA and SFC: the total number of iterations of the standard $K$-means algorithm in SFC and FCA.

| Total number of iterations | ADULT | BANK | CENSUS |
|---|---|---|---|
| SFC (Backurs et al., 2019) | 15 | 10 | 32 |
| FCA | 57 | 110 | 93 |

2. To further analyze whether applying an early-stopping to FCA can maintain reasonable performance, we conduct an additional experiment: we fix the number of $K$-means iterations to 1 per outer iteration of FCA, then perform a total of 10 outer iterations. With this setup, the total number of iterations for FCA becomes 10, which is comparable to or smaller than that of SFC (15 for Adult, 10 for Bank, and 32 for Census dataset, as shown in Table 21). We observe that the performance of FCA with this early-stopping is slightly worse than the original FCA (where the $K$-means algorithm runs until convergence), but it still outperforms SFC (see Table 22). However, this early-stopping approach would be not recommended, at least for the datasets used in our experiments. This is because running the $K$-means algorithm takes less than a second or a few seconds, while the computation of finding the joint distribution dominates the overall runtime. Additionally, the original FCA slightly outperforms the early-stopped version with only 10 iterations.

*Table 22.* Performance comparison of SFC, FCA with a total of 10 iterations, and the original FCA (updating until convergence). The data are not $L_2$-normalized.

| Dataset / `Bal`* | ADULT / 0.494 | | BANK / 0.649 | | CENSUS / 0.969 | |
|---|---|---|---|---|---|---|
| Performance | Cost (↓) | Bal (↑) | Cost (↓) | Bal (↑) | Cost (↓) | Bal (↑) |
| SFC (Backurs et al., 2019) | 3.399 | 0.471 | 3.236 | 0.622 | 69.437 | 0.940 |
| FCA (total 10 iterations) | 1.923 | 0.489 | 1.992 | 0.644 | 33.967 | 0.955 |
| FCA (original) | 1.875 | 0.492 | 1.859 | 0.647 | 33.472 | 0.959 |

C.3.9. COMPARISON OF FCA AND SFC BASED ON $K$-MEDIAN CLUSTERING COST

As SFC is originally designed for the $K$-median clustering objective, we compare FCA and SFC based on the $K$-median clustering cost (i.e., $L_1$ cost) for a more fair comparison. In specific, FCA for $K$-median clustering cost is modified as follows: (i) The $L_2$ norm in eq. (3) is replaced by the $L_1$ norm. (ii) The cluster centers are found by minimizing the $L_1$ distance, as we discuss in Appendix A.4.

The results are presented in Table 23, which shows that FCA still outperforms SFC. It implies that the fairlet-based method still may not always find the most effective matching in view of clustering utility (cost), even when the given clustering objective more suited to fairlet-based approaches (e.g., $L_1$ norm) is considered.

Let $\texttt{Cost}_1 = \frac{1}{n} \sum_{(\mathbf{x},s) \in \mathcal{D}} \|\mathbf{x} - \mu_{k(\mathbf{x},s)}\|_1$ be the $K$-median clustering cost.

*Table 23.* Comparison of $\texttt{Cost}_1$ and $\texttt{Bal}$ of FCA and SFC. The data are not $L_2$-normalized.

| Dataset / $\texttt{Bal}^\star$ | ADULT / 0.494 | | BANK / 0.649 | | CENSUS / 0.969 | |
|---|---|---|---|---|---|---|
| Performance | $\texttt{Cost}_1$ ($\downarrow$) | $\texttt{Bal}$ ($\uparrow$) | $\texttt{Cost}_1$ ($\downarrow$) | $\texttt{Bal}$ ($\uparrow$) | $\texttt{Cost}_1$ ($\downarrow$) | $\texttt{Bal}$ ($\uparrow$) |
| Standard (fair-unaware) | 1.788 | 0.206 | 1.989 | 0.391 | 21.402 | 0.030 |
| SFC (Backurs et al., 2019) | 2.979 | 0.471 | 3.056 | 0.622 | 29.597 | 0.940 |
| FCA ✓ | 2.032 | 0.492 | 2.383 | 0.647 | 22.927 | 0.959 |

