# OpenReview forum: "Fair Clustering via Alignment"
_ICML.cc/2025/Conference — ICML 2025 poster_

### Official Review · Reviewer_Nujs · 2025-03-08

**Overall Recommendation:** 3

**Summary:**

The paper introduces an in-processing fair clustering approach that matches two instances from different protected groups and assigns them to the same cluster. The approach directly minimizes the clustering cost with respect to both the matching map and cluster centers simultaneously. The proposed method is theoretically proven and evaluated on three datasets and four baselines/competitors. The experimental results show that the proposed method outperforms the baselines and effectively controls the trade-off at multiple fairness levels.
## update after rebuttal

**Claims And Evidence:**

The decomposition process is proved theoretically, and the experimental results support their claim. However, in the experiments, the authors do not compare the results of proposed method with the method of Chietichetti et al. (2017).

**Essential References Not Discussed:**

None

**Experimental Designs Or Analyses:**

In general, the experimental results are promising. However, 3 datasets are not yet appropriate  to make a comprehensive assessment. Besides, the selection of features (on each dataset) is not yet well explained. For example, the feature “fnlwgt” (Adult dataset) is usually ignored in many relevant works.

**Methods And Evaluation Criteria:**

The proposed approach is technically sound. However, the authors do not use the well-known measures to evaluate the clustering quality.

**Other Comments Or Suggestions:**

None

**Other Strengths And Weaknesses:**

The idea of matching data from different protected groups is similar to the fairlet-based method, but the authors clarified their contrast and superiority.

**Questions For Authors:**

Please refer to my comments above.

**Relation To Broader Scientific Literature:**

The paper introduces a new in-processing fair clustering method which outperforms the baselines in terms of performance and complexity.

**Theoretical Claims:**

I have checked the proofs of theoretical claims and they are sound.

---

> ### Author Rebuttal · Authors · 2025-04-01
>
> *To reviewer Nujs: We sincerely appreciate your review and thank you for the opportunity to improve our work. Please refer to our point-by-point responses below.*
>
> ------
>
> ### Claims And Evidence
>
> > 1: However, in the experiments, the authors do not compare the results of proposed method with the method of Chietichetti et al. (2017).
>
> - In the current version, instead of directly comparing with Chietichetti et al. (2017), we compared FCA with SFC (Backurs et al., 2019), which developed a more scalable fairlet-based method.
>     Notably, Backurs et al., (2019) reported SFC's competitive/better performance when compared to Chietichetti et al. (2017) across three real datasets.
> - However, in light of your suggestion, we have newly conducted experiments comparing FCA and Chietichetti et al. (2017).
>     The results are provided below, showing ourperformance of FCA, which we will add to Appendix C.3.1 in the camera-ready version.
>
>     | Dataset / Bal*  | Adult / 0.494   |  | Bank / 0.649    |  | Census / 0.969  |  |
>     |:-------|:---:|:---:|:---:|:---:|:---:|:---:|
>     |           | Cost (↓)       | Bal (↑)      | Cost (↓)       | Bal (↑)      | Cost (↓)       | Bal (↑)      |
>     | Chietichetti et al. (2017)  | 0.507           | 0.488        | 0.378           | 0.639        | 1.124           | 0.941        |
>     | SFC        | 0.534           | 0.489        | 0.410           | 0.632        | 1.015           | 0.937        |
>     | FCA ✓      | **0.328**           | **0.493**        | **0.264**           | **0.645**        | **0.477**           | **0.962**        |
>
> ------
>
> ### Methods And Evaluation Criteria
>
> > 1:  However, the authors do not use the well-known measures to evaluate the clustering quality.
>
> - The reason why we consider the `cost' for measuring clustering quality is because it is the clustering objective function to be minimized.
> - **(Comparison in terms of the silhouette score)**
>     However, as you suggested, measuring the cost alone may not fully capture the clustering quality, we additionally consider another measure, called the silhouette score.
>     The silhouette score is computed as the average of $ (d_{\text{near}} - d_{\text{intra}}) / \max(d_{\text{intra}}, d_{\text{near}})$ over all data points, where $d_{\text{intra}}$ denotes the intra-cluster distance and $d_{\text{near}}$ represents the average distance to the nearest neighboring cluster.
>
>     Surprisingly, the results in the following table show that **FCA is also superior or competitive to baselines in terms of the silhouette score**.
>     We will add it to Appendix C.3.1 of the camera-ready version.
>
>     | Adult / Bal* = 0.494 | Silhouette (↑) | Bal (↑)      |
>     |:---:|:-----:|:---:|
>     | Standard (fair-unaware) | 0.227          | 0.223        |
>     | FCBC  | 0.173          | 0.443        |
>     | SFC   | 0.071          | 0.489        |
>     | FRAC  | 0.156          | 0.490        |
>     | FCA ✓ | **0.176**          | **0.493**        |
>
>
> ------
>
> ### Experimental Designs Or Analyses
>
>
> > 1: However, 3 datasets are not yet appropriate ... Besides, the selection of features (on each dataset) is not yet well explained...
>
> - **(The number of datasets)**
>     Please note that **we analyzed 6 datasets in total**.
>     That is, not only the three tabular datasets, but we also analyzed on two image datasets (Section 5.3 and Appendix C.3.2) and a large synthetic dataset (Appendix C.3.7).
> - **(Analysis on an additional dataset)**
>     However, in light of your comment, we additionally conducted an analysis on Credit Card dataset from Yeh and Lien (2009), which was also used in Bera et al., (2019) and Harb \& Lam, (2020).
>     We used gender as the sensitive attribute and set $K = 10.$
>     The results are provided in the table below (to be added to Appendix in the camera-ready version), showing the outperformance of FCA.
>
>     |  CreditCard / Bal* = 0.656   | Cost (↓)           | Bal (↑)         |
>     |:---:|:------:|:---:|
>     | Standard (fair-unaware) | 0.392  | 0.506           |
>     | FCBC  | 0.492  | 0.629           |
>     | SFC   | 0.682  | **0.653**           |
>     | FRAC  | 0.510  | 0.649           |
>     | FCA ✓ | **0.402**  | **0.653**           |
>
>     (References)
>     Yeh and Lien (2019): The comparisons of data mining techniques for the predictive accuracy of probability of default of credit card clients. Expert Systems with Applications, 2009.
>
> - **(About the feature selection)**
>     Existing fair clustering works use **continuous variables** as features (Backurs et al. 2019; Bera et al., 2019; Esmaeili et al., 2021; Ziko et al., 2021), so we have made the same choice.
>     For reference, the features we consider in this paper are the same as VFC (Ziko et al., (2021)).
>     Furthermore, please note that the feature `final weight (fnlwgt)' has also been selected in the prior works (Esmaeili et al., 2021; Ziko et al., 2021).
>
>     We will add the above detailed explanation to Appendix C.1 in the camera-ready version.

---

> > ### Comment · Reviewer_Nujs · 2025-04-05
> >
> > Thank you very much for your clarification. I will keep my score.

---

### Official Review · Reviewer_4J8b · 2025-03-13

**Overall Recommendation:** 3

**Summary:**

This paper introduces a fair clustering method called Fair Clustering via Alignment (FCA), which aims to balance the trade-off between fairness and clustering utility. The authors propose a decomposition of the fair k-means clustering objective into two components: the transport cost and the clustering cost. The key idea is to alternately align data from different protected groups into a common space and optimize cluster centers in this aligned space. The authors claim that FCA theoretically guarantees approximately optimal clustering utility for any given fairness level without complex constraints. Empirical results demonstrate the effectiveness of FCA in terms of both fairness and clustering utility.

**Claims And Evidence:**

The claims made in the paper are generally well-supported by both theoretical analysis and empirical results. The authors provide a clear theoretical foundation for their decomposition of the fair clustering objective, and they validate their claims through extensive experiments on benchmark datasets.

**Essential References Not Discussed:**

There are no critical references missed.

**Ethical Review Concerns:**

This paper may need an ethics review because it deals with algorithmic fairness in clustering, which directly relates to issues of discrimination and bias in machine learning.
1. It addresses clustering with respect to sensitive attributes such as race and gender. While the goal is to ensure fairness, there is a potential risk that the algorithm could inadvertently reinforce or mask existing biases if the underlying data contains structural inequalities.
2. It defines fairness based on proportional balance between groups, but it does not address potential fairness concerns beyond this measure.

**Ethical Review Flag:**

Flag this paper for an ethics review.

**Ethics Expertise Needed:**

["Discrimination / Bias / Fairness Concerns"]

**Experimental Designs Or Analyses:**

The experimental design is sound and comprehensive. The authors compare FCA against several baseline methods across multiple datasets, including pre-processing, in-processing, and post-processing approaches.

**Methods And Evaluation Criteria:**

The proposed method, FCA, is well-motivated and makes sense for the problem of fair clustering. The authors use a combination of optimal transport theory and standard clustering algorithms to align data from different protected groups and optimize cluster centers. The evaluation criteria (Cost and Balance) and the benchmark datasets are widely used.

**Other Comments Or Suggestions:**

1. It is better to include a more detailed discussion on the limitations of the approach, particularly in terms of scalability and applicability to non-binary sensitive attributes.
2. Consider discussing potential strategies to further reduce the computational cost of solving the Kantorovich problem.
3. Expanding the approach to clustering with non-Euclidean distance metrics is better.

**Other Strengths And Weaknesses:**

Strengths:
1. The alignment-based approach introduces a novel and effective method for enforcing fairness in clustering.
2. FCA provides both theoretical guarantees and practical improvements over existing clustering methods.
3. FCA-C enhances the method's flexibility by enabling fairness–utility trade-offs.
The empirical evaluation is thorough, covering diverse datasets and baseline methods.

Weaknesses:
1. The computational complexity of solving the Kantorovich problem may hinder scalability to very large datasets.
2. The method requires careful tuning of the fairness control parameter in FCA-C for optimal performance.
3. The paper primarily addresses binary sensitive attributes, with limited exploration of extending FCA to multiple protected groups.

**Questions For Authors:**

Please refer to the weakness and the other comments.

**Relation To Broader Scientific Literature:**

The paper builds on established research in fair clustering, particularly extending fairlet-based methods and in-processing approaches.

**Theoretical Claims:**

The theoretical claims are supported by rigorous proofs, which are provided in the supplementary material. The key theoretical contribution is the decomposition of the fair clustering objective into transport and clustering costs, which is proven in Theorem 3.3. The authors also provide an approximation guarantee for their algorithm in Theorem 4.3.

---

> ### Author Rebuttal · Authors · 2025-04-01
>
> *To reviewer 4J8b: We sincerely appreciate your review and the opportunity to improve our work. Please refer to our point-by-point responses below.*
>
> ------
> ### Weaknesses
>
> > 1: The computational complexity ...
> - **Section 4.3 introduces a partitioning technique to reduce the computational complexity.**
> Our experiments (Figures 6–7 and Table 9 in Appendix C.3.3) demonstrate that it yields reasonable results and significant runtime reduction.
> To evaluate scalability, **we already conducted experiments on a dataset of one million data points, showing that FCA outperforms VFC (Appendix C.3.7).**
>
> > 2: The method requires ...
> - We think that $\epsilon$ is not a tuning parameter, but $\epsilon$ itself serves as a fairness level in FCA-C.
> In particular, on Page 5 we define $\mathbf{A}\_{\epsilon}$ as the set of assignment functions where the sum of unfairness across clusters is bounded by $\epsilon.$
> FCA-C always converges to a local minimum for any given $\epsilon.$
> - On the other hand, Proposition 3.2 implies that we can control balance by controlling $\epsilon.$
>
> > 3: The paper primarily addresses ...
> - We have already discussed extending FCA to handle multiple groups (see Appendix A.3).
> To numerically validate this extension, we newly conducted an analysis using Bank dataset with 3 groups single/married/divorced (following VFC (Ziko et al., 2021)).
> The results are in the table below (to be added to Appendix A.3): **FCA can practically handle multiple groups and outperforms VFC** (a lower cost $0.222 < 0.228$ with a higher balance $0.182 > 0.172$).
>
> | Bank | 3 groups (single/married/divorced)| |
> |:---:|:---:|:--:|
> | Bal* = 0.185  | Cost (↓)| Bal (↑)|
> | VFC   | 0.228  | 0.172|
> | FCA ✓ | **0.222**   | **0.182** |
>
> ------
> ### Other Comments Or Suggestions
> > 1: It is better ...
> - Please see our response to 'Weaknesses' above.
>
> > 2: Consider discussing ...
> - **(Sinkhorn algorithm)**
>     We also applied the Sinkhorn algorithm and compared with solving the Kantorovich problem using the linear programming.
>     The Sinkhorn algorithm of Cuturi et al., (2013) optimizes  $\mathbf{C} + \lambda \cdot ent(\Gamma),$ where $\mathbf{C}$ is the cost matrix defined in Section 4.1, $\lambda$ is a regularization parameter, and $ent(\Gamma)$ denotes the entropy of the coupling matrix $\Gamma.$
>
>     The result table below suggests that:
>     careful tuning of $\lambda$ is crucial, while the runtime reduction may not be significant in practice ($\lambda = 0.01$) and a large regularization ($\lambda = 1.0$) significantly degrades performance while reducing runtime (10\% decrease).
>     This experiment also suggests that reducing computational cost (beyond the partitioning technique) for solving the Kantorovich problem would be difficult but is a promising future study.
>     We will include this discussion in Appendix.
>     | Adult / Bal* = 0.494 | Cost (↓) | Bal (↑) | Runtime / iteration (sec) |
>     |:---:|:---:|:---:|:---:|
>     | FCA (Sinkhorn, λ = 1.0) | 0.350 | 0.271 | 4.98|
>     | FCA (Sinkhorn, λ = 0.1) | 0.315 | 0.463 | 5.12|
>     | FCA (Sinkhorn, λ = 0.01) | 0.330 | 0.491 | 5.55|
>     | FCA (Linear program) | 0.328 | 0.493 | 5.67|
>
>     (References)
>     Cuturi et al., (2013): Sinkhorn Distances: Lightspeed Computation of Optimal Transport
>
> > 3: Expanding ...
> - In Appendix A.4, we already showed that FCA can be similarly applied to $L_{p}$ norms for all $p \ge 1.$
> Experimentally, Appendix C.3.9 (Table 17) shows that under the $K$-median ($L_{1}$ norm) setting, FCA outperforms the fairlet-based method SFC.
> - During the rebuttal, **we realized that the argument for the extension to $L_{p}$ norm in Appendix A.4 can be applied to any distance satisfying the triangle inequality**.
> We will add this comment in the camera-ready version.
>
> ------
>
> ### Ethical Review Concerns
> > 1: This paper may need an ethics review ...
> - Our paper primarily aims to **achieve proportional (group) fairness in clustering, in line with many existing works**.
> We acknowledge that focusing solely on proportional fairness may overlook other fairness notions.
> In light of your feedback, **we will add the following paragraph to our Impact Statement in the camera-ready version**:
> - **(A paragraph to be added to `Impact Statement')**
>
>     *While this work focuses on proportional (group) fairness, we acknowledge that such an approach may overlook other fairness notions, such as individual fairness or social fairness (e.g., balancing clustering costs across groups), which are not explicitly addressed within the proportional fairness framework.
>     We therefore encourage readers to interpret our results with this scope in mind: our goal is to improve the trade-off between clustering cost and proportional fairness level, but care must be taken when considering the broader implications for other fairness notions.
>     In conclusion, we anticipate future studies that integrate fair clustering research with various notions of fairness*.

---

> > ### Comment · Reviewer_4J8b · 2025-04-05
> >
> > Thanks for the rebuttal made by the authors. I will keep my score.

---

### Official Review · Reviewer_AM68 · 2025-03-14

**Overall Recommendation:** 3

**Summary:**

In group fair clustering, a clustering objective is optimized in light of a group fairness constraint. If each data point is assigned some class, we require the proportions of these classes in each cluster to be the same as, or close to, their proportions out of the total dataset.

This work considers the problem of group fair clustering using in-processing algorithms to simultaneously find a fair alignment between points and a good k-means clustering. They propose an algorithm which uses an alignment, or a joint probability distribution over points of two different classes, together with a select set of centers to design an objective that optimizes both at the same time. While either the alignment or the set of centers is fixed, the other is optimized. The algorithm iterates, alternating between the two, to find a solution.

Alignment is found using optimal transport methods from Kantorovich. The center selection is performed via one of many clustering techniques, including K-means and gradient descent. A variant of this algorithm that allows for non-perfect fairness is also introduced as a paramterized option.

On the theory side, they show that their designed combined objective is equivalent to optimizing for the clustering objective in light of the fairness constraint. This can then be used to show their parameterized algorithm approximates the clustering objective with small violation to the fairness constraint. The approximation factor is within a constant factor of the vanilla clustering algorithm used, and the violation is linearly dependent on an error constraint epsilon.

This paper also validates their findings with experiments. They select a handful of baseline fair clustering algorithms that represent in-processing, pre-processing, and post-processing fairness techniques. These experiments show that, on tested cases, their algorithm provides the best fairness-cost tradeoff. It is also numerically stable and does not require too much additional computation time.

**Claims And Evidence:**

Their claims are supported by theoretical and empirical evidence. Theoretical evidence comes as proofs deferred to the appendix. Experiments are partially in the appendix but the major results are shown in the paper to support their findings.

**Essential References Not Discussed:**

None that I am aware of

**Experimental Designs Or Analyses:**

I did not find any issues

**Methods And Evaluation Criteria:**

The methods seemed to have no issues.

**Other Comments Or Suggestions:**

N/A

**Other Strengths And Weaknesses:**

Strengths
- New methods
- Formulations very nicely represent the original problem
- Technically involved
- Experimentally verified

Weaknesses
- Extremely notationally dense, with a fair amount of notation that is not introduced properly (it generally assumes a pretty strong understanding of advanced probability and its notation)
- New methods aren't entirely novel, they bear some similarities to existing methods

**Questions For Authors:**

1. Would you say your techniques are sort of like iterated preprocessing? You do have a kind of "before" and "after" part of the algorithm, where before attempts to achieve fairness and after then selects centers accordingly.

2. Can you clarify what you mean by numeric stability?

3. How does the epsilon parameter control a tradeoff? In theorem 4.3, I only see it affecting the fairness violation.

4. Are the benchmark clustering methods designed for K-means? I know much of fair clustering literature has looked at k-center and k-median instead of k-means.

**Relation To Broader Scientific Literature:**

Fair clustering is a relatively new yet also pretty extensively studied area of research. Many methods have been proposed, particularly for group fair clustering. To my knowledge, less has been done regarding group fair k-means, which is the problem this paper approaches. This paper additionally proposes a new way to combine the fairness constraint and clustering objective into a single objective that turns out to be equivalent to the given problem. This allows them to iteratively optimize the two terms of the combined objective to arrive on a solution. This is somewhat like a repeated pre-processing method, which is interesting in its own right. Combined objectives have also been studied before, but in the context of a different formulated problem, where fairness is not a constraint but given to be part of the objective.

**Theoretical Claims:**

Theoretical claims were believable, but for the most part, proved in the appendix. I did not check the appendix.

---

> ### Author Rebuttal · Authors · 2025-04-01
>
> *To reviewer AM68: We sincerely appreciate your review and thank you for the opportunity to improve our work. Please refer to our point-by-point responses below.*
>
> ------
>
> ### Weaknesses
>
>
> > 1: Extremely notationally dense ...
>
> - Answer:
>     - When preparing this work, we initially considered summation-based notations for expressions such as
>         $ \frac{1}{n\_{s}} \sum\_{i=1}\^{n\_{s}} \sum\_{k=1}\^{K} \mathcal{A}\_{s}(\mathbf{x}\_{i})\_{k} \bigg( \frac{\Vert\mathbf{x}\_{i}-\mathbf{T}(\mathbf{x}\_{i})\Vert^2}{4} + \left\Vert \frac{\mathbf{x}\_{i} + \mathbf{T}(\mathbf{x}\_{i})}{2}-\mu\_k \right\Vert\^2 \bigg), $
>         for eq. (2) in Theorem 3.1.
>         However, we decided to use the probability-based notations instead, since the readability of the proofs is much improved.
>     - We were also concerned about this and **made efforts to explain the probability-based formulations by use of summation-based notation in several instances (e.g., the clustering cost $C(\mu, \mathcal{A})$ in Section 2, the definition of balance in lines 119-120, and assignment functions of FCA in Section 4)**.
>
> > 2: New methods aren't entirely novel ...
>
> - Answer:
>     - We agree that FCA is similar to the fairlet-based methods in the sense that the idea of matching is used.
>     In this sense, FCA can be viewed as a novel extension of the fairlet-based methods.
>     - Although the fairlet-based methods also rely on matching, it would be not optimal in view of fair clustering. In contrast, **our method guarantees to find a matching map, which provides a (local) optimal fair clustering, by Theorems 3.1 and 3.3.**
>     Also, please see Remark 3.2 for further discussion.
>     - It would be hard to modify the fairlet-based methods to control fairness level (for non-perfect fairness).
>     FCA can be modified to find an (local) optimal solution to control fairness level (i.e., FCA-C algorithm).
>     - This novel extension is made possible by **the novel reformulation of the fair clustering objective function using the matching map (or alignment)**, as presented in Section 3.
>
> ------
>
> ### Questions
>
> > 1: Would you say ...
>
> - Answer:
>     - We thank you for the insightful question.
>     Yes, you are right.
>     However, in fact, **the 'before' step of our approach (i.e., finding matchings) is not solely for achieving fairness, but also simultaneously for minimizing the clustering cost**.
>     Specifically, it solves a modified Kantorovich problem, where the modification is aimed at minimizing the clustering cost.
>     - Furthermore, the proposed algorithm can be extended to control the fairness level (e.g., non-perfect fairness), while other pre-processing algorithms such as fairlet-based methods are not designed for it.
>
> > 2: Can you clarify ...
>
> - Answer:
>     - Numerical stability means that **FCA does not suffer from numerical instability compared to VFC**.
>     The main reason for this stability is that FCA does not have any regularization parameters while VFC requires to choose a fairness regularization parameter, which can make the algorithm numerically unstable.
>     - Specifically,
>     (i) FCA is more robust to dataset pre-processing, while VFC operates reliably when data points are $L_{2}$-normalized.
>     (ii) FCA can achieve near-perfect fairness, whereas VFC may fail to do so.
>     In detail, without $L_{2}$-normalization - especially in high-dimensional settings - VFC can become numerically unstable due to overflow.
>     See Appendix C.3.1 for details.
>
>
> > 3: How does the epsilon ...
>
> - Answer:
>     - **(Fairness bound by $\epsilon$)**
>     **Please refer to Proposition 4.2, which shows that the fairness level (balance) is bounded by $\epsilon.$**
>     - **(The trade-off)** The clustering cost for a larger $\epsilon$ is always lower than that for a smaller $\epsilon.$
>     That is, the optimal fair clustering with $\epsilon$ is included in $\mathbf{A}_{\epsilon'}$ for $\epsilon'>\epsilon.$
>     - **(Note: definition of $\epsilon$)**
>     $\epsilon \in [0, 1]$ represents the size of the set $\mathcal{W}$ (i.e., $\mathbb{Q}((\mathbf{X}\_{0}, \mathbf{X}\_{1}) \in \mathcal{W}) \le \epsilon$), as defined below eq. (5).
>     The set $\mathcal{W}$ consists of the aligned data points to which the standard $K$-means clustering cost is applied.
>     That is, $\epsilon$ used in the FCA-C algorithm is the fairness level: as $\epsilon$ decreases, the resulting clustering of FCA-C becomes fairer.
>
>
>
> > 4: Are the benchmark ...
>
> - Answer:
>     - Yes, you are right.
>     Among the four baseline methods we consider in our study (FCBC, SFC, FRAC, and VFC), three methods have been developed for $K$-means (as well as other $K$-clustering), while SFC is originally designed for $K$-median.
>     - **Please also note that, in Appendix C.3.9 we already discussed how to modify FCA for the $K$-median clustering setting**.
>     As shown in Table 17, FCA outperforms SFC under the $K$-median clustering setting.

---

> > ### Comment · Reviewer_AM68 · 2025-04-02
> >
> > Thank you for your clarifications, they were helpful. I don't see enough reason to change my score, however. It is a nice work that should be published somewhere, possibly ICML.

---

### Official Review · Reviewer_1gG2 · 2025-03-15

**Overall Recommendation:** 2

**Summary:**

The paper is focused on fair clustering. In particular, it suggests a new method based on decomposing the clustering cost which claims to have higher flexibility than prior methods in trading the clustering cost and fairness violations. Theoretical analysis and experiments comparing to baselines are conducted to prove these claims.

**Claims And Evidence:**

I don't find the claims convincing. Convincing evidence would be through a theorem with superior guarantees in comparison to prior work. Or experiments where the results are much better.

**Essential References Not Discussed:**

Essential References are discussed.

**Experimental Designs Or Analyses:**

-The authors can clarify this, but it does not seem that the performance of the algorithm is really much better than existing baselines. Further, some the baselines work for many groups instead of just two as done in this paper. Also, in Table 1 for Adult doesn't FCBC have a smaller cost?

**Methods And Evaluation Criteria:**

Yes. Theorems and experiments are valid ways to establish guarantees. The issue is that the arguments in both are not strong.

**Other Comments Or Suggestions:**

See weaknesses above.

**Other Strengths And Weaknesses:**

### Strengths:

-the objective of the paper is interesting.

### Weaknesses:

-the main issue I think is that this is not the first paper to trade-off the clustering cost and fairness. While some claims are made that prior work follows an "in-processing" or "post-processing" approach. The issue is that the paper does not directly establish superior performance.  For example, in Theorem 4.3 what is \epsilon? Further O(\tau) is not sufficient, what is the exact constants hidden in the O(.) notion. Prior work was specific in including the factors. How can we be convinced that this is a better algorithm if the approximation factors are not better, based on the current presentation since they are not given they may even be worse. For the experiments, please my points above on experiments above.

More points on weaknesses:

-lines (184-187) claim that previous methods are sub-optimal, but that has to be the case since the problem is NP-hard. Even this method is sub-optimal as otherwise it would be solving an NP-hard problem exactly.

-the paper assumes that there are only two groups which is a weakness in comparison to prior algorithms that can handle an arbitrary number of groups.

### Minor Points:

-presentation can be improved by:
     1-not referring to Remark B.1 on the appendix and maybe including the main intuition in the text.
     2-finding another way instead of the boxes on page 5 which are very heavy in text

**Questions For Authors:**

See weaknesses above.

**Relation To Broader Scientific Literature:**

Prior work is well-cited. Although greater connection to Ziko et al and Esmaeili at al would be good since both trade-off clustering cost and fairness.

**Theoretical Claims:**

I did not verify the correctness of any proofs

---

> ### Author Rebuttal · Authors · 2025-04-01
>
> *To reviewer 1gG2: We sincerely appreciate your review and the opportunity to improve our work. Please refer to our point-by-point responses below.*
>
> ------
> ### Claims And Evidence
> > 1: I don't find ...
> - First, we would like to highlight our main contributions:
>     - A novel reformulation of the fair clustering objective function using the matching map (Section 3).
>     - Our method guarantees finding a matching map that provides a (local) optimal fair clustering (Theorems 3.1 and 3.3).
> - For the theoretical aspect, please see our response to 'Weaknesses' below, where we suggest that **our proposed algorithm is competitive with existing methods in terms of approximation rate**.
> - For discussion on the experimental aspect, please see our response to 'Experimental Designs Or Analyses' below, which emphasizes **the empirical superiority of our proposed algorithm**.
>
> ------
> ### Experimental Designs Or Analyses
> > 1: The authors can clarify this ...
> - The main contribution of our work is the development of **a numerically stable in-processing fair clustering algorithm**.
> While in-processing algorithms such as VFC require difficult regularization parameter tuning,
> **FCA does not have any regularization, guaranteeing a (local) optimal solution without tuning** (Theorems 3.1 and 3.3).
> For example, FCA achieves near-perfect fairness and is more robust to data pre-processing than VFC (Section 5.2 and Appendix C.3.1).
> - As expected, FCA significantly outperforms the fairlet-based pre-processing algorithm SFC: achieving nearly 40% lower cost on average and higher fairness (Table 1 in Section 5.2).
> - Surprisingly, FCA performs even better on visual clustering tasks, competing with two image-specific algorithms and dominating SFC and VFC (Table 3 in Section 5.3).
>
> > 2: Further, some ...
> - Please see our response to 'Weaknesses-3' below.
>
> > 3: Also, in Table 1 ...
> - Although FCBC achieves a lower cost than FCA, it is less fair (i.e., lower balance).
> **Table 1 compares the costs at the highest fairness levels each algorithm can achieve**.
> FCBC achieves a lower cost because the highest fairness level it can achieve is lower than that of FCA.
> - To compare fairly, we obtained a fair clustering using FCA-C with a cost of 0.313 (similar to the cost of FCBC, 0.314).
> The table below shows FCA-C achieves a higher balance (0.473 vs. 0.443), suggesting FCA offers a better trade-off than FCBC.
> This result will be added to Appendix C.3.1.
>
>     | Bal* = 0.494 | Cost (↓) | Bal (↑) |
>     |:-:|:----:|:---:|
>     | FCBC| 0.314  | 0.443 |
>     | FCA-C ✓  | 0.313  | **0.473** |
>
> ------
> ### Relation To Broader Scientific Literature
> > 1: Although greater connection ...
> - We appreciate the suggestion.
> In fact, we already mentioned the two methods in Section 1 and Appendix C.2.3.
> Experimentally, FCA outperforms VFC and FCBC (Tables 1, 2, 5, and 6; Figures 4 and 5).
>
> ------
> ### Weaknesses
> > 1: the main issue ...
> - Our theoretical bound in Theorem 4.3 is not claimed to be the sharpest one, but is to claim that FCA is not much worse than the global optimal fair clustering.
> The main advantages of FCA include (i) greater numerical stability than an in-processing method VFC (Tables 2 and 6; Figures 3–5) and (ii) empirical superiority over post-processing methods (Tables 1 and 5).
> - A detailed version of Theorem 4.3 is in **Theorem B.3**:
> **FCA-C provides a $(\tau+2)$-approximate solution with a violation $3R\epsilon$, where $R$ is a bound on the data norm ($\sup_{\mathbf{x} \in \mathcal{X}} \Vert \mathbf{x} \Vert^{2} \le R$).**
> - **A comparison of the detailed approximation rate** with several existing methods:
>     - Bera et al. (2019) achieved a $(\tau + 2)$-approximation and a violation $3.$
>     **FCA-C provides the same approximation rate of $\tau + 2,$ but can attain a smaller violation** when $R = 1,$ as $\epsilon \in [0, 1].$
>     - Schmidt et al. (2018) provided a **$(5.5\tau + 1)$-approximation, which is worse than $\tau + 2$** for $\tau > 2/9,$ e.g., $\tau = 8(\log K) + 2$ for the K-means++ algorithm.
>     - We will include this discussion following Theorem 4.3 in the camera-ready version.
>     - (References) Schmidt et al. (2018): Fair Coresets and Streaming Algorithms for Fair k-means.
> - (Definition of $\epsilon$)
> **Please see our response to 'Questions'-3 for Reviewer AM68 due to space limitation.**
>
> > 2: lines (184-187) ...
> - The term 'suboptimal' in Remark 3.2 is used **not because the clustering problem is NP-hard**, but because our method guarantees finding a matching map providing a (local) optimal fair clustering (Theorems 3.1 and 3.3), whereas the matching map of the fairlet-based methods do not offer this guarantee.
>
> > 3: the paper assumes that ...
> - **Please see our response to 'Weaknesses'-3 for Reviewer 4J8b due to space limitation.**
>
> > 4: (Minor) presentation ...
> - Thank you for your careful review.
> (1) Remark B.1 will be moved to Section 3.1 and (2) the boxes will be replaced by algorithm expressions, in the camera-ready version.

---

### Decision · Program_Chairs · 2025-05-01

**Decision:**

Accept (poster)

**Comment:**

The paper proposes fair clustering with alignment (FCA), part of a family of approaches that seek to balance some form of minimization of fairness constraint or cost while also minimizing overall clustering cost (e.g., traditional k-means objective, traditional k-center objective, and so on).  That general body of work has been quite active for over ten years, so the problem fits within scope and within a greater research community in AI/ML as well as OR/MS.  Reviewers broadly appreciated the paper, both from a theory and an experimental results/comparison point of view.  The one weakly negative reviewer, 1gG2, brings up some good points about the theoretical analysis - this AC appreciates the authors' detailed rebuttal to these points, and (i) encourages the authors to update their working draft with these changes/comments and (ii) note that this AC is relatively down-weighted the weakly negative score because of that rebuttal.  All other reviewers were weakly positive.  Would also encourage the authors to consider expanding to other standard settings in fair clustering (e.g., non-Euclidean metrics, multi-group membership, etc), either in this or future work.